# Mobility-driven synthetic contact matrices as a scalable solution for real-time pandemic response modeling

Laura Di Domenico[1,7], Paolo Bosetti [2], Chiara E. Sabbatini[3,8], Lulla Opatowski [4,5] & Vittoria Colizza [3,6] ✉

Accurately capturing time-varying human behavior remains a major challenge for real-time epidemic modeling and response. During the COVID-19 pandemic, synthetic contact matrices derived from mobility and behavioral data emerged as a scalable alternative to empirical contact surveys, yet their comparative performance remained unclear. Here, we systematically evaluate synthetic and empirical age-stratified contact matrices in France from March 2020 to May 2022, comparing contact patterns and their ability to reproduce observed epidemic dynamics. While both sources captured similar temporal trends in contacts, empirical matrices recorded 3.4 times more contacts for individuals under 19 than synthetic matrices during school-open periods. The model parameterized with synthetic matrices provided the best fit to hospital admissions and best captured hospitalization patterns for adolescents, adults, and seniors, whereas deviations remained for children across both models. Neither matrix allowed models to fully reproduce serological trends in children, highlighting the challenges both approaches face in capturing their disease-relevant contacts. The weekly update of synthetic matrices enabled smoother reconstructions of hospitalization trends during transitional phases, while empirical matrices required strong assumptions between survey waves. These findings support synthetic matrices as a reliable, flexible, cost-effective operational tool for real-time epidemic modeling, and highlight the need for routine collection of age-stratified mobility data to improve pandemic response.

Mathematical models of infectious disease transmission represented a critical tool to guide real-time public health response during the COVID-19 pandemic[1,2]. However, one of the main challenges was accurately integrating changes in human behavior into transmission models[3,4]. The shifts in mobility and contact patterns produced by unprecedented social distancing measures significantly impacted the spread of SARS-CoV-2[5-7]. How to best characterize these shifts and integrate them in real-time modeling remains an open issue.

[1]Institute of Social and Preventive Medicine, University of Bern, Bern, Switzerland. [2]Mathematical Modelling of Infectious Diseases Unit, Institut Pasteur, Université Paris Cité, U1332 INSERM, Paris, France. [3]Sorbonne Université, INSERM, Pierre Louis Institute of Epidemiology and Public Health, Paris, France. [4]Institut Pasteur, Epidemiology and Modelling of Antibiotic Evasion, Université Paris Cité, Paris, France. [5]UVSQ, INSERM, CESP, Anti-Infective Evasion and Pharmacoepidemiology Team, Université Paris-Saclay, Montigny-Le-Bretonneux, France. [6]Department of Biology, Georgetown University, Washington, DC, USA. [7]Present address: Data Science Institute, Hasselt University, Hasselt, Belgium. [8]Present address: Santé publique France, French National Public Health Agency, Saint-Maurice, France. ✉e-mail: vittoria.colizza@inserm.fr

A key factor in the transmission of respiratory diseases is the pattern of social contacts between age groups[8]. This aspect can be incorporated into transmission models through a contact matrix, where each element represents the average number of daily contacts that individuals in one age group have with individuals in another age group. Starting from pioneering work in mid-2000s[9], an increasing number of studies used population-based surveys[10–12] to build static contact matrices describing age-stratified population-level mixing in European countries, as well as to generate synthetic contact matrices in other countries accounting for socio-demographical structures[13–15]. These pre-pandemic contact matrices were essential for early pandemic modeling[16–18]. But they became increasingly inadequate as the pandemic progressed and various interventions were implemented, such as school closures and remote working that affected age groups differently. Models relying on static matrices, or those assuming uniform rescaling of contacts[19–22], struggled to accurately estimate the impact of these interventions across ages, e.g. in terms of hospitalizations and seroprevalence.

Ideally, time-varying contact matrices can be constructed from repeated social contact surveys, but such surveys are resource-intensive and often difficult to implement in real-time. During the COVID-19 pandemic, the CoMix project conducted successive waves of social contact surveys in representative samples of the populations of over 20 countries in Europe[23]. Yet, only the UK continuously collected data throughout the pandemic through weekly waves[24], and only four countries (the UK[25], Belgium[26], the Netherlands[27], and Norway[28]) covered also the first pandemic wave, the one reporting the largest shifts in behavior. In France, survey data throughout the pandemic were available through SocialCov[29,30], a project collecting contact data from a convenience sample recruited online. While surveys with frequent waves, such as weekly data collection, can provide valuable information on social mixing patterns, they still present considerable challenges for real-time analysis. Processing raw survey data to construct accurate contact matrices in real-time is resource-intensive, requiring efficient data cleaning, aggregation, and interpretation to reflect dynamic behavioral shifts. When survey data were unavailable, transmission models had to rely on alternative proxies to estimate changes in social mixing.

Mobility data[31–34] proved essential[35], providing insights into movement flows and location-specific activity in response to restrictions and recommendations. Early in the pandemic, members of our team developed a novel framework to generate mobility-based synthetic contact matrices for France, capturing shifts in contact patterns driven by the epidemic and governmental measures. Initially introduced to assess the impact of the first lockdown[18], this framework was later expanded to integrate various data sources throughout the pandemic[36]. The synthetic contact matrices were constructed by applying age-specific contact reductions to the pre-pandemic contact matrix[12] based on contact setting and contact type. Google mobility data informed adjustments in workplace-related contacts, school attendance data shaped changes in contacts at school, and pandemic survey data on physical contact avoidance reduced skin-to-skin interactions. These data streams enabled us to produce weekly, real-time synthetic contact matrices throughout the pandemic[18,36–40].

Although mobility-based synthetic contact matrices offer a promising alternative for real-time modeling, their comparative performance relative to empirical contact matrices remains unexplored. Meanwhile, traditional empirical contact matrices - though widely used - are limited by infrequent updates, which constrains their applicability for dynamic modeling. This study addresses two main objectives. First, we compared synthetic contact matrices informed from mobility and behavioral data[37] and empirical matrices derived from social contact surveys[29,30], focusing on age-specific contact numbers and matrix structure over time. Second, we assessed the capacity of each matrix type to reproduce age-specific hospitalization and seroprevalence data within a COVID-19 transmission model accounting for viral variants, vaccination, and immunity. This framework enables us to evaluate how well each matrix captures disease-relevant contacts over time. By using the COVID-19 pandemic in France as a case study, we provide insight into how different matrix sources can inform real-time transmission modeling for future outbreak response.

## Results

### Comparison of contact patterns over time

We derived weekly synthetic contact matrices from March 2020 to May 2022 using pre-pandemic empirical contact data (Fig. 1a) and behavioral data collected during the pandemic. These matrices incorporated age-specific contact reductions across different locations and contact types (physical or non-physical contacts) (Fig. 1b). Workplace contacts were adjusted using Google mobility data[31] related to workplace presence (Fig. 1c), while reductions in physical contacts across all settings outside the household were informed by health protective behaviors from the French CoviPrev survey[41] (Fig. 1d). Both indicators showed a sharp decline during the first lockdown, followed by a gradual upward trend throughout the pandemic. Although they approached pre-pandemic levels by May 2022, those levels were not yet fully recovered. Contacts in other settings (school, transport, leisure) were reduced in the matrix according to school attendance data, school closure schedules and non-essential businesses (Tables S5 and S6). These data sources captured changes in behavior in response to public health measures, spontaneous adaptation to rising case numbers, and seasonal variations. Detailed methodology for matrix construction is available in the Methods and Supplementary Information.

We compared the weekly synthetic contact matrices with empirical contact matrices from seven waves of the SocialCov survey[29,30] conducted during different phases of the pandemic in France (Fig. 2 and Fig. 3a). SocialCov recruited participants through the governmental app TousAntiCovid. Since the survey sample was not representative of the French population, it was adjusted using sampling with replacement in each wave to reproduce the age and gender distribution in France. While synthetic matrices were available weekly throughout the study period, empirical matrices were only available for the seven specific SocialCov survey waves. Therefore, direct comparison between the two sources was restricted to these time points (Fig. 2).

We first compared the two matrix sources in terms of average number of contacts (overall and by age group). In the first lockdown (spring 2020), synthetic matrices estimated 3.4 average daily contacts, closely matching survey-based estimates of 3.6 (Fig. 3b). This corresponded to a 76% reduction in contacts compared to pre-pandemic values. Over time, the number of synthetic contacts gradually increased, modulated by intermittent school closures and social distancing measures, reaching 10.4 contacts by May 2022 (age-stratified trends shown in Fig. 3e, f). To contextualize these dynamics, we examined correlations with the Normalcy Index, which captures behavioral adaptation, and with the Stringency Index, which quantifies the intensity of social distancing measures. The synthetic contacts time series was highly correlated with both indices (Normalcy Index: Pearson's coefficient = 0.84, p-value $< 10^{-199}$, Fig. 3c, S11b; Stringency Index: Pearson's coefficient = 0.82, p-value $< 10^{-196}$, Fig. S11b), supporting the plausibility of the behavioral signals integrated into the synthetic matrices. This correlation reflects concurrent behavioral shifts rather than overlap in data inputs. Empirical SocialCov matrices also showed an increasing trend in the average number of contacts (correlated with both the Normalcy and Stringency indexes, Fig. S11c), but contact numbers after the first lockdown were on average nearly twice as high as synthetic estimates across the remaining six survey waves. The empirical-to-synthetic contact ratio decreased from 2.1 in

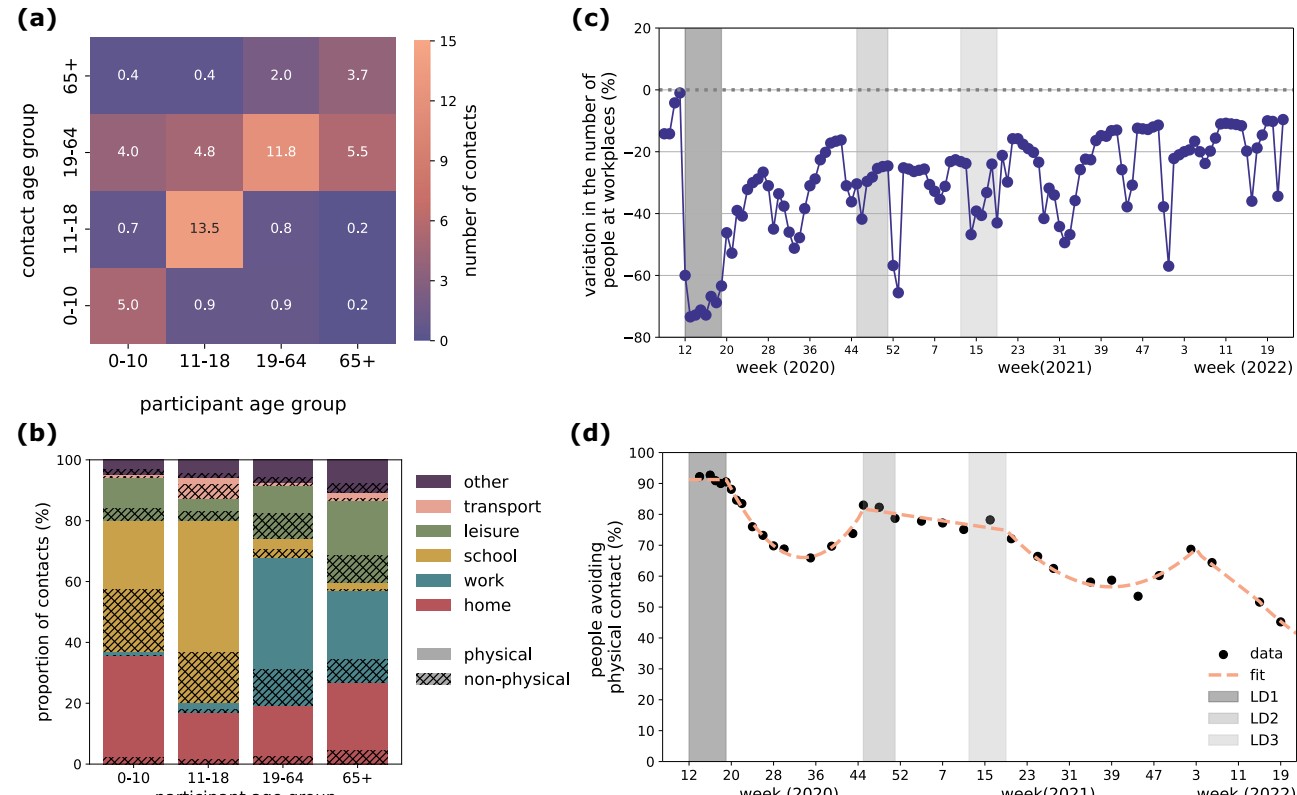

**Fig. 1 | Input data for synthetic contact matrices. a** Pre-pandemic empirical contact matrix $M$ estimated for France, for a regular weekday from ref. 12 The element $M_{ij}$ represents the average number of contacts that one individual in participant age group $i$ (x-axis) engages with individuals in the contact age group $j$ (y-axis). **b** Breakdown of the total number of contacts for each age group by location (color) and type (pattern), where physical means skin-to-skin contact. **c** Google mobility data related to workplaces[31]. The plot shows the weekly average

of the daily variation in mobility, excluding weekends. The mobility variation is computed by Google as the variation in the number of people visiting workplaces with respect to a pre-pandemic baseline. **d** Survey data (black dots) and piece-wise polynomial fit (dashed line) of the proportion of people declaring to avoid physical contacts over time, from the French survey CoviPrev[41]. In panels (**c**) and (**d**), vertical gray bars indicate the periods of the three national lockdowns in France.

December 2020 to 1.5 in May 2022, with a dip to 1.2 during summer 2021 (Fig. 3d). These differences were primarily driven by children and adolescents, with contacts being 3-4 times higher in empirical matrices compared to synthetic matrices, except during school closures in April 2020 and August 2021 when they mostly aligned (Fig. 3d, f). In contrast, contact numbers for adults and seniors were largely consistent between the two sources, with empirical-to-synthetic contact ratios ranging between 0.9 and 1.4 (Fig. 3d, e).

We then compared contact matrices in terms of mixing patterns using a set of metrics accounting for matrix elements, structure, and contact distribution. Element-wise comparisons revealed that, when restricting to mixing among adults and seniors, the number of contacts engaged by adults and seniors (both within and across these two age groups) was similar in the two matrix types (Fig. 4a). In contrast, substantial discrepancies were observed for contacts involving younger individuals − especially in within-group contacts during school-open periods (Fig. 4b, Fig. 2b, c). Analysis of age assortativity showed that empirical matrices exhibited weaker within-group mixing compared to synthetic matrices, particularly for adolescents, adults, and seniors (Fig. 4c, d). In contrast, children had more assortative contacts in the empirical data (Fig. 4d). Moreover, in the empirical matrices, young individuals (i.e. children and adolescents) contributed to 50% of total connectivity, twice as in the synthetic contact matrices (Fig. S12b). Overall, these suggest that empirical matrices featured more intergenerational mixing and a stronger influence of younger age groups on overall contact patterns, compared to synthetic matrices. Cosine similarity−a metric defined between 0 and 1 assessing the similarity in mixing structure, independent of overall contact volume−

was higher during periods of school closure, indicating a larger agreement in age-specific mixing patterns between the two matrix sources (Fig. S10a). When schools were open, particularly in March 2021, similarity dropped, reflecting greater deviations in mixing among young age groups. Notably, cosine similarity between empirical and synthetic matrices was consistently higher than that between synthetic matrices and homogeneous mixing (Figure S10a), reinforcing that synthetic matrices retain meaningful age structure beyond proportional mixing.

We also compared synthetic and SocialCov matrices with empirical contact data from the CoMix study[42], conducted in France between December 2020 and April 2021. Due to its limited temporal coverage and inclusion of minors in only two waves (Fig. S12), CoMix was not used in the transmission model; however, its matrices showed lower contact rates than SocialCov and age assortativity similar to synthetic matrices (Figs. S13, S14).

**Transmission dynamics and model selection**

To assess the impact on transmission dynamics of the differences measured between the two types of contact matrices, we used an age-stratified compartmental model for COVID-19 (Fig. S1), fitting it to daily age-stratified hospital admissions and age-stratified seroprevalence data, using either weekly synthetic contact matrices obtained from mobility data or pandemic survey-based matrices (Fig. 5a). We also considered a model using the static pre-pandemic contact matrix for comparison. For the model using pandemic survey-based matrices, we extended the seven empirical contact matrices to periods beyond the survey waves based on assumed similarity of mixing conditions. For

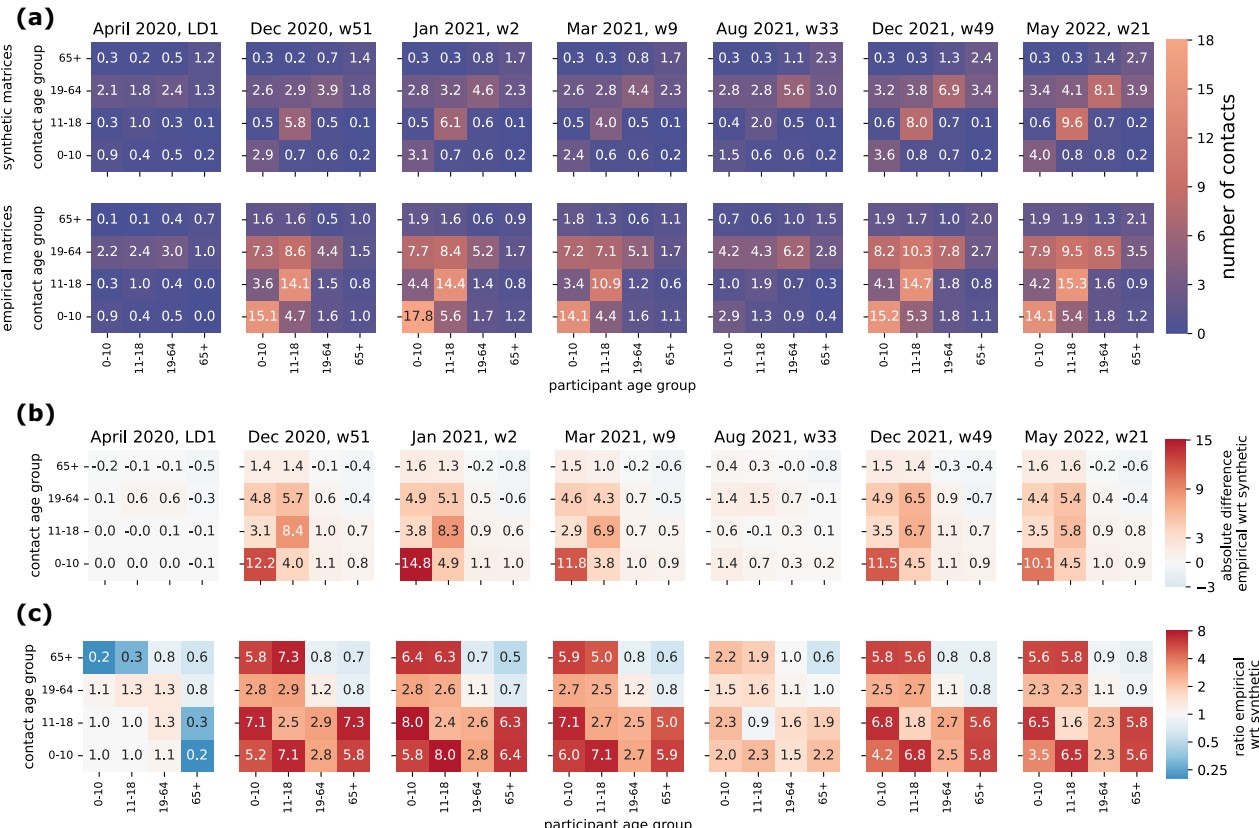

**Fig. 2 | Illustrations of mixing matrices. a** The matrices shown in the first and second row indicate the synthetic and empirical matrices, respectively. From left to right, they refer to seven periods, corresponding to periods of data collection in the SocialCov survey (April 2020, Dec 2020, Jan 2021, Mar 2021, Aug 2021, Dec 2021, May 2022; details in Table S7). In each matrix, the element $M_{ij}$ represents the average number of daily contacts that an individual in age group $i$ (x-axis) has with individuals in age group $j$ (y-axis), after correction for reciprocity. **b** Absolute differences in contact matrix elements. In each matrix, the element $D_{ij}$ represents the absolute difference $M_{ij}$ (empirical) $-M_{ij}$ (synthetic). **c** Element-by-element ratio. In each matrix, the element $R_{ij}$ represents the ratio $M_{ij}$ (empirical) / $M_{ij}$ (synthetic). The resulting matrix $R$ is symmetric because the synthetic and empirical matrices satisfy the condition of reciprocity.

example, we used the matrix estimated for August 2021 for all periods with school holidays, including periods prior to the data collection (i.e., this retrospective approach implies inputting matrices also before the actual survey date; see Methods and Fig. 5b). Our fitting approach estimated a baseline transmission rate per contact in the pre-lockdown phase and a time-varying correcting factor to adjust the transmission rate over time. This correcting factor acts as a global rescaling of the contact volumes of the matrices, accounting for latent drivers of transmission, without altering the internal structure of the contact matrices.

We evaluated each matrix type based on model fit quality. All models captured the observed trajectory of daily hospital admissions well (Fig. 5a, Fig. S3). This consistency reflects the use of short fitting windows (averaging three weeks), which allow flexible alignment with the hospitalization curve. However, the quality of fit varied across models. It was best in terms of both AIC (Table S9) and mean absolute error (Fig. 5c) for the model parameterized with synthetic contact matrices. Notably, this model captured smooth transitions in the epidemic trajectory during periods of gradual behavioral changes (e.g. curfews and school closures between the second wave and third wave), whereas the empirical and pre-pandemic matrix models showed abrupt and unrealistic shifts in epidemic trends (Figure S4).

The median correcting factor over time was estimated at 1.17 (0.59 [0.88, 1.34] 1.81, indicating Q2.5 [Q25, Q75] Q97.5 quantiles) for the model using synthetic matrices and 0.77 (0.45 [0.65, 1.07] 1.35) for the one using empirical matrices (Fig. 6a). In total, 66% and 53% of the correcting factors in the synthetic and empirical matrix models,

respectively, fell within the range 0.75–1.25, indicating that modest adjustments were sufficient in most weeks across both cases. The correcting factors showed temporal correlation across the models informed with synthetic and empirical matrices (Fig. S6). For the model using synthetic matrices, the correcting factor was close to 1 during the Alpha, Delta, and Omicron BA.1 waves, while for the model using empirical matrices, it was around 1 in 2020 and during the Omicron BA.2 wave (Figure S5). The synthetic matrices showed the largest deviations from 1 during summer 2020 (Fig. 6b). By contrast, the model using a static pre-pandemic contact matrix required consistently larger adjustments (median 0.5, quantiles 0.26 [0.43, 0.66] 0.87), reflecting the need to substantially scale down contact volumes to match observed dynamics during the pandemic (Fig. 6a).

We then compared the three models in terms of their ability to reproduce observed age-stratified data, specifically hospitalizations and seroprevalence. Age-specific hospitalization patterns were more accurately reproduced by the model parameterized with synthetic contact matrices, particularly for adolescents, adults and seniors. In contrast, the model using empirical matrices showed larger deviations in these age groups (Fig. 5d, Fig. S8). The model based on the pre-pandemic contact matrix underestimated hospitalizations among seniors (Fig. 5d, Fig. S8), likely reflecting outdated contact patterns not representative of pandemic conditions. For children, all models showed notable deviations between predicted and observed hospitalizations. Regarding serological data, estimates for adults and seniors were broadly consistent across all three models and aligned well with observations (Fig. 6e, f). In contrast, model predictions diverged more

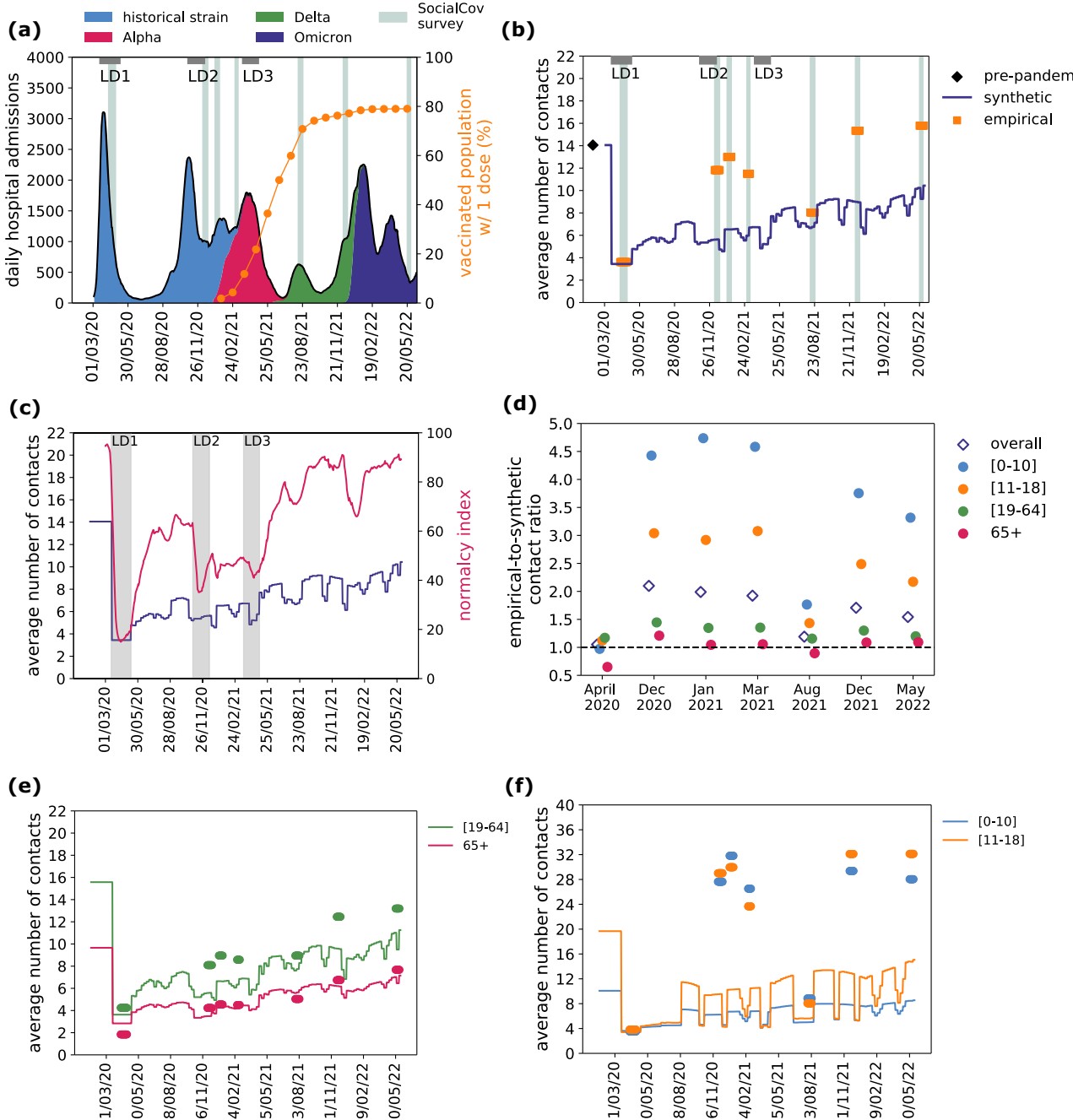

**Fig. 3 | Synthetic and empirical contacts over time. a** Timeline of the COVID-19 pandemic in France. Trajectory of daily hospital admissions (left y-axis) is shown in black, with colored areas indicating the proportion of SARS-CoV-2 circulating variants. Proportion of vaccinated population with 1 dose (right y-axis) is shown in orange. The gray horizontal bars annotated with LD indicate the periods of the three national lockdowns. Data collection periods for the empirical contacts (SocialCov surveys) are indicated with vertical shaded areas. **b** Average number of contacts over time, in the synthetic (dark blue line) and empirical contact matrices (orange). The value of the pre-pandemic empirical contact matrix used for baseline

is shown in black. **c** Average number of contacts over time in the synthetic matrix (dark blue line, left y-axis) shown in comparison with the Normalcy Index (pink line, right y-axis). Vertical gray bars indicate the periods of the three national lockdowns. **d** Ratio of the number of contacts estimated in the empirical contact matrices with respect to the synthetic matrices (empirical-to-synthetic contact ratio), broken down by survey wave (x-axis) and by age group (filled dots) or overall (void diamonds). **e**, **f** Average number of contacts over time by age groups (adults and seniors in (**e**), children and adolescents in **f**), in the synthetic (lines) and in the empirical contact matrices (dots), color-coded as in (**d**).

substantially for younger age groups. The model informed by empirical contact matrices consistently overestimated seroprevalence in children and adolescents after summer 2020. This overestimation matched the observed seropositivity in adolescents in June 2021, but substantially exceeded the values reported in February 2021, where the synthetic model provided a closer match (Fig. 6d). For children, neither model successfully reproduced the observed temporal trend in

2021: the model parameterized with empirical matrices overestimated seropositivity, while the model with synthetic matrices underestimated it (Fig. 6c).

We tested alternative calibration strategies, fitting the model using (i) age-stratified hospitalizations only or (ii) total hospitalizations, while reserving age-stratified serological and hospitalization data for validation (Figs. S17–S21). In both cases, model performance,

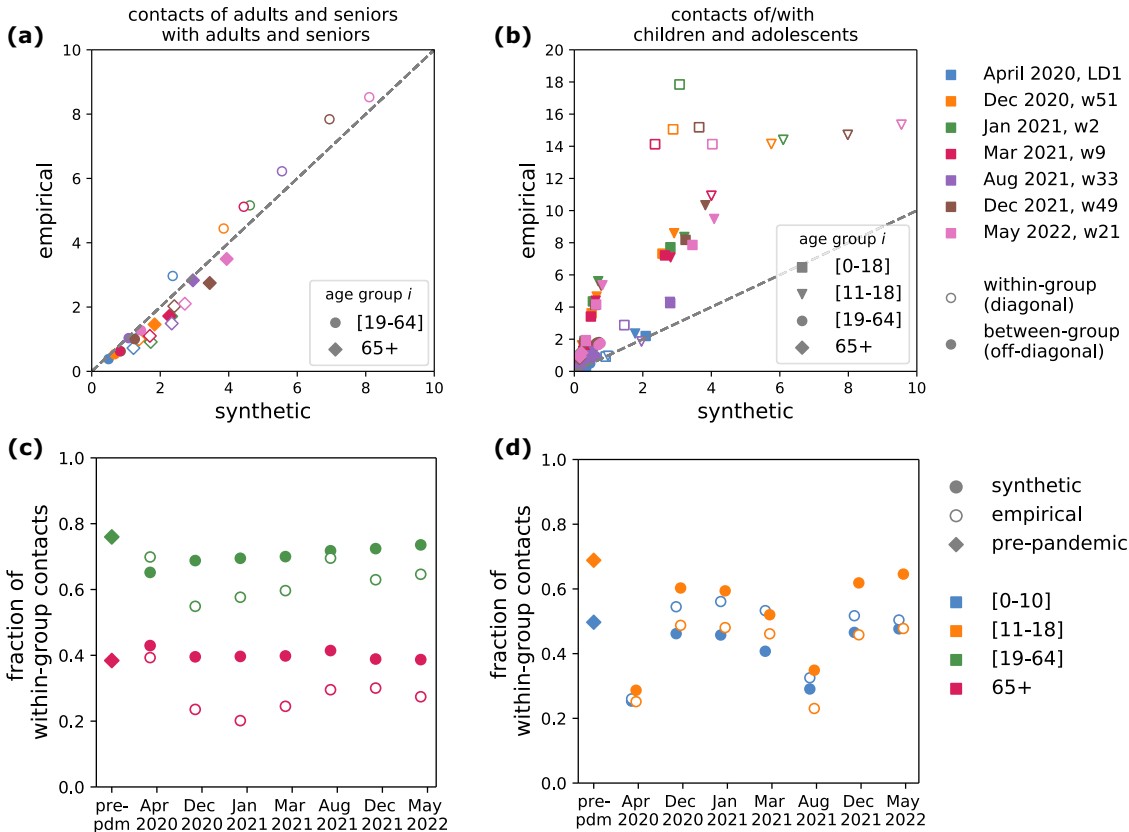

**Fig. 4 | Comparison of contact patterns. a** Contacts of adults and seniors with adults and seniors, i.e. matrix elements $M_{ij}$ with $i, j \in \{[19{-}64], 65+\}$, in the synthetic (x-axis) vs the empirical matrices (y-axis). Colors indicate the seven waves. Void and filled dots indicate within-group contacts (diagonal elements of the contact matrix) and between-group contacts (off-diagonal elements). **b** As in (**a**),

showing the rest of the elements of the matrix involving individuals in the younger age groups. **c, d** Fraction of within-group contacts for each age group (color-coded), defined as $M_{ii}/\sum_j M_{ij}$, in the synthetic (filled circles), empirical (void circles) and pre-pandemic matrix (diamonds).

the distribution of the correcting factor and the outcomes of model comparisons remained robust to the choice of calibration targets. The model informed by the synthetic contact matrices reproduced the main age-stratified hospitalization and serological patterns—though systematic deviations persisted for children—even when age-stratified hospitalization and serological data were not used for fitting.

To disentangle the relative role of the correcting factor and the contact matrices in capturing observed transmission dynamics and age-specific trends, we carried out a series of sensitivity tests using modified versions of the synthetic matrices. These tests systematically altered who is most at risk (by changing age-specific contact volumes while preserving total contacts; Figs. S22–S24) and from whom the risk comes (by modifying mixing patterns while preserving both total and age-specific contact volumes; Figs. S25–S27). Across all tests, the correcting factor remained comparable to that of the model parameterized with the original synthetic contact matrices. However, substantial differences emerged in age-stratified outcomes—both for hospitalizations and seroprevalence—highlighting that the correcting factor alone cannot compensate for structural changes in contact patterns. Notably, none of the models integrating the test matrices improved model fit compared to the original synthetic matrix model (Tables S14, S15).

Additional sensitivity analyses supported the robustness of our findings across alternative model specifications and assumptions. We first examined the role of physical contact avoidance in shaping the synthetic matrices (Section 4.2.4 in the Supplementary Information). In the main analysis, we reduced total contacts based on self-reported avoidance of physical contact, as a proxy for reduced transmission

risk. However, this assumes that avoided physical contacts were entirely suppressed, whereas some may have continued in a non-physical form (e.g., masked or distanced), likely leading to an over-estimation of the contact reduction. To test the impact of this conservative assumption, we generated a set of weekly matrices without applying this reduction. As expected, the average number of contacts across age groups increased (Fig. S28). Both the AIC and the mean absolute error indicated that this model performed better than the empirical matrix model, but still worse than the original synthetic matrix model incorporating contact avoidance (Table S16, Fig. S29). While this change improved the agreement with serological data for adolescents, it simultaneously worsened predictions of senior hospitalizations and children seroprevalence, underscoring age-specific trade-offs (Fig. S30). Varying the specification of empirical contact matrices – by incorporating weekend contact patterns or using May 2022 as a proxy for pre-pandemic behavior – did not improve model fit compared to the synthetic matrix model (Fig. S15, Table S11). Finally, assuming a lower relative susceptibility for children and adolescents (70% with respect to adults for all variants) did not improve alignment with serological data (Fig. S16, Table S12).

## Discussion

Modeling for real-time outbreak response requires continuously updated contacts that accurately capture shifts in behavior[19,43,44]. Traditional surveys, while highly valuable, are often costly, logistically challenging, and subject to delays that can limit their usefulness in dynamic contexts[45]. In response to these challenges, synthetic contact matrices based on mobility data were used for the first time during the

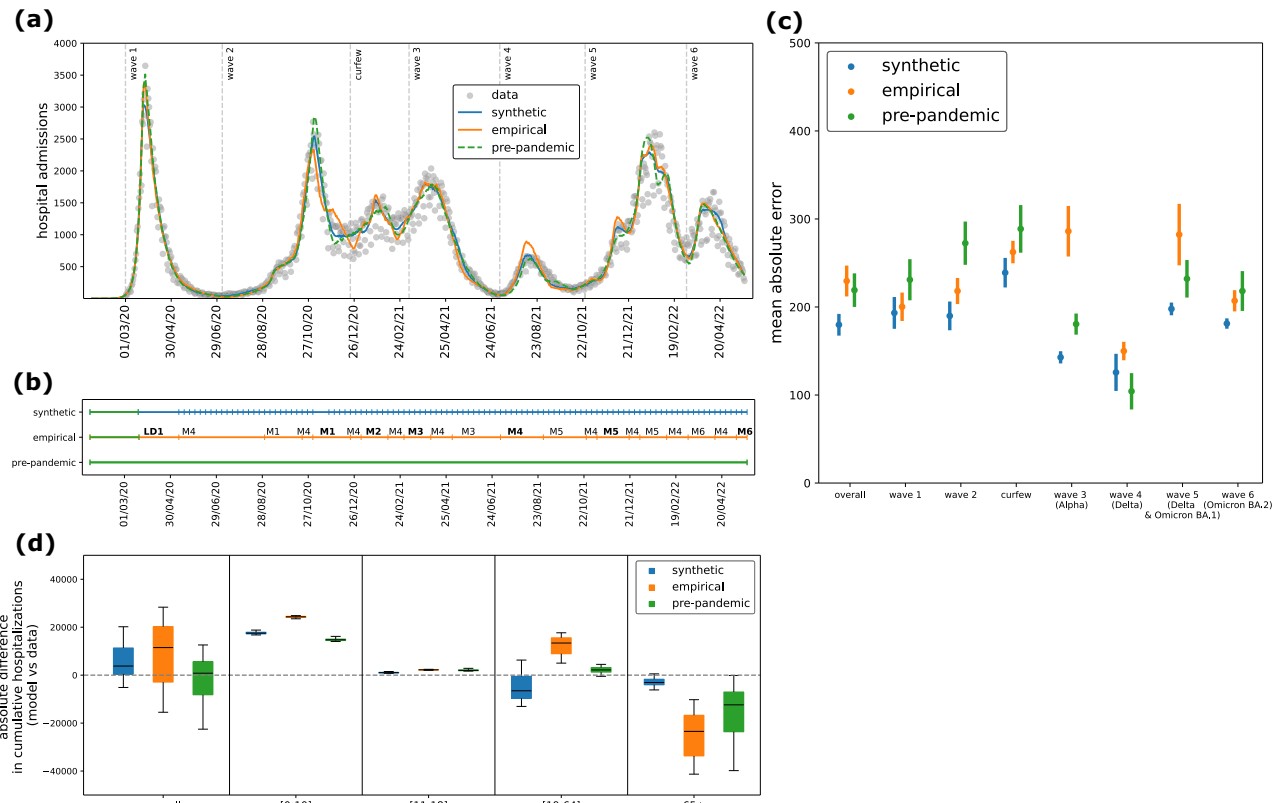

**Fig. 5 | Model fit. a** Model median trajectory of daily hospital admissions (median values out of 100 independent stochastic simulations), obtained by fitting the transmission model using weekly synthetic contact matrices (blue), extended empirical contact matrices (orange) and a constant pre-pandemic matrix (green). Data used for the fit are displayed with gray dots. Uncertainty around the median trajectory is displayed in Fig. S3. The vertical dashed lines define the epidemic phases used in (**c**). **b** Sequence of contact matrices used in the three transmission models. Each row corresponds to one of the three models, i.e. the model informed with synthetic mobility-based contact matrices (first row), with empirical contact matrices (second row), or with a static pre-pandemic contact matrix (third row); each tick indicates a change in the contact matrix used in the model. Colors indicate the source of the matrix, i.e. pre-pandemic (green), synthetic (blue) and pandemic empirical (orange). The empirical contact matrices estimated from the 7 pandemic survey waves are denoted with LD1, M1, M2 up to M6, following the notation in

Methods and Table S7. They have been extended beyond the survey period to cover the whole study period. They are highlighted in bold in the periods that overlap the survey wave. **c** Mean absolute error (MAE) of daily model predictions with respect to the daily observed data, on the overall period (March 2020 – May 2022) and broken down by epidemic phase (epidemic waves and in-between periods). Dots and lines represent the average MAE and 95% confidence interval computed across n = 100 independent stochastic runs. For each stochastic run, we computed the MAE as the sum of absolute differences between daily model predictions and the observed daily data, divided by the number of days in the epidemic phase under consideration. **d** Absolute difference (overall and by age class) of the cumulative number of hospital admissions (from March 1, 2020 to May 22, 2022) predicted by the three models with respect to observations. The box plot indicates median (line), interquartile range (box), and 2.5% and 97.5% quantiles (whiskers) out of n = 100 independent stochastic runs.

COVID-19 pandemic to enable real-time modeling of transmission dynamics[18,36–40,46,47]. This study systematically compared such synthetic matrices with empirical contact data collected during multiple phases of the pandemic in France[29,30]. While both sources exhibited similar overall temporal trends, notable differences emerged, particularly in children contacts and mixing patterns. Models parameterized with synthetic matrices better reproduced observed total and age-specific hospitalizations for adolescents, adults, and seniors, than those based on empirical or pre-pandemic contact patterns. However, for children, deviations from observed serological trends remained across all models. These findings support the value of mobility-informed synthetic matrices as a flexible, operationally viable tool for epidemic modeling, especially when real-time or age-stratified empirical contact data are sparse.

All models well captured the overall trajectory of total hospitalizations, and results remained robust across all calibration targets tested for sensitivity. This consistency comes from the inference framework, which relies on short, discrete time windows for estimating the correcting factor, limiting the accumulation of fitting errors and allowing sufficient flexibility to align model outputs with observed data. Despite this flexibility, the model informed by weekly synthetic

contact matrices outperformed those using empirical or pre-pandemic matrices, achieving better fit metrics (AIC and mean absolute error). Its performance was particularly strong during transitional phases of the epidemic, such as periods marked by school closures, curfews, or gradual easing of restrictions, where it captured smoother changes in hospitalization trends. In contrast, models using empirical or pre-pandemic contact patterns produced unrealistic discontinuities during these phases. The higher performance of the synthetic matrices in these contexts originates from their key advantage, i.e., the ability to be updated weekly. This update is essential for real-time modeling applications, supporting improved situational awareness, scenario planning, and timely policy response. In contrast, empirical matrices generally have limited temporal resolution, therefore requiring strong assumptions to fill gaps between infrequent survey waves. We tested various specifications of the empirical matrices, and the model parameterized with mobility-based synthetic matrices consistently outperformed them. While more frequent surveys could improve the empirical matrix model, they come with substantial resource demands and require automated tools and streamlined methods to transform survey data into actionable insights efficiently[23,48]. Alternative methods for temporally extending empirical matrices may hold promise, but

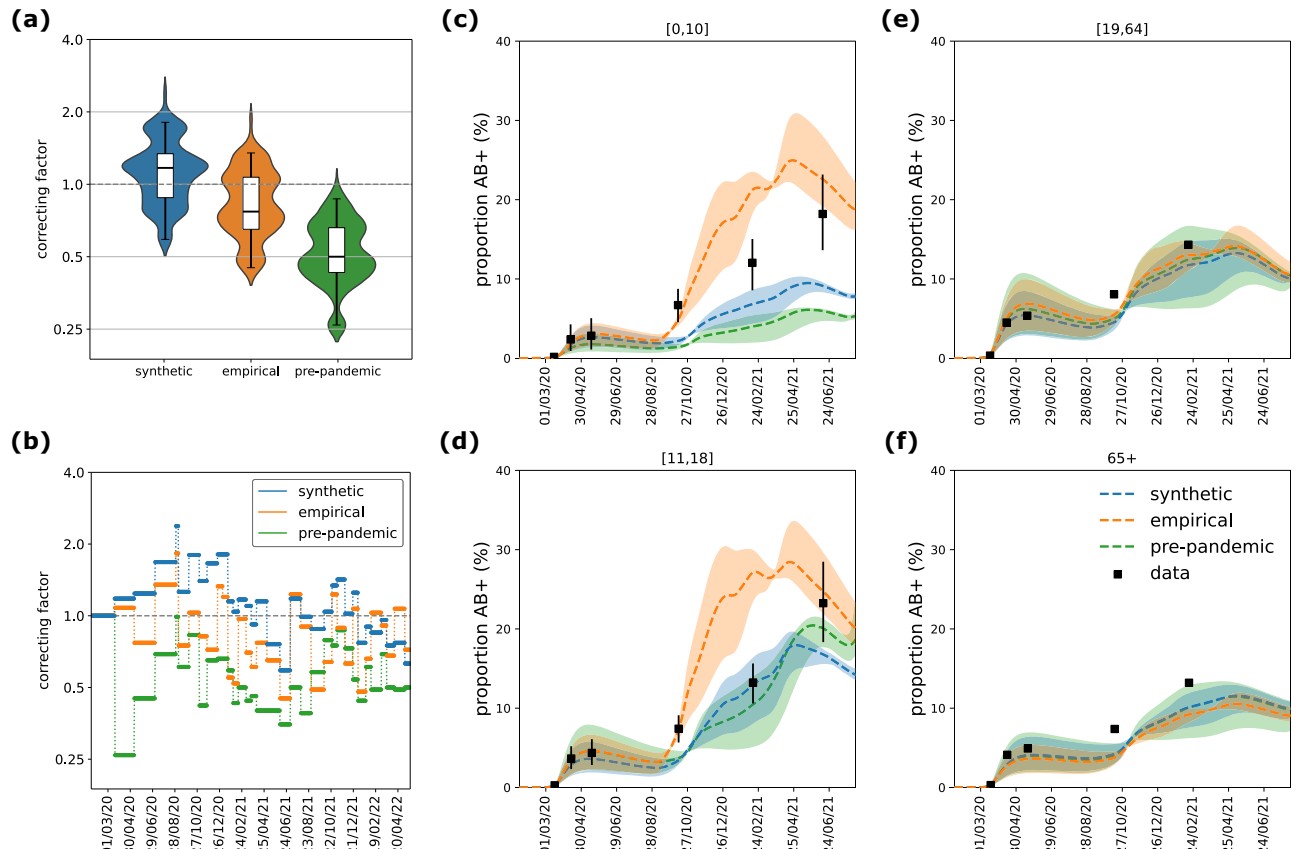

**Fig. 6 | Model comparison. a** Distribution of the correcting factor, fitted with the model using synthetic matrices (blue), empirical matrices (orange), and a constant pre-pandemic matrix (green). The box plot indicates median (line), interquartile range (box), and 2.5% and 97.5% quantiles (whiskers) of the values of the correcting factor from the start of the first lockdown to the end of the study period (March 2020 – May 2022, n = 794 days). The distribution by pandemic phase is shown in Fig. S5. **b** Correcting factor over time, fitted with the model using synthetic matrices, empirical matrices, and a constant pre-pandemic matrix. **c–f** Proportion of antibody-positive population over time, estimated with the three models, by age class (children [0,10] in (**c**), adolescents [11–18] in (**d**), adults [19–64] in (**e**), seniors 65+ in (**f**)). Dashed lines and shaded areas indicate respectively the median and 95% probability ranges (2.5% and 97.5% quantiles) computed across n = 100 independent stochastic simulations. Black symbols indicate estimates from serological data[78].

they are often context-specific and remain an open area for future research.

The correcting factor estimated in our model acts primarily as a global rescaling of contact volumes, enabling alignment between modeled and observed transmission without altering the internal structure of the contact matrices. It pragmatically absorbs residual variation in transmission intensity not captured by available behavioral or epidemiological inputs. Both synthetic and empirical matrices required moderate adjustments for most of the study period to reproduce observed hospitalizations. The largest deviation from 1 of the correcting factor occurred for the synthetic matrices during summer 2020—the first summer of the COVID crisis—when contacts were highly impacted and not easily reflected by workplace presence or school calendars due to the summer holidays. While empirical matrices performed slightly better during this period, they were constructed retrospectively using data collected in summer 2021, illustrating the limitations of infrequent survey-based approaches for real-time use. By contrast, the model using the static pre-pandemic matrix required consistently large corrections to modulate contact volumes over time. Interestingly, the correcting factor showed some temporal correlation across models, suggesting the presence of unmodeled factors—such as changes in testing, masking, or contact duration—that affect transmission potential independently of contact frequency. Crucially, our tests showed that altering the structure of the contact matrices had substantial effects on age-specific outcomes even when the correcting factor remained stable, underscoring that this factor

alone cannot compensate for mis-specification of mixing patterns. While it provides flexibility for the fit, it does not mask structural limitations in the underlying contact data. These results therefore also underscore the central role of matrix structure in shaping model projections. Importantly, model comparisons and correcting-factor estimates remained consistent when fitting to different calibration targets—including total hospitalizations, age-stratified hospitalizations, or both age-stratified hospitalization and serological data—indicating that our conclusions do not depend on the specific choice of calibration inputs. Notably, incorporating serological data did not materially alter the performance of the synthetic matrices, suggesting that these matrices remain informative for real-time use even in settings where serological surveillance is absent. Future research could investigate the feasibility of fitting age-specific correcting factors that change age-specific total contacts while maintaining mixing preferences; although feasible in principle, achieving parameter identifiability would likely require advanced iterative sampling methods[49].

Differences between synthetic and empirical contact matrices were most pronounced in children and adolescents. Empirical matrices consistently reported higher contact rates for these age groups, particularly during school-open periods and in within-group contacts, contributing to an overestimation of seroprevalence in both children and adolescents. In contrast, the synthetic matrices underestimated seroprevalence levels in children but provided a closer fit to adolescent serological data during certain periods. These discrepancies reflect the challenge of accurately capturing school-related and

intergenerational mixing dynamics. They could be explained by a set of factors. Synthetic matrices applied uniform reductions in physical contacts across all age groups using data from the CoviPrev survey[41], which only included adults. This may have underestimated school contacts, as children likely had fewer opportunity to avoid physical interactions in structured environments like classrooms. Indeed, contact estimates from the two matrix types reconcile during periods of school closure. However, increasing child contact numbers in the synthetic matrices—by assuming that avoided physical contacts were replaced with non-physical ones rather than removed—did not improve model fit of seroprevalence in children. Conversely, empirical matrices may have overestimated these contacts. First, safety measures like mask-wearing in schools and staggered schedules likely mitigated the risk of transmission, meaning that the same number of reported contacts would not translate to the same transmission. Second, SocialCov surveys relied on aggregated contact reporting by age group rather than individual contact listings, which may have facilitated reporting – especially for parents reporting for their children – at the cost of accuracy. Previous work has shown that aggregated formats tend to report higher contact numbers compared to detailed individual listings[27]. This is further supported by our comparison with CoMix matrices, which reported substantially fewer contacts for children compared to SocialCov, while their assortativity was more similar to that of the synthetic matrices. Overall, the smaller number of contacts generated synthetically in the younger age groups better captured the disease dynamics of hospitalizations when integrated into the transmission model.

Both the synthetic and the empirical model accurately captured hospitalization and serological rates among adults and seniors, reflecting consistent contact patterns in those age groups across matrix types. The synthetic matrices incorporated reductions in workplace attendance derived from mobility data, under the assumption that fewer people at work equated to fewer contacts[36,46]. This operational assumption – approximating between density- and frequency-dependent transmission – proved effective, as the synthetic matrix model aligned closely with age-stratified hospitalization and serological data for adults and seniors. These findings support the use of mobility data as a valuable proxy for time-varying contacts in workplace settings during a pandemic. While earlier studies showed strong correlation between mobility and COVID-19 spread in the early pandemic phase[50–52], this relationship weakened over time as behavioral adaptations and masking became more prominent[53–56], underscoring the limitations of using inappropriate proxies or relying on simplistic linear associations between mobility and transmission[54]. Still, our findings demonstrate that workplace attendance data, when carefully selected as a setting-specific proxy and integrated nonlinearly into a synthetic framework, remain a meaningful and robust input for capturing transmission dynamics.

Our study has a set of limitations. First, the mobility and behavioral data used to construct synthetic matrices (e.g., workplace attendance, preventive behavior) were not age-stratified, but some of these proxies indirectly introduced age specificity. However, no finer age breakdown was available. Relatedly, while additional age classes with the same assumptions were possible, we limited our synthetic matrices to four broad age classes; future work could assess a better trade-off between detail and real-time feasibility. Generating age-detailed synthetic matrices in real time remains constrained in the absence of age-stratified mobility[57] and behavioral data, which would be essential to refine assumptions and capture more accurate age-specific changes in contact patterns over time. Second, our model assumed distinct daily contacts, thus ignoring the repetition of contacts and potentially underestimating transmission risks[58]. This limitation is inherent to the chosen modeling framework and could only be addressed by adopting an agent-based framework. Relatedly, we used average contact rates and did not account for individual

heterogeneity in contacts or superspreading events[59–61]. Moreover, lacking data to support differential transmission risks, our model assumes that all contacts contribute equally to transmission, regardless of their nature (physical or non-physical) or setting. While we tested the impact of excluding self-reported physical contact avoidance from the synthetic matrices, this resulted in a worse fit, improving agreement in some age-specific outcomes while worsening in others, further highlighting the complexity of translating behavioral proxies into transmission-relevant inputs. Third, we considered age-specific susceptibility for the original strain[62,63], and assumed homogeneous susceptibility across age groups for the variants due to limited evidence and following other works[22,64]. However, sensitivity analyses considering age-specific susceptibility also for variants showed that our best model, informed by synthetic contact matrices, continued to outperform other models in terms of AIC and error. Fourth, due to a lack of age-specific estimates, we assumed that the average time to seroreversion for young individuals was the same as that for adults. This assumption may limit the accuracy of comparisons between model predictions and serological data for children and adolescents. Finally, we used a piecewise constant function for the correcting factor. The abrupt changes in epidemic trends produced by the empirical and pre-pandemic matrix models could, in principle, be mitigated by adopting a continuous functional form for the correcting factor, such as a spline. Nevertheless, our findings show that the synthetic model, despite relying on a piecewise definition of the correcting factor, successfully captured smooth transitions, indicating that the observed discontinuities arise from limitations in the empirical and pre-pandemic matrices themselves.

By systematically comparing synthetic and empirical contact matrices, we demonstrated that mobility-informed synthetic matrices offer a reliable, timely, and operationally scalable alternative for modeling transmission dynamics, particularly when survey-based contact data are limited or delayed. While empirical matrices can provide greater granularity in contact reporting, their infrequent collection and retrospective application may hinder real-time responsiveness. Synthetic matrices, by integrating mobility and behavioral data, captured key epidemic features and age-specific trends with higher fidelity, supporting their use in public health decision-making. These findings advocate for greater integration of non-traditional digital data sources, such as mobility, into epidemiological modeling frameworks. These data streams are more flexible, scalable, and cost-effective than empirical surveys, making them valuable tools for real-time outbreak monitoring and response. As the world prepares for future pandemics, our study underscores the importance of leveraging real-time data to inform public health interventions and improve crisis management.

## Methods

### Pre-pandemic baseline contact matrix
We used pre-pandemic contact data collected from a large-scale survey in France in 2012[12], distinguishing between contacts engaged during regular weekdays, weekends or school holidays. We derived a social contact matrix corrected by reciprocity, and broken down by location (home, school, work, transport, leisure, other) and type of contact (skin-to-skin or non-physical contact at short distance), using the Social Contact Rates (SOCRATES) Data Tool[48] (Fig. 1a,b). The original survey collected Supplementary Professional Contacts (SPC), i.e. participants with more than 20 daily professional contacts were asked not to report them but rather to provide their total number and age distribution. We included SPC only in the elements of the work matrix involving adults and seniors. This baseline matrix was then adapted to the French population in 2020 using demography data, applying an appropriate density correction (following ref. 15). Through the density correction, the original matrix $M_{ij}$ (whose elements represent the average number of contacts an individual in age group $i$ establishes

with individuals in age group $j$) is projected to the demographic structure of 2020 by defining a new matrix $M'_{ij} = M_{ij} * (N/N_j) * (N'_j/N')$, where $N$, $N_j$, refer to the total population and the population in age group $j$, respectively, in the year of the survey, while $N'$, $N'_j$ refer to the population in 2020. This density correction preserves reciprocity, so that the projected matrix fulfills the condition $M'_{ij}N_i = M'_{ji}N_j$. We considered the contact matrix estimated for a regular weekday, defined as Monday-Friday excluding holidays. We did not model explicitly the weekday/weekend effect, which is absorbed in the calibration procedure of the transmission model (see section Inference Framework below). However, we accounted for the impact of school holidays. We used the pre-pandemic contact data collected during spring school holidays to model the synthetic pandemic contact matrix during spring, winter and Christmas holidays. We made some assumptions when modeling summer holidays in the lack of available pre-pandemic contact data. See the section below and the Supplementary Information (Section 4) for further details.

## Construction of synthetic contact matrices

We built time-varying synthetic contact matrices on a weekly basis (except for lockdown periods) from March 2020 to May 2022. We considered four age groups: children [0–10], adolescents [11–18], adults [19–64] and seniors (65+ years old). We chose a limited number of groups to parametrize the matrices using the available behavioral data (missing finer age-stratification) with minimal assumptions, while capturing epidemiologically relevant groups and ensuring interpretability and real-time feasibility of the framework. Matrices were obtained by applying reductions to the components of the pre-pandemic contact matrix (location and type of contact) to simulate the social mixing conditions experienced during the pandemic. In particular, we parametrized the matrices to account for (i) impact of adoption of remote working in reducing contacts at work and on transports, (ii) impact of full or partial school closure (or remote learning) and impact of school holidays on school-related contacts, (iii) impact of social-distancing measures on contacts associated to non-essential activities, (iv) impact of adoption of health preventive behaviors, such as avoiding physical contacts. We used proxy contact data collected during the pandemic to inform the parameterization of the synthetic contact matrices. In particular, we used Google mobility data[31] related to workplaces to adjust the number of contacts in the work matrix over time (Fig. 1c). Google measures the change in the number of visitors to a specific location with respect to a pre-pandemic baseline; the mobility change related to workplaces can thus be interpreted as an effective reduction in attendance at work. We assumed that such reduction in attendance produces a non-linear reduction of contact rates (Section 4.2.1 in the Supplementary Information) that is in between the frequency-dependent and the density-dependent assumption[65]. We also used data on voluntary school attendance in the exit phase of the first lockdown, and the calendar of school holidays to adjust school-related contacts (Table S5). Finally, we used data from the CoviPrev survey[41] on declared avoidance of physical contacts during the pandemic to reduce the proportion of skin-to-skin contacts (Fig. 1d). We assumed that physical contacts were removed rather than replaced with non-physical contacts, thus representing a lower-bound scenario; this assumption was tested in a sensitivity analysis (see section Sensitivity Analyses below). The framework for building the synthetic contact matrices is fully detailed in Section 4.2 of the Supplementary Information, and summarized in Table S6. The resulting average number of contacts over time was tested for correlation with the Normalcy Index and the Stringency Index (Fig. 3c and Fig. S13). The Normalcy Index[66] is a measure of the impact of the pandemic on human behavior, integrating multiple daily indicators of human activities in a score from 0 to 100, with 100 representing the pre-pandemic level. It is important to note that the Normalcy Index and the indicators used for constructing synthetic matrices are not the same: the Normalcy Index aggregates eight broad measures of societal activity (e.g. retail, transport, leisure, office occupancy), while our synthetic matrices mainly relied on workplace mobility and physical contact avoidance. The latter indicator has no equivalent in the Normalcy Index. The Stringency Index[67] is a composite measure of nine response metrics (e.g. as school closure, restrictions on public gatherings) to quantify the strictness of government policies for epidemic control, in a scale from 0 to 100, with 0 indicating absence of measures.

## SocialCov contact surveys

Seven survey waves were conducted in France to collect data on contact behavior[29,30]. Survey participants were recruited online and contact matrices were adjusted to the French population for representativeness. SocialCov recruited participants through convenience sampling via the governmental app TousAntiCovid. In particular, the survey was promoted via the news channel of the app which invited individuals aged 18 and above to complete the questionnaire. If participants declared living with minors (i.e. individuals under the age of 18), they were invited to complete a second specific questionnaire for one of them, in his/her presence. Since the survey sample was not representative of the French population, synthetic populations were generated for each campaign to better reflect the age and gender distribution in France, using sampling with replacement from the SocialCov participant pool; the matrices were bootstrapped to account for sampling variance[30]. The first survey was conducted during the first lockdown (matrix LD1)[29]. The survey was then implemented in 6 additional waves[30], in December 9–22, 2020 (M1), January 10–21, 2021 (M2), March 2–10, 2021 (M3), August 12–24, 2021 (M4), December 6–17, 2021 (M5), and May 20–29, 2022 (M6). Contacts were defined as either a physical contact (such as a kiss or a handshake) or a close contact (such as face-to-face conversation at less than 1 m distance). Total contacts (physical and non-physical) for each individual were truncated at 50 to reduce the impact of outliers. We used contacts reported on Monday-Friday excluding holidays, to allow comparison with the synthetic contact matrices that were built based on a regular weekday pre-pandemic matrix. In a sensitivity analysis, we used matrices weighted by weekday and weekend. Participants in the survey reported contacts aggregated by age group, and were used to produce age-stratified 10×10 contact matrices. We aligned the age groups of the contact survey with the four age groups used for the synthetic matrices to allow comparison (see Section 4.3 of the Supplementary Information).

## Comparison of contact patterns

From the set of weekly synthetic contact matrices, we extracted the ones with the closest matching period to the seven survey waves and made direct comparison between the two sets of matrices (Table S7). We summarized the information contained in each contact matrix through different metrics in order to make the comparisons. We computed (i) the average number of contacts, overall and by age class, (ii) a measure of matrix correlation based on the cosine similarity, (iii) the proportion of average connectivity due to young individuals (<19 y.o., i.e., including children and adolescents), (iv) an index for age-assortativity and (v) the age-specific proportion of within-group contacts. All these quantities are mathematically defined in the Supplementary Information (Section 5). Indicators (iv) and (v) are different measures of the degree assortativity, quantifying the extent to which contacts occur between individuals who share their same characteristics (in our case, the relevant characteristic is age). Contact matrices estimated from empirical data usually show some assortativity with age, i.e. they have a strong diagonal component, as individuals tend to mix with individuals with similar age[9]. More assortative contacts have implications on disease spread, increasing the epidemic risk and leading to lower herd immunity thresholds[68,69].

We further compared the synthetic and SocialCov matrices with two empirical matrices from the CoMix study (Section 8.3 in the Supplementary Information), which was conducted in France between December 2020 and April 2021. CoMix provided complementary age-stratified data, but its limited temporal coverage and inclusion of minors in only two of seven waves restricted this comparison to two periods.

## Transmission model

We integrated the two sets of contact matrices (synthetic and empirical) into an age-stratified transmission model to simulate the unfolding of the COVID-19 pandemic in France from early 2020 to May 2022. The model has been presented in depth in previous works[18,36,37]. We used a stochastic age-structured two-strain transmission model with vaccination, parameterized using French data on demography[70], age profile[70], and vaccine uptake[71]. Transmission dynamics follows a compartmental scheme illustrated in Figure S1, which accounts for latency period, pre-symptomatic transmission, asymptomatic and symptomatic infections with different degrees of severity, and individuals affected by severe symptoms requiring hospitalization. Contact rates for each disease stage are adjusted to model the effect of spontaneous change of behavior due to severe illness and the impact of testing and self-isolation (Section 4.4 of Supplementary Information). We did not differentiate transmissibility by contact type or setting, assuming all contacts contribute equally to transmission. The model reproduces the co-circulation of two strains, and was applied to describe the Wuhan-Alpha period (February 2020 – May 2021), the Alpha-Delta period (June 2021 – August 2021), and the Delta-Omicron period (September 2021 – May 2022). Epidemiological parameter values and sources for the Wuhan strain are reported in Table S1. The model is parametrized with age-dependent susceptibility and disease severity. For the Wuhan strain, children and adolescents have a relative susceptibility of 70% with respect to adults. Variant-dependent parameters include the generation time, the transmission advantage, the infection-hospitalization ratio (Table S2). The model is further stratified by (i) vaccine dose, to build vaccine coverage in the population over time according to data on vaccine doses administered in France[71], and (ii) time since vaccination, to model steps of waning in vaccine effectiveness (Fig. S2). The model accounts for possible re-infection with Omicron after a prior infection, with waning in protection against re-infection. We distinguished between different levels of protection conferred by vaccine-only, natural (infection-only) and hybrid immunity (Section 3 of the Supplementary Information). In parallel to disease stage progression, we also modeled seropositivity to compare model results with seroprevalence data from Sante publique France measuring the presence of IgG-type antibodies. Following the modeling approach adopted in ref. 21, we assumed that upon becoming infectious (i.e. while exiting the $E$ compartment in Fig. S1), infected individuals also enter in a compartment $AB_{pre}$ (in parallel to $I_p$) which represents the pre-seropositivity compartment. Then, individuals move to the seropositivity compartment $AB^+$ after seroconversion, and finally they move to the seronegativity compartment $AB^-$ after seroreversion. We informed the average time spent in $AB_{pre}$ and $AB^+$ based on estimates from the literature for IgG-type antibodies. We used 12 days for seroconversion[72] for all age groups, and 200 days for seroreversion for adults[73–76]. Given the evidence of slower seroreversion for more severe infections, we used 400 days for seniors[77]. In the lack of estimates specific for children and adolescents, we assumed the same value as adults. We then compared the proportion of $AB^+$ over time predicted by the model with French national seroprevalence estimates[78] collected by Sante publique France, available by age group, and corrected by test sensitivity. Since the IgG test does not distinguish between antibodies from infection and vaccination, and no information on vaccination status was collected from serologically

tested individuals, we restricted our analysis to seroprevalence data collected before the start of the vaccination campaign. This ensures comparability with model estimates of antibody-positive individuals, which reflect infection-derived immunity only.

## Inference framework

The model was fitted to daily age-stratified hospital admission data[79] and age-stratified seroprevalence data[78] since the start of the pandemic (February 2020) up to May 22, 2022. We used a maximum likelihood approach to fit a step-wise transmission rate per contact (see Section 6 of the Supplementary Information for full details). More specifically, in the pre-lockdown phase (February – March 2020), we fitted the starting date of the epidemic and the baseline transmission per contact $\beta_{pre-LD}$. Then, we fitted a correcting factor $\alpha_{phase}$ of the transmission rate, calibrated sequentially in successive time-windows (Table S8), each one representing a different pandemic phase, based on epidemic activity, behavior and interventions implemented (e.g., pre-lockdown, lockdown, exit phase, summer, curfew). As changes in the variant's transmissibility are explicitly modeled through the transmission advantage, the parameter $\alpha_{phase}$ is meant to absorb other factors potentially affecting the transmission that are not captured by the time-varying contact matrices, e.g., mask usage or outdoor/indoor activity, or mis-specification of contacts. This correcting factor can be thus interpreted as a global correction of the contact matrices used in the model. The product of the fitted $\alpha_{phase}$ at time $t$ with the corresponding contact matrix represents the effective contacts needed to reproduce the observed epidemic dynamic. The closer the correcting factor to 1, the better the alignment between the time-varying contact matrices and the effective contact rates needed to reproduce the total epidemic dynamic. However, acting as a global rescaling of the contact matrix, the correcting factor cannot alter the underlying matrix structure and mixing, responsible for the age-stratified model outcomes.

## Model comparison

We compared the outcomes of three transmission models. The first model is informed with weekly mobility-based synthetic contact matrices, as done in our previous works during the pandemic[18,36–40]. The second model is informed with empirical contact matrices estimated from survey data (SocialCov[29,30]) collected during the pandemic. The contact matrices available from 7 distinct surveys were extended beyond the survey period in order to cover the time span of the model, as pictured in Fig. 5b. We used the matrix estimated for summer 2021 as a proxy for school holidays, as this was the only survey wave fully occurring in a period with schools closed (excluding the lockdown). Both models used the empirical pre-pandemic contact matrix in the period prior to the implementation of the lockdown. The two models with time-varying contact matrices (either synthetic or empirical) were compared to a third reference model integrating the same static pre-pandemic contact matrix throughout the whole pandemic period. To identify the model which better reproduced the change of behavior and the resulting epidemic dynamic, we compared the model outcomes in terms of (i) Akaike Information Criterion (AIC) as a measure of goodness of fit, (ii) mean absolute error of the model trajectory of daily hospital admissions with respect to the data, (iii) distribution of the correcting factor, (iv) age-specific model estimates of the proportion of antibody-positive individuals over time (accounting for seroconversion and seroreversion) in comparison with seroprevalence data, and (v) age-specific model estimates of hospital admissions compared to age-stratified data. While the correcting factor provides indication about the adjustment required to reproduce the total epidemic, the comparison of age-stratified model outcomes with observations is an indicator of the ability of the matrices to capture contact structure and mixing patterns.

## Sensitivity analyses

We carried out sensitivity analyses on (i) the susceptibility of young individuals with SARS-CoV-2 variants; (ii) the integration of the empirical contact matrices into the transmission model, including the effect of weekend or using the empirical matrix of May 2022 as a pre-pandemic matrix; (iii) alternative specifications of the likelihood using other calibration targets, i.e., age-stratified hospitalizations only (without accounting for seroprevalence data) and total hospitalizations (without accounting for age-stratified data). To demonstrate that matrix structure (encoding which group is most at-risk for infection and from whom) plays a critical role in shaping epidemic outcomes, we also carried out multiple tests by systematically varying the structure of the synthetic matrices, introducing modifications in (iv) age-specific contact volumes or (v) mixing patterns. Finally, we tested an alternative parameterization of the synthetic matrices without assuming contact removal due to avoidance of physical contacts. All sensitivity analyses are reported in Section 9 of the Supplementary Information.

## Reporting summary

Further information on research design is available in the Nature Portfolio Reporting Summary linked to this article.

## Data availability

Mobility-driven synthetic contact matrices and SocialCov contact matrices are available at https://github.com/EPIcx-lab/COVID-19/tree/master/mobility_driven_synthetic_contact_matrices. All other data used in the analyses are available at the references cited in the Methods: pre-pandemic contact data[12], Google mobility data[31], CoviPrev behavioral survey data[41], Normalcy Index[66], Stringency Index[67], French population data[70], vaccine uptake[71], hospital admission data[79], seroprevalence data[78].

## Code availability

Code of the transmission model is publicly available at https://github.com/EPIcx-lab/COVID-19/tree/master/mobility_driven_synthetic_contact_matrices.

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

## Acknowledgements

This study was partially funded by: ANR grant DATAREDUX (ANR-19-CE46-0008-03) to V.C.; EU Horizon 2020 grant MOOD (H2020-874850) to V.C., L.D.D.; Horizon Europe grants VERDI (101045989) and ESCAPE (101095619) to V.C.

## Author contributions

V.C. and L.D.D. conceived and designed the study. V.C., L.D.D., and C.E.S. developed the framework for the synthetic contact matrices. L.D.D. and C.E.S analyzed the data to build the synthetic contact matrices. P.B. and L.O. collected and analyzed the data from the SocialCov survey. L.D.D. developed the code for the comparison. L.D.D. and C.E.S. performed the numerical simulations. L.D.D. analyzed the results. L.D.D., P.B., L.O., and V.C interpreted the results. L.D.D. drafted the article. All authors contributed to and approved the final version of the Article.

## Competing interests

The authors declare no competing interests.
