## [Transparent Peer Review file · Nature Communications]

Mobility-driven synthetic contact matrices as a scalable solution for real-time pandemic response modeling

Corresponding Author: Dr Vittoria Colizza

Version 0:

Reviewer comments:

Reviewer #1

(Remarks to the Author)

Summary

The authors present a comparison of two methods to generate dynamic contact matrices among 4 age groups for France between 2020 March and 2022 May: 1) synthetic matrices made by adjusting pre-pandemic contact matrices using mobility data and assumptions; and 2) empirical matrices, estimated from 7 serial surveys in France over a similar period, adjusted for non-representative sampling. As a control case 3), a static pre-pandemic matrix was also compared. The aims explored are interesting and important: i) how do these matrices compare to each other, and ii) which method performs better for transmission modelling (of covid-19), especially given that (1) can likely provide more timely and inexpensive estimates than (2).

The model design for aim (ii) includes numerous important features for covid-19 dynamics, such as age-stratified susceptibility & hospitalization, different viral strains, empiric vaccine uptake over time, natural/vaccine/hybrid & waning immunity, which all appear to be diligently parameterized. The discussion is also overall excellent, except some overstated conclusions. However, the inference framework for aim (ii) does not allow for the necessary evidence to address aim (ii) and aim (i) remains insufficiently explored. My overall recommendation is to split these aims into two papers to address each more thoroughly. Do to the above, I do not find the current results particularly useful.

Major Issues

+ Model-based inference framework for aim (ii): Conclusions such as "Our study demonstrates that mobility-based synthetic matrices performed well in capturing the dynamic changes in contact behavior across age groups throughout the pandemic" are not well supported by the results, and are in some ways unfalsifiable in the current analysis framework, due to the unconstrained correcting factor alpha and lack of age-specific calibration targets. In other words, what results would cause the authors to draw a different conclusion?

- Calibration targets: The only data used to fit the model were total hospitalizations per week. However, age-stratified hospitalization data appeared to be available (Suppl 7.3 & Figure S5), as well as age-stratified seropositivity prevalence over time (Figures 5 & S9). It's not clear why these age-stratified data weren't also used for fitting, as they could, crucially, help inform relative incidence / hospitalization across age groups. The two main roles of contact matrices in a model are indeed to inform: 1) who is most at-risk (via mean total contacts per group) and 2) from whom (via non-random mixing patterns). Neither of these aspects is directly tested in the current experiments, so we cannot draw strong conclusions about how well the contact matrices capture transmission dynamics. This point is underscored by the comparable performance of the pre-pandemic matrix (mean error & Figure S5).

- Correcting factor (alpha): Variations in alpha over time for all three contact matrix cases (Figure 5b) were of a similar magnitude as variations in the mean numbers of contacts over time (Figure 2e-f), about 3-fold over time. We could crudely interpret this as: alpha explained about the same proportion of epidemic dynamics as the contact matrices themselves. While I commend the authors for explicitly modelling so many determinants of epidemic dynamics, the gap filled by alpha (~0.6--1.8 x reproduction number) is still large enough to almost completely absorb the influence of contact matrices on overall incidence / hospitalizations. While the magnitude of alpha varied across matrix approaches, time trends were highly

correlated across approaches (Figure 5b), suggesting that all matrices were failing to capture some key driver(s) of epidemic dynamics. The model is essentially overparameterized by alpha, which might allow reconstruction of hospitalizations using almost any contact matrices.

- One suggestion would be to incorporate specific correcting factors for age and/or contact-types, which could then be used to examine age/type-specific discrepancies between estimated matrix contacts versus effective contacts.

+ Comparing contact matrices for aim (i):

- The level of detail in the contact matrices is quite limited for a paper focused on comparing contact matrices (e.g. only 4 age groups). Perhaps this is a limitation of the available data, but it doesn't appear so (for either SocialCov or COMES-F). Also, it's well known that epidemic dynamics are fundamentally different with vs without risk heterogeneity (e.g. increasing R0 and differences in herd effects). I would have hoped to see more age groups and/or further stratification of age groups into high/low risk groups. Adding even two risk groups may help fit the data better, and thereby reduce the range of alpha in aim (ii).

- The comparison of empirical and synthetic contact matrices focuses almost entirely on total numbers of contacts per age group (rather than who contacts whom). This aspect is well done (e.g. time trends in discrepancies, and discrepancies by age group). However, differences in mixing *patterns* by both age and contact type should be explored to address this aim fully. It seems that contacts can be stratified by type in both data sources. For example, ages [0-10] and [11-18] had the largest discrepancies between empirical and synthetic matrices. While this was acknowledged, were differences driven by within or between age-group contacts, and for which types/locations of contacts? The four comparison metrics (Suppl 5) are useful, but these are discussed too briefly in the results and they remain rather abstract (e.g. what is a "low" measure of cosine similarity? Maybe a random mixing matrix can serve as a reference.)

- Relatedly, only Figure 1a (in main text or supplement) depicts a complete mixing matrix (all 4 x 4 age group combinations), but this reflects the pre-pandemic matrix. Give the aims of comparing contact matrices, I was expecting to see more illustrations of complete mixing matrices, so that changes to mixing *patterns* can be examined (not just total contacts as in Figure 2e-f).

Medium issues

- Overall, I found the methods-last structure of the paper (imposed by the journal I think) inefficient and confusing. The results section provides only a trickle of methods information, which distracted me from the results themselves with questions I had about the methods. However, reading the methods before the results section, ideas like the correcting factor were assumed to be already introduced. Still more methods are relegated to the supplement, so I am never quite sure where to look for a particular methods detail. An outline of the experiments is needed.

- A related issue is that the research questions / aims are not stated explicitly anywhere. Also, the last paragraph of the introduction includes: "Although mobility-based synthetic contact matrices ... their accuracy relative to empirical contact matrices remains unexplored" which suggests that the empirical matrices will be taken as a "gold standard" in comparisons, but this is not really how the results are framed. Similarly: "evaluating the effectiveness of these two approaches in modeling the pandemic dynamics" should define "effectiveness" and "pandemic dynamics" more concretely.

- It should be clarified in the methods how contacts were reported by individuals < 18 in waves 2-7 of SocialCov. In Suppl 4.3 [310], this is indirectly suggested, but only the discussion acknowledges that parents might have reported for children. This is notable since ages [0-10] had the largest discrepancies between synthetic and empiric matrices.

- Although acknowledged in the limitations, the *removal* of physical contacts (rather than replacing them with non-physical contacts) does not seem plausible. Since most covid-19 models make no distinction between these types, I think it is better to either ignore the level of avoidance (q at [273] in the Supplement) and treat all contacts equally, or use distinct rates of transmission (beta) for physical vs non-physical contacts in order to use q.

- Figure S1 is very helpful, but analogous diagrams for the other strata (especially vaccination, prior infection, waning, and VoC) are needed too. These can be 1-2 separate subfigures (not drawing all compartments / joint strata in one diagram).

Minor Issues

Main

- [41] Somewhere in this paragraph, a more precise definition of "contact matrices" should be given, e.g. "in which matrix elements reflect the average number of contacts per day formed by individuals in one group with another."

- [58] I think "convenient" should be "convenience" here?

- [72] The term "contact layer" here and throughout might be confusing. Why not simply "contact types"?

- [73] "These data streams enabled to produce weekly ..." is not grammatically correct - add "enabled *us* to"?

- [101] "to reproduce age and gender distribution in France" same - add "reproduce *the* age"?
- [102] Consider adding "**average* daily contacts" for precision.
- [117] The meaning of "normalized contact matrices, which disregard contact intensity" is not clear - is this normalization redundant with the definition of cosine similarity?
- [138] As suggested above, another comparator could be pure random (so-called proportionate) mixing matrix -- which could be used in both aims (i) and (ii).
- [145] Why is the IQR reported, rather than 95% range? I would think the latter is more useful, but maybe both could be given like: median (Q2.5 [Q25, Q75] Q97.5)
- [147] "suggesting that small adjustments were needed" as noted above, I do not agree these are small, especially those alpha outside IQR ...
- [163] "confirmed overall these findings" grammar - maybe "**generally* confirmed these findings"
- [173] "strongly valuable" grammar - maybe "**highly* valuable"
- [197] "survey frequency of surveys" typo?
- [233] I would not fully agree that "However, these limitations critically depend on the mobility metrics chosen" since masking can erode the correlation with effectively no impact on mobility metrics.
- [245] It's not clear what is meant by "optimal methods for extending empirical contact matrices".
- [262] The subject of "underestimate transmission risks" is not clear here: isn't transmission risk overestimated by ignoring repeat contacts?
- [274] The word "robust" here is too strong.
- [333] The overall value of these correlations with Normalcy Index and Stringency Index with respect to the aims are not clear to me.
- [340] As far as I can tell, the only citation for the 7 SocialGov surveys is #25 in Eurosurveillance, which only comments on the first survey. This is notable since later surveys included all ages, while the first was restricted to 18+. I noticed when looking for information about how responses for young children/babies were obtained.
- [365-372] The implications of assortativity for reproduction number & herd effects might be stated briefly somewhere here.
- [375-408] I just want to reiterate that the level of model complexity is impressive.
- [414] Perhaps Table S7 could be explicitly mentioned here regarding time windows.
- [436] It would help to explain the rationale / strengths for each performance metric. For example, AIC would "punish" 20% error in predicting 50 or 500 hospitalizations more equally, whereas absolute error would "punish" 20% error on 500 hospitalizations ~10 times more than on 50.
- [437] Relatedly, is there any benefit to using AIC over simple (log-)likelihood? I think k is the same across approaches, but this is never stated explicitly.
- [462] I commend the authors on making their code available via GitHub.
- [Figure 1] Ideally, the matrix figure (a) could use a perceptually uniform colormap, and the proportion figure (b) use a different, qualitative map. The hashing for non/physical is very nice. Perhaps (d) could extend the range from 30-100 to 0-100% to minimize misinterpretation.
- [Figure 2] Overall these subfigures are excellent, and nicely communicate a lot of information.
- [Figure 3] (a-b) Instead of these subfigures, which omit between-group mixing and have indistinguishable data points, perhaps an image like 1a can show the difference matrix ($M1_{ij} - M2_{ij}$) so readers can visually see where differences are largest; I think that is the goal here? Also, perhaps (d) can use the same x-axis as (c), as I'm not sure what the current axis (% contacts < 19) is communicating?
- [Figure 4] Is excellent.
- [Figure 5] (and S4, S9): Perhaps alpha should be plotted in log2 y-scale, since it is a multiplicative factor? Also, is it possible to scale densities so their maximum width is consistent, to improve clarity (especially Figure S4)?

Supplement

- [65] Perhaps "dimensions" is a better term than "layers" to describe independent population stratifications?
- [76] Table S1 should clarify that the relative infectiousness with mild and severe symptoms (I_{ms} , I_{ss}) are the same.
- [78-105] To improve ease of use, this section (Suppl 2) could have the information added to Table S1 or summarized in a new Table S2?
- [86] I cannot find the base probability of transmission (β_{pre-LD}) anywhere. I understand this is fitted, but can the posterior mean (95%) be reported?
- [178] The elements in matrix C_{ij} are described as "per-capita rates of contact between age-group i and j ". This key definition is not very precise, especially to distinguish C_{ij} from M_{ij} . My understanding is that C_{ij} is equivalent to the "intrinsic connectivity matrix" in Arregui et al (2018) - perhaps this terminology can be used or added in parentheses for readers already familiar? Though, it does not add much precision. Another definition of C_{ij} is the expected number of contacts formed by a group- i individual with group- j individuals, if 100% of the population was in group- j , or equivalently, C_{ij} / N is the expected number of contacts formed by a group- i individual with group- j individuals per individual in group- j .
- [204-221] While I appreciate the authors' consideration that not all declared work contacts will be symmetrically reported (e.g. a bus driver and passenger may report one another as "work" and "travel" contacts, respectively), I would think that most public-facing occupations would be infeasible for remote-work, and that occupations where remote-work is feasible would include mainly "symmetrically reported" (work-work) contacts. So, I don't agree that the two situations described here are equally likely, hence the average: $(1-a) + (1-a)^2 / 2$; and I would suggest using the $(1-a)^2$ approach throughout, unless the authors can provide some examples of public-facing occupations where remote-work is feasible?
- [322-340] Suppl 4.4: The direct removal of contacts among those isolating essentially implies a density-dependent effect, since the isolating contacts are not "replaced" among everyone else. A frequency-dependent approach would add a lot of complexity to calculations with probably negligible difference in transmission because infection prevalence is never very high, so the current approach is fine, but it may be worth spelling out the implied assumption.
- [365-394] Supple 5.4: It may be more clear to simply omit the continuous domain functions and integrals, and focus on the discrete implementation via sums.
- [398] The force of infection equation shown here is extremely simplified compared to the modifiers described (and the GitHub code), such as infectiousness, susceptibility, etc. The true equation spelling out the various factors involved would be much clearer. Also, since the denominator " N " differs from the usual " N_j ", it may be worth reminding readers how this derives from the definition of " C_{ij} ".
- [492] Figure S5d should likely log-transform the y-axis to improve the visualization of differences, and because I think 50% relative difference is "equally bad" as 200%?

(Remarks on code availability)

I have only briefly examined the code on GitHub to verify questions I had. It seems reproducible but would need to be fully tested.

Reviewer #2

(Remarks to the Author)

Mobility-Driven synthetic contact matrices: a scalable solution for real-time pandemic response modelling

Overall, this is an interesting and well-written manuscript based on a thorough analysis, comparing the use of weekly synthetic transmission matrices, building on an elaborate method for exploring mobility data with matrices from online social survey waves to model the evolution of the pandemic in France. The study adds novelty and deserves publication. However, some clarification on the methods is warranted.

As pointed out by the authors, during a crisis where situational awareness are needed for policy recommendations, their method is easily automatized and can readily be used, compared to social survey studies that require data washing and more elaborate measurement.

Main comments:

The "empirical matrices" are based on data from the SocialCov survey, with a reference to Bosetti et al. Eurosurveillance 26 (2021). However, this reference only addresses the primary wave exclusively conducted among adults 18+ years. Hence, it is unclear how data was collected among children and adolescents in the later waves. Do they involve parents reporting on behalf of their children? Are adolescents included in the survey? This is important information and should be properly documented. A sentence in the discussion hints that parents were answering on behalf of their children.

The methods for calculation of a typical weekday is not specified. Presumably, the SocialCov survey includes information collected on different weekdays. How is a "typical" weekday calculated? There are likely differences between contacts on Mondays vs. mid-week.

It is not clear to this reader what specific contacts were used to construct the "empirical matrices". It is stated that both

physical and non-physical contacts were collected –if the total number of contacts were used, it should be stated.

The imputation process (for the first wave of data collection) is not specified, and there are no specifics on the sampling procedure when reweighing the data. Were the matrices bootstrapped to account for sampling variance?

The paper does not consider the French CoMix data because the survey was conducted over only 5 months (late 2020 to spring 2021). However, the CoMix survey design allows for a more detailed description of contacts in children. I believe it would be informative and of general interest to include this data and compare average connectivity, cosine similarity and the proportion of young connectivity, even if the time period is restricted.

How was the serological status calculated for vaccinated individuals?

Minor comments:

Line 55: It is stated that (in the CoMix survey) only UK, Belgium and the Netherlands covered the first pandemic wave (also expressed in the reference that it cited). However, Norway was part of the CoMix survey and conducted six waves between April and September 2020, <https://bmcpublihealth.biomedcentral.com/articles/10.1186/s12889-024-18853-8>

Line 76 “accuracy” relative to empirical contact matrices. I would rephrase as both the synthetic and empirical matrices are estimates.

Suppl. L 316. The matrix obtained from the first wave of data collection is named LD, while the national lockdowns are named LD1-LD3. Does this imply that this matrix was also used for later national lockdowns?

(Remarks on code availability)

Reviewer #3

(Remarks to the Author)

This manuscript addresses an important question about whether it is sufficient to use ‘synthetic’ contact matrices, constructed using mobility and other publicly available data, or if it is essential to undertake resource-intensive efforts to collect empirical contact data to capture changing behaviors. The authors address this question in the context of the COVID-19 pandemic, though their findings are likely generalizable to other circumstances in which behavior may be changing substantially over the course of an epidemic. The authors undertake a detailed assessment of this question, by descriptively comparing contact matrices generated by each approach, evaluating the extent to which different contact assumptions align with hospitalization data and seroprevalence estimates from 2020-22. This is a thorough, clear, and important paper.

I have a few minor comments for the authors to consider.

1. Some discussion of the range and distribution of contacts reported, and if / how this was taken into account in the modeling, should be included. The matrices are nicely shown in Fig 1, but all that is shown is the averages (and averages by age group). Were the full distributions of contact accounted for in the modeling, or were only the average rates by age used? I think the latter is defensible, but for an airborne infection such as SARS-CoV-2 where contact heterogeneity (‘superspreading’) may play an important role in transmission dynamics, the distributional assumptions should be noted.
2. Were there any important differences (contact definition, mode of survey, survey structure, etc.) in the data collection for the empirical contact studies during the pandemic and the study from which the pre-pandemic baseline was drawn for the synthetic matrices (2012 survey)? In the author’s opinion, would these design differences contribute to the differences in fit under the synthetic vs. empirical data?
3. Please provide an explanation of ‘cosine similarity’. I am unfamiliar with this method and a brief explanation would help a reader to understand how to best interpret this analysis.
4. I have not heard of the Normalcy Index (only the Stringency Index), but it is the only one shown in Fig 2c as a comparison to the synthetic contact rates. Can the Stringency index be shown as well? I understand this to be the more commonly used measure. If the authors would like to show the Normalcy Index, some additional explanation of how the indicator is constructed would be useful. Are the ‘indicators’ being used to construct the Normalcy Index similar to those being used to generate the synthetic contact patterns? If so, we would already expect them to be correlated.
5. Lovely figures, but I think Figure 3 is largely redundant with Figure 2. Would recommend placing Figure 3 in the Supplement.
6. I do not understand Figure 3D; more explanation may be required.
7. A comment - the authors note this in their Discussion, but I feel that setting this up as a comparison of the empirical and synthetic contact data is somewhat unfair, since the empirical studies were conducted so sparsely across the pandemic. In fact, the empirical studies seem to do surprisingly well given the low frequency sampling.
8. From Figure 4a, it appears that all of the contact assumptions allow for a reasonably good fit to hospitalization data. How do the authors reconcile this with the quite large discrepancies between the synthetic and empirical contact rates? Can this be attributed to the low rate of hospitalization among kids, among whom the discrepancy between synthetic and empirical rates was highest?

Overall, an important contribution to the literature on this topic.

(Remarks on code availability)

Version 1:

Reviewer comments:

Reviewer #1

(Remarks to the Author)

Summary

The authors have done extensive edits and additional simulations, which have fully addressed the medium/minor issues, and mainly addressed the major issues. Some issues remain:

Major Issues

+ Inference framework for aim (ii):

- Interpretation of correcting factor (alpha): Following new sensitivity analyses of several alternate synthetic contact matrices, where alpha distributions remained similar to those in main analyses, the authors conclude: "the correcting factor is not a reliable indicator of contact matrix performance." I disagree -- this was not my concern with alpha: values near 1 would indeed indicate contact matrices are "correct" given the model & calibration targets. Rather, I still feel the issue here is that the calibration targets provide little information / error signal regarding mixing *patterns*, especially in the main paper, without age-stratification.

- Calibration targets: I still feel that all available age-stratified data (hospitalizations & seroprevalence) should be used as calibration targets, and in the main analyses (not just supplement). The justification about mimicking real-time modelling does not make sense to me, because it discards information we do actually have now, that could better answer the overarching question before the next epidemic (whether new contact surveys are needed during the epidemic). This real-time framing is also not really internally consistent, given the need to fit alpha using hospitalization data from multiple weeks into the future.

+ Sensitivity analyses contact matrices in aim (ii):

- More intuitive names for cases would be very helpful, e.g. TEST1: "adult-,elderly+", TEST2: "random mixing", TEST3: "less within-group", TEST4: "keep physical" etc.

- Random mixing: For TEST2, "random mixing" should not modify the total numbers of contacts per group. If $X_i = N_i * C_i$ is the total numbers of contacts "offered" by group i , then $X_{ij} = \text{outer}(X_i, X_i) / \text{sum}(X_i)$ gives the random mixing matrix at the population-level, and $M_{ij} = X_{ij} / N_i$ gives the per-person matrix. This is notable because TEST2 performed much worse than the main synthetic or TEST3 matrices, likely due to changing the numbers of contacts per group, but a comparison with random mixing that maintains group-specific contacts would be very useful.

- Reciprocity: Throughout the additional contact matrices explored (e.g. TEST 1-3, Supp 9.4), it's not clear if resulting contact matrices are re-balanced for reciprocity after modifications. Contact matrices violating reciprocity will effectively cause modelled infections to be transmitted "into the void" or acquired "from the void" for various groups, depending on the direction of imbalance -- which evidently undermines the objective of assessing contact matrix performance.

* I suspect matrices have not been re-balanced because, in TEST3, it is not possible to reduce within-group contacts by 25% for every group simultaneously, while maintaining reciprocity. For example, in the 2x2 case: suppose $N_i = [80, 20]$, $C_i = [1, 6]$. Under random mixing (without loss of generality) we have $X_{ij} = [[32, 48], [48, 72]]$ at the population-level. If we reduce the diagonal 32 and 72 each by 25% (yielding 24 and 54), the missing contacts to add back to the off-diagonals are 8 and 18, which would violate reciprocity since X_{ij} must remain symmetric.

+ The added results on element-wise comparison of contact synthetic vs empirical matrices for aim (i) is great.

Medium Issues

- Figure 2: This is a great addition. However, I think appending the absolute & relative differences as shown in Figures S10 & S11 to Figure 2 would significantly improve the ease of appreciating differences among contacts within the main text. Relatedly, the colour scales should be symmetric (so +1 vs -1 absolute differences are the same intensity, and likewise for 2 vs 1/2 relative differences).

- [182] Regarding: "empirical and pre-pandemic matrix models showed abrupt and unrealistic shifts in epidemic trends" - is this not just the result of modelling of alpha as piecewise constant, rather than deficiencies in the empiric data themselves? It could likely be solved using a spline for alpha instead, with knots at the current intervals. I would rephrase this.

- The manuscript makes several references to "infections" as a modelled outcome to compare with data, but the available data (and actual modelled outcome) reflect seroprevalence, which can wane over time due to seroreversion, so is not synonymous with cumulative infections. The more precise term "seroprevalence" should be used throughout.

- GitHub: I cannot find the code which actually runs the model fitting. Is alpha for each phase calibrated separately, or jointly?

Minor Issues

- I appreciate the addition of explicit aims at the end of the introduction. However, I feel the language used lacks precision:

what are "contact patterns" and "contact volumes" - why not use language defined above: "contact matrices"? e.g. "First, we compared *synthetic* contact matrices derived from mobility data versus *empirical* matrices derived from social contact surveys..." and to sign-post the key words "synthetic" and "empirical" used throughout. Similarly, what is "capacity of each matrix type to support epidemic modelling of covid-19 ..." - why not "reproduce (age-specific) covid-19 hospitalization (and seroprevalence) data within a transmission modelling framework"?

- Figure 2: are these data before or after correcting for reciprocity?

- [resp.p15] "We also argue that the ability of all models to reconstruct total hospitalizations is not primarily due to the flexibility of alpha, but rather to the modeling framework itself. In particular, alpha was fitted over short, discrete time windows (averaging 3 weeks), which naturally limits the accumulation of large fitting errors in total hospitalizations." Since alpha is essentially the only fitted parameter throughout the epidemic, I do not understand the point being made here.

- [resp.p18] Regarding age-specific correcting factors: To reduce overfitting, multiple group-specific calibration targets can be used together (e.g. hospitalizations, seroprevalence, etc.), alongside informative priors for fitted parameters (e.g. log-normal with mean 1 for alphas), with iterative sampling methods to improve inference efficiency (e.g. doi.org/10.1111/j.1541-0420.2010.01399.x). Relatedly, such factors do not need to distort mixing, since mixing preferences can be maintained even with changing group-specific marginal contact rates (e.g. doi.org/10.1016/0025-5564(91)90014-a). So, age-specific alphas are indeed feasible and would be useful.

- [resp.p18] Relatedly: "Using more age groups would have forced us to introduce additional assumptions to extrapolate or impute this missing behavioral information, thereby reducing the empirical grounding of the matrices." Sorry, this is not true: you have already made the same number of assumptions, they are just grouped together and less explicit (e.g. all individuals aged 19-65 are identical). In other words, when counting unique assumptions, some are "bigger" than others.

- The supplement is missing page numbers.

(Remarks on code availability)

I could not find the code which actually runs calibration / model fitting on github - e.g. the "likelihood" function is defined but not called, and data is loaded from a file calibration.csv. Some parameters seem to have different names from the paper, which is confusing, but not terrible. I did not try running the code myself.

Reviewer #2

(Remarks to the Author)

The authors have conducted a thorough revision of the manuscript, improving its clarity, exposition, and methodological descriptions, and adding analyses that further support their conclusions. I find the article substantive and offering novel insights into the complex challenge of representing contacts in infectious-disease models, with clear relevance for real-time crisis response.

All of my prior comments have been addressed appropriately, and I have no further comments.

(Remarks on code availability)

Reviewer #3

(Remarks to the Author)

This is a thorough response and the authors have adequately addressed my previous concerns.

(Remarks on code availability)

I reviewed the code previously.

Version 2:

Reviewer comments:

Reviewer #1

(Remarks to the Author)

The authors have now fully addressed my concerns with the manuscript. I again commend the extensive additional simulations, professional figures, and the updated analysis code on GitHub for transparency.

(Remarks on code availability)

I briefly reviewed the code at a high level. It appears to be reproducible through Python / Jupyter notebooks, but I did not attempt to run it myself.

POINT-BY-POINT REPLY TO REVIEWERS' COMMENTS

Mobility-driven synthetic contact matrices: a scalable solution for real-time pandemic response modeling

Laura Di Domenico, Paolo Bosetti, Chiara E. Sabbatini, Lulla Opatowski, Vittoria Colizza

Table of Contents

REVIEWER #1	2
Major issues.....	3
Medium issues.....	23
Minor Issues.....	28
Main.....	28
Supplement.....	34
REVIEWER #2	38
Main comments.....	38
Minor comments.....	43
REVIEWER #3	45
Minor points.....	45
REBUTTAL APPENDIX	50

REVIEWER #1

The authors present a comparison of two methods to generate dynamic contact matrices among 4 age groups for France between 2020 March and 2022 May: 1) synthetic matrices made by adjusting pre-pandemic contact matrices using mobility data and assumptions; and 2) empirical matrices, estimated from 7 serial surveys in France over a similar period, adjusted for non-representative sampling. As a control case 3), a static pre-pandemic matrix was also compared. The aims explored are interesting and important: i) how do these matrices compare to each other, and ii) which method performs better for transmission modelling (of covid-19), especially given that (1) can likely provide more timely and inexpensive estimates than (2).

The model design for aim (ii) includes numerous important features for covid-19 dynamics, such as age-stratified susceptibility & hospitalization, different viral strains, empiric vaccine uptake over time, natural/vaccine/hybrid & waning immunity, which all appear to be diligently parameterized. The discussion is also overall excellent, except some overstated conclusions. However, the inference framework for aim (ii) does not allow for the necessary evidence to address aim (ii) and aim (i) remains insufficiently explored. My overall recommendation is to split these aims into two papers to address each more thoroughly. Due to the above, I do not find the current results particularly useful.

We thank the reviewer for the detailed assessment of our manuscript and for considering the discussion of our findings overall excellent. In this revised version, we have significantly expanded and deepened the analyses to strengthen the robustness of our findings and better support both aims of the study.

For aim (i) - comparing contact matrices - we have enriched the analysis with additional quantitative comparisons and structural metrics to provide a clearer, more comprehensive picture of how the synthetic, empirical, and pre-pandemic matrices differ over time, and in their age-specific characteristics and mixing patterns. We have further applied these metrics to the test matrices introduced in the additional analyses performed in this revision.

For aim (ii) — evaluating matrix performance in transmission modeling — we conducted a set of new analyses exploring alternative calibration targets and systematically varying the structure of the contact matrices. These analyses directly address the reviewer's concerns about the inference framework. In particular, they shed light on the specific limitations of the correcting factor, as originally pointed out by the reviewer. Although the correcting factor provides flexibility in fitting overall hospital incidence, our results show that it cannot compensate for modifications in age-specific contact volumes or mixing patterns. Changes in either of these features had clear and measurable effects on age-stratified epidemic outcomes, which the correcting factor was unable to mitigate. This confirms the critical importance of age-specific contact structure — both in terms of who is at risk and from whom — in shaping transmission dynamics. **The revised manuscript now clarifies this distinction, and better defines the scope and limits of the correcting factor in the inference framework.**

Given the length of this rebuttal, we report here a summary of the new tests performed and their key conclusions, before entering the point-by-point reply to the reviewer's comments:

- Sensitivity analysis with age-stratified hospitalization data as calibration target → Demonstrated that model results were robust to this alternative fitting approach, with no change in correcting factors or fit quality (**Fig. S19-S21** and **Table S13**).
- TEST1: Altered total number of contacts per age group while preserving overall contact volume → Showed that changing who is most at-risk affected epidemic outcomes (hospitalizations and infections by age), while the correcting factor remained mostly unaffected (**Fig. S22-S24** and **Table S14**). This demonstrates that (1) the correcting factor is not a reliable indicator of contact matrix performance for transmission modeling and (2) it cannot absorb structural distortions in age-specific contact volumes. The time series of contacts per age group in the synthetic matrices is therefore essential.

- TEST2: Imposed random mixing while preserving total contacts → Demonstrated poor model performance during Omicron and in adults/seniors, highlighting the importance of non-random mixing patterns generated in the synthetic matrices (**Fig. S25-S27** and **Table S15**). The correcting factor showed the same limitations as in TEST1.
- TEST3: Reduced assortativity without altering total contacts per group → Isolated the effect of non-random mixing; deviations from observed data confirmed that structural features of the matrix (from whom contacts arise) strongly impact age-specific outcomes (**Fig. S25-S27** and **Table S15**). The time series of mixing patterns across age groups in the synthetic matrices is therefore essential. The correcting factor showed the same limitations as in TEST1.
- Comparison with pre-pandemic and empirical matrices → Highlighted that synthetic matrices better captured smooth epidemic transitions (e.g., post-wave 2, curfew, **Fig S4**), thanks to their dynamic updating — a key operational advantage for real-time modeling.

We have substantially revised the manuscript to incorporate both the new analyses and a clearer, more accurate interpretation of the findings. With these substantive additions and clarifications, we believe the revised manuscript offers robust support for both aims of the study and provides a comprehensive response to the reviewer's key criticisms.

Major issues

+ Model-based inference framework for aim (ii): Conclusions such as "Our study demonstrates that mobility-based synthetic matrices performed well in capturing the dynamic changes in contact behavior across age groups throughout the pandemic" are not well supported by the results, and are in some ways unfalsifiable in the current analysis framework, due to the unconstrained correcting factor α and lack of age-specific calibration targets. In other words, what results would cause the authors to draw a different conclusion?

- Calibration targets: The only data used to fit the model were total hospitalizations per week. However, age-stratified hospitalization data appeared to be available (Suppl 7.3 & Figure S5), as well as age-stratified seropositivity prevalence over time (Figures 5 & S9). It's not clear why these age-stratified data weren't also used for fitting, as they could, crucially, help inform relative incidence / hospitalization across age groups. The two main roles of contact matrices in a model are indeed to inform: 1) who is most at-risk (via mean total contacts per group) and 2) from whom (via non-random mixing patterns). Neither of these aspects is directly tested in the current experiments, so we cannot draw strong conclusions about how well the contact matrices capture transmission dynamics. This point is underscored by the comparable performance of the pre-pandemic matrix (mean error & Figure S5).

To respond to the reviewer's comments, we carried out several additional tests including (i) age-stratified calibration targets and (ii) variations of the synthetic matrices in terms of mean total contacts per group (point 1 in the reviewer's comment) and mixing patterns (point 2). This allowed us to draw more accurate conclusions on the role of the contact matrices in capturing the transmission dynamics.

Age-stratified calibration targets

We did not use age-stratified seropositivity data as calibration targets due to operational constraints, as these data were not available in real time and only became accessible several months later. While this study is conducted retrospectively, its aim is to assess the applicability of mobility-informed synthetic contact matrices for real-time pandemic response. To this end, we aimed to replicate the conditions that are likely to arise during an actual pandemic. In the context of France, serological data are not expected to be readily available in near real-time, as there is currently no planned systematic framework for routinely testing a representative sample of the population, such as the one

implemented in the UK during COVID-19. For these reasons, we used age-stratified serological data for validation purposes rather than for calibration.

On the other hand, age-stratified hospital data were routinely available throughout the crisis, but were not used for fitting purposes in our original manuscript. As with the serological data, we initially used age-stratified hospital data as a form of real-time validation of the fit - whereas serological data, by contrast, could only serve for validation months later. Prompted by the reviewer's remark, we performed a sensitivity analysis using age-stratified hospitalization data for the fit. The likelihood is as follows:

$$L(Data|\theta) = \prod_{t=t_1}^{t_n} \prod_{age=1}^4 Poiss(H_{obs}^{age}(t)|H_{pred}^{age}(t, \theta)),$$

where the terms H_{obs}^{age} and H_{pred}^{age} represent daily hospitalizations per age class, observed or predicted by the model, respectively, with $age \in \{[0-10], [11-18], [19-65], 65+\}$. We fitted the three main models using this likelihood.

The results (shown below) are robust to this alternative specification of the likelihood including age-stratified hospital data. No significant variation is observed in the resulting correcting factors, in the mean absolute errors on total hospitalizations, or in the age-stratified outcomes (**Fig. S19-S21**). The model built on synthetic matrices continues to be preferred in terms of AIC (**Table S13**), supporting the consistency of our conclusions.

This sensitivity analysis is now included in the SI of the revised manuscript (Section 9.3) and mentioned in the Methods, Results and Discussion of the revised manuscript. We report here the changes made to the manuscript.

Methods (lines 555-558)

We carried out sensitivity analyses on [...] (iii) an alternative specification of the likelihood using age-stratified hospital data.

Results (lines 240-243)

The distribution of the correcting factor and the results of the model comparison were robust to an alternative specification of the likelihood using age-stratified hospital data as calibration targets rather than total admissions (Figs. S19-S21).

Discussion (lines 263-264)

All models well captured the overall trajectory of total hospitalizations, and results remained robust when using age-stratified hospitalizations as calibration targets.

Table S13. Goodness of fit of the three models, using age-stratified calibration targets.

Model	AIC (original version)	AIC (REVISED)
Synthetic	27624	67420
Empirical	34356	94992
Pre-pandemic	34690	69353

Figure S19. Sensitivity of the model fit using age-stratified calibration targets.

Figure S20. Sensitivity in the cumulative number of predicted hospitalizations, using a likelihood with age-stratified calibration targets.

Figure S21. Sensitivity in the proportion of antibody-positive population, using a likelihood with age-stratified calibration targets.

Variations in the total number of contacts per age group

We then tested the role of the synthetic contact matrix in the model relative to the aspect (1) mentioned by the reviewer, i.e. the total number of contacts per age group as a measure of who is most at-risk. To this purpose, we generated a modified set of time-varying contact matrices (**TEST1**), derived from the original synthetic contact matrices by altering the average number of contacts per age group, while preserving the total number of contacts in the population at each time step. Specifically, at each time step, we reduced the average number of contacts engaged by adults by

67% (in the ballpark of observed contact reductions) and increased those in the elderly to preserve the total number of contacts of the original synthetic matrix (**Fig. S22**). Contact levels in adolescents and children were left unchanged.

Figure S22. Comparison of synthetic matrices and matrices used in TEST1.

We then fitted the transmission model using the TEST1 sequence of matrices, calibrating on total hospitalizations. The results are summarized below:

1. The correcting factor was lower than with the original synthetic matrices, but still close to 1 (**Fig. S23a,b**; median 0.87 and quantiles 0.44 [0.65, 1.01] 1.37 for model TEST1; median 1.16 and quantiles 0.6 [0.87, 1.33] 1.81 for the synthetic model).
2. The mean absolute error on the full trajectory of total hospitalizations was slightly higher than the original model, with discrepancies visible especially in the Omicron phase (**Fig. S23c**). Substantial deviations emerged in age-specific outcomes: hospitalizations among adults and seniors deviated significantly from observed data under TEST1 (**Fig. S24a**), even when fitting age-stratified hospitalizations (**Fig. R2**, Rebuttal Appendix).
3. TEST1 led to lower predicted infections in adolescents and adults relative to serological data (**Fig. S24c,d**).
4. In terms of model fit, the test model performed worse than the original synthetic model according to the Akaike Information Criterion (AIC, **Table S14**).

These findings collectively indicate that the correcting factor alone (and in particular, its deviation from the value 1) is not a reliable indicator for the overall performance of synthetic contact matrices in capturing disease-relevant mixing patterns during the pandemic. The TEST1 experiment shows that substantially modifying contact rates for adults and seniors - while preserving the overall number of contacts - has a relatively small impact on the correcting factor values obtained through model fitting. Therefore, on this point, we agree with the reviewer's criticism and **have modified our conclusions in reference to this aspect**.

On the other hand, despite providing larger flexibility to the model (as will be shown in later analyses), **the correcting factor cannot compensate for the impact that structural variations in contact**

patterns have on the age-stratified trajectories. The fit results from TEST1 show that these altered contact patterns led to an underestimation of hospitalizations in adults, an overestimation in seniors, and reduced infection levels in both adolescents and adults — outcomes that persisted even when age-stratified fitting was used. These deviations from observations underscore that the role #1 of the contact matrix — as identified by the reviewer, namely, identifying who is most at risk through group-specific contact rates — has epidemic consequences that cannot be offset by adjusting a global correcting factor.

The TEST1 experiment therefore directly addresses part (1) of the reviewer's criticism above. It is now reported in the SI (Section 9.4), and its relevance for interpreting the correcting factor has been added to the revised Results and Discussion sections of the paper. We report changes made to the manuscript at the end of this reply (see Conclusions section).

In the following section of our response, we address aspect #2 of the role of the contact matrix, namely its capacity to capture *from whom* the risk originates — that is, the structure of non-random mixing across age groups.

Table S14. Goodness of fit of the model, using the synthetic matrices or the matrices in TEST1.

Model	AIC
Synthetic	27624
TEST1	27885

Figure S23. Sensitivity of model fit when using the synthetic matrices or matrices in TEST1.

Figure S24. Sensitivity of model validation when using the synthetic matrices or matrices in TEST1. Notice that the span of the y scale in panel a differs in one order of magnitude compared to analogous panels for the other tests.

Variation in mixing patterns

To assess the impact of non-random mixing across age groups, we carried out two additional tests. In **TEST2**, we generated a sequence of time-varying contact matrices under the assumption of random mixing. These matrices preserved the total number of contacts in the population from the original synthetic matrices but redistributed them proportionally to the population sizes of each age group. This was the test suggested by the reviewer. However, this test also altered the number of contacts per age group — a factor previously investigated in TEST1.

To disentangle these two effects, we introduced **TEST3**, which altered the structure of mixing without changing the total number of contacts per age group. Specifically, we reduced within-group contacts by 25% — consistent with empirically observed variations — and reallocated the resulting surplus to between-group contacts, proportionally to the original contact rates. This approach allowed us to reduce assortativity while preserving the total number of contacts per age group, therefore isolating the role of contact structure in shaping transmission dynamics.

We measured the differences of TEST2 and TEST3 matrices with the original synthetic matrices across the following metrics (**Fig. S25**):

- cosine similarity;
- assortativity index;
- proportion of contacts produced by <19 y.o.;
- age-specific proportion of assortative contacts, defined as the fraction of contacts that individuals in a given age group establish within their own group. For each age group i , this corresponds to the ratio $M_{ii}/\sum_j M_{ij}$, where M_{ii} is the number of within-group contacts and $\sum_j M_{ij}$ is the total number of contacts for group i . This last measure was not present in our original submission, and has been introduced in the revised manuscript to respond to another comment of this reviewer (below).

Figure S25. Comparison of matrices in TEST2 and TEST3 with the synthetic matrices.

We then fitted the transmission model using these new sequences of matrices, TEST2 and TEST3, calibrating on total hospitalizations. The results are summarized below:

1. The correcting factor remained robust against the changes in the matrices introduced in TEST2 and TEST3 (**Fig. S26a,b,c**).
2. The mean absolute error on total hospitalizations was comparable to that of the original model for TEST3 in all pandemic phases, while TEST2 produced markedly higher errors during the Omicron wave (**Fig. S26d**). However, substantial deviations emerged in age-specific outcomes: both TEST2 and TEST3 produced higher hospitalizations in seniors and lower hospitalizations in adults (**Fig. S27a**), even when fitting age-stratified hospitalizations (**Fig. R3-R4**, Rebuttal Appendix).
3. TEST3 led to lower predicted infections in adults — and especially in adolescents — relative to serological data (**Fig. S27c,d**).
4. In terms of model fit, both test models performed worse than the original synthetic model according to the Akaike Information Criterion (AIC), particularly TEST2 (**Table S15**).

These results underscore the importance of the structure of age-specific mixing in shaping epidemic outcomes. TEST2, which assumed random mixing, failed to adequately reproduce observed hospitalizations — particularly during the Omicron wave, and more generally across the pandemic for adults and seniors. This highlights the critical role of non-random mixing in accurately capturing age-specific contributions to spread and disease burden. Part of these discrepancies, however, stem from the fact that TEST2 also altered the number of contacts per age group. To isolate the role of mixing structure, TEST3 introduced a more targeted modification by reducing assortativity while preserving the total number of contacts per group. Despite this constraint, the model still produced notable age-specific deviations from observed hospitalization and serological data. Together, these findings demonstrate that it is not only the number of contacts per group that matters (point investigated above, in TEST1), but also *from whom* those contacts arise (as highlighted by TEST3). The internal structure of the contact matrix — even when overall contact volumes are preserved — has substantial influence on both the magnitude and distribution of epidemic indicators. This reinforces the role of synthetic contact matrices in capturing time-varying non-random mixing patterns as a fundamental component of realistic transmission modeling, beyond what a global scaling factor can adjust for.

These tests therefore directly address part (2) of the reviewer's criticism above.

The TEST2 and TEST3 experiments are now reported in the SI (Section 9.5), and their relevance for interpreting the role of contact matrices has been added to the revised Results and Discussion sections of the paper. We report changes made to the manuscript at the end of this reply (see Conclusions section).

Table S15. Goodness of fit of the model, using the original synthetic matrices, matrices in TEST2 or matrices in TEST3.

Model	AIC
Synthetic	27624
TEST2	30133
TEST3	27716

Figure S26. Sensitivity of model fit when using the synthetic matrices or matrices in TEST2 or TEST3.

Figure S27. Sensitivity of model validation when using the synthetic matrices or matrices in TEST2 or TEST3.

Comparison with pre-pandemic matrix

We now focus on the part of the reviewer’s comment relative to the comparable performance of the pre-pandemic contact matrix.

The two main roles of contact matrices in a model are indeed to inform: 1) who is most at-risk (via mean total contacts per group) and 2) from whom (via non-random mixing patterns). Neither of these aspects is directly tested in the current experiments, so we cannot draw strong conclusions about how well the contact matrices capture transmission dynamics. This point is underscored by the comparable performance of the pre-pandemic matrix (mean error & Figure S5).

We acknowledge that the model informed with the pre-pandemic matrix produced results — in terms of mean absolute error (MAE) and age-stratified outcomes — that were closer to those obtained with the synthetic matrices than with the empirical ones. However, important distinctions remain. First, in terms of age-stratified hospitalizations, the model informed with the pre-pandemic matrix produced lower hospitalizations in seniors and higher hospitalizations in adults, compared to the observed data and the synthetic model (Fig. S8b, also reported in the main text as Fig. 5d). Moreover, during periods marked by gradual changes in social behavior — such as school closures, curfews, and gradual lifting of restrictions — the model using synthetic contact matrices more accurately captured the trajectory of hospital admissions (Fig. S4). For example, during the interval between the second wave and the onset of the third ("wave 2" to "curfew"), only the synthetic matrix model accurately reproduced the decline and subsequent resurgence in hospitalizations. In contrast, both the empirical and pre-pandemic matrix models produced discontinuities or "jumps" in the epidemic curve, reflecting unrealistic abrupt shifts in epidemic trends. This pattern suggests that, despite the flexibility of the correcting factor, these matrices failed to track gradual behavioral and policy-driven shifts in contact patterns, which are essential for realistic simulation of transmission dynamics.

From an operational perspective, this result is particularly important. In real-time modeling to support public health decision-making — such as forecasting under evolving intervention scenarios —

accurately capturing the dynamic evolution of the epidemic trajectory is critical. The higher performance of the synthetic model in these transitional phases underscores their added value for situational awareness, scenario evaluation, and timely policy response.

Fig. S4 has been included in the revised Supplementary Information, and its relevance for comparing the synthetic and pre-pandemic model has been added to the revised Results and Discussion sections of the paper. We report here the changes made to the manuscript.

Results (lines 175-184, lines 198-204)

All models captured the observed trajectory of daily hospital admissions well [...]. However, the quality of fit varied across models. It was best in terms of both AIC (Table S9) and mean absolute error (Fig. 5c) for the model parameterized with synthetic contact matrices. Notably, this model captured smooth transitions in the epidemic trajectory during periods of gradual behavioral changes (e.g. curfews and school closures between the second wave and third wave), whereas the empirical and pre-pandemic matrix models showed abrupt and unrealistic shifts in epidemic trends (Fig. S4).

[...]

Age-specific hospitalization patterns, which were not used to fit the models, were more accurately reproduced by the model parameterized with synthetic contact matrices, particularly for adolescents, adults and seniors. [...] The model based on the pre-pandemic contact matrix underestimated hospitalizations among seniors (Fig. 5d, Fig. S8), likely reflecting outdated contact patterns not representative of pandemic conditions.

Discussion (lines 267-275)

[...] the model informed by weekly synthetic contact matrices outperformed those using empirical or pre-pandemic matrices, achieving better fit metrics (AIC and mean absolute error). Its performance was particularly strong during transitional phases of the epidemic, such as periods marked by school closures, curfews, or gradual easing of restrictions, where it captured smoother changes in hospitalization trends. In contrast, models using empirical or pre-pandemic contact patterns produced unrealistic discontinuities during these phases. The higher performance of the synthetic matrices in these contexts originates from their key advantage, i.e. the ability to be updated weekly. This update is essential for real-time modeling applications, supporting improved situational awareness, scenario planning, and timely policy response.

Figure S4. Model fit between the second and the third wave.

Figure 5d. Comparison of model predictions with hospital admission data by age class.

Conclusions

In light of these additional analyses, we are now able to more directly and rigorously address the reviewer’s criticisms. First, our sensitivity analysis using age-stratified hospitalization data as calibration targets confirmed the robustness of our model framework and its conclusions, regardless of the chosen fitting approach. Second, by systematically varying the structure of the synthetic contact matrices — either by altering age-specific contact volumes (TEST1) or modifying mixing patterns (TEST2 and TEST3) — **we demonstrated that both *who is most at risk* and *from whom the risk originates* are critical determinants of epidemic dynamics, despite the large flexibility provided by the correcting factor.** These structural properties significantly affect age-specific outcomes and cannot be captured or corrected by the correcting factor alone, whose stability across tests confirms its limited interpretative power in evaluating contact matrix performance. **On this point, we thank the reviewer for prompting us to do a more thorough evaluation of this parameter.** Importantly, the results confirm that both the age-specific number of contacts and their distribution across age groups — as encoded in the synthetic matrices — are essential to reproducing observed epidemic trends. Finally, the higher ability of synthetic matrices to capture smooth epidemic transitions — especially in response to policy and behavioral changes — highlights their operational value for real-time epidemic modeling. Altogether, these findings substantiate the central role of mobility-informed synthetic contact matrices as a reliable, flexible, and interpretable tool for tracking and projecting age-specific transmission dynamics throughout a pandemic.

We report here the changes made to the manuscript.

Methods (lines 558-562)

To demonstrate that matrix structure (encoding which group is most at-risk for infection and from whom) plays a critical role in shaping epidemic outcomes, we also carried out multiple tests by systematically varying the structure of the synthetic matrices, introducing modifications in (iv) age-specific contact volumes or (v) mixing patterns.

Results (lines 215-226)

To disentangle the relative role of the correcting factor and the contact matrices in capturing observed transmission dynamics and age-specific trends, we carried out a series of sensitivity tests using modified versions of the synthetic matrices. These tests systematically altered who is most at risk (by changing age-specific contact volumes while preserving total contacts; TEST1 in the Supplementary Information, Figs. S22-S24) and from whom the risk comes (by modifying mixing patterns while preserving both total and age-specific contact volumes; TEST3, Figs. S25-S27). Additionally, we evaluated a model assuming random mixing (TEST2, Figs. S25-S27). Across all tests, the correcting factor remained comparable to that of the model parameterized with the original synthetic contact

matrices. However, substantial differences emerged in age-stratified outcomes – both for hospitalizations and infections – highlighting that the correcting factor alone cannot compensate for structural changes in contact patterns. Notably, none of the models integrating the test matrices improved model fit compared to the original synthetic matrix model (Tables S14-S15).

Discussion (lines 297-301):

Crucially, our tests showed that altering the structure of the contact matrices had substantial effects on age-specific outcomes even when the correcting factor remained stable, underscoring that this factor alone cannot compensate for mis-specification of mixing patterns. While it provides flexibility for the fit, it does not mask structural limitations in the underlying contact data. These results therefore also underscore the central role of matrix structure in shaping model projections.

- Correcting factor (alpha): Variations in alpha over time for all three contact matrix cases (Figure 5b) were of a similar magnitude as variations in the mean numbers of contacts over time (Figure 2e-f), about 3-fold over time. We could crudely interpret this as: alpha explained about the same proportion of epidemic dynamics as the contact matrices themselves. While I commend the authors for explicitly modelling so many determinants of epidemic dynamics, the gap filled by alpha (~ 0.6--1.8 x reproduction number) is still large enough to almost completely absorb the influence of contact matrices on overall incidence / hospitalizations. While the magnitude of alpha varied across matrix approaches, time trends were highly correlated across approaches (Figure 5b), suggesting that all matrices were failing to capture some key driver(s) of epidemic dynamics. The model is essentially overparameterized by alpha, which might allow reconstruction of hospitalizations using almost any contact matrices.

We thank the reviewer for this insightful comment regarding the role and flexibility of the correcting factor. We agree that, in the original manuscript, the interpretation of alpha was underdeveloped.

In response, we have conducted a series of new analyses aimed at clarifying the relative contributions of alpha and contact matrix structure. These include a sensitivity analysis using age-stratified hospitalization data as calibration targets, as well as controlled perturbations of the synthetic contact matrices (**TEST1–3**), described before. These experiments revealed a key result: alpha remained relatively stable across tests, both in median value and distribution, even when the underlying matrix structure was substantially perturbed (**Fig. S23, S26**). However, these same matrix changes led to marked deviations from observations in age-specific outcomes, including hospitalizations and predicted infections by age, and worsened model fit (as reflected in the AIC, **Table S14, S15**). This indicates that while alpha can absorb some temporal variation — as expected for a parameter that accounts for unmodeled behavioral and epidemiological factors — it does not compensate for structural mismatches in contact matrices, especially those affecting age-specific mixing. In other words, even though alpha and the mean number of contacts exhibit comparable levels of temporal variability, they are not interchangeable in their effects on epidemic dynamics. **Therefore, these analyses demonstrate that alpha is not a reliable standalone indicator of model fit or contact matrix quality.**

We also argue that the ability of all models to reconstruct total hospitalizations is not primarily due to the flexibility of alpha, but rather to the modeling framework itself. In particular, alpha was fitted over short, discrete time windows (averaging 3 weeks), which naturally limits the accumulation of large fitting errors in total hospitalizations. In this sense, the structure of the fitting approach plays a role in enabling good agreement with aggregate outcomes. However, this does not imply that alpha absorbs the influence of the contact matrices. In fact, despite the same time-discretization applied across models, the mean absolute error and AIC of the model using empirical matrices remained significantly worse than those of the synthetic model. This indicates that

the weekly frequency of the synthetic matrices and their structure improved the epidemic trajectory. In this regard, we acknowledge that more frequent surveys could potentially benefit a model informed with empirical matrices. However, these data were not collected, and such collection would be more costly with respect to the synthetic model.

We have added this element to the Discussion (lines 263-281):

All models well captured the overall trajectory of total hospitalizations [...] This consistency comes from the inference framework, which relies on short, discrete time windows for estimating the correcting factor, limiting the accumulation of fitting errors and allowing sufficient flexibility to align model outputs with observed data. Despite this flexibility, the model informed by weekly synthetic contact matrices outperformed those using empirical or pre-pandemic matrices [...] The higher performance of the synthetic matrices in these contexts originates from their key advantage, i.e. the ability to be updated weekly [...] In contrast, empirical matrices generally have limited temporal resolution, therefore requiring strong assumptions to fill gaps between infrequent survey waves. While more frequent surveys could improve the empirical matrix model, they come with substantial resource demands and require automated tools and streamlined methods to transform survey data into actionable insights efficiently.

Moreover, we agree with the reviewer that the correlation in alpha across contact matrix types suggests the presence of unmodeled factors — such as changes in testing, masking, or contact duration — that affect transmission potential independently of contact frequency. **We have added this element in the revised Discussion** (lines 294-296) with the following sentence:

Interestingly, the correcting factor showed some temporal correlation across models, suggesting the presence of unmodeled factors — such as changes in testing, masking, or contact duration — that affect transmission potential independently of contact frequency.

To quantify the correlation of the correcting factor across models, we included a figure in the Supplementary Information (**Fig. S6**) with the Pearson's correlation coefficient. We report the figure below. We found that the correlation between correcting factors was the highest for the models informed with synthetic and empirical matrices.

Fig. S6. Correlation of correcting factors across models.

Conclusions

In conclusion, we agree with the reviewer that alpha, on its own, is not an interpretable indicator of contact matrix performance in capturing disease-relevant contacts. The expanded analyses in this revision provide a more rigorous framework for evaluating contact matrices, demonstrating that only matrices with both plausible age-specific contact volumes and structured mixing can accurately reproduce observed dynamics. **We have clarified these points in the revised Results, Discussion,**

and Methods sections. We report here the changes made to the manuscript. Some pieces of text were already mentioned in the previous replies.

Methods (lines 527-534)

This correcting factor can be thus interpreted as a global correction of the contact matrices used in the model. The product of the fitted α _phase at time t with the corresponding contact matrix represents the effective contacts needed to reproduce the observed epidemic dynamic. The closer the correcting factor to 1, the better the alignment between the time-varying contact matrices and the effective contact rates needed to reproduce the total epidemic dynamic. However, acting as a global rescaling of the contact matrix, the correcting factor cannot alter the underlying matrix structure and mixing, responsible for the age-stratified model outcomes.

[...]

While the correcting factor provides indication about the adjustment required to reproduce the total epidemic, the comparison of age-stratified model outcomes with observations is an indicator of the ability of the matrices to capture contact structure and mixing patterns.

Results (lines 170-174)

Our fitting approach estimated a baseline transmission rate per contact in the pre-lockdown phase and a time-varying correcting factor to adjust the transmission rate over time. This correcting factor acts as a global rescaling of the contact volumes of the matrices, accounting for latent drivers of transmission, without altering the internal structure of the contact matrices.

Discussion (lines 283-301)

The correcting factor estimated in our model acts primarily as a global rescaling of contact volumes, enabling alignment between modeled and observed transmission without altering the internal structure of the contact matrices. It pragmatically absorbs residual variation in transmission intensity not captured by available behavioral or epidemiological inputs. Both synthetic and empirical matrices required moderate adjustments for most of the study period to reproduce observed hospitalizations. The largest deviation from 1 of the correcting factor occurred for the synthetic matrices during summer 2020—the first summer of the COVID crisis—when contacts were highly impacted and not easily reflected by workplace presence or school calendars due to the summer holidays. While empirical matrices performed slightly better during this period, they were constructed retrospectively using data collected in summer 2021, illustrating the limitations of infrequent survey-based approaches for real-time use. By contrast, the model using the static pre-pandemic matrix required consistently large corrections to modulate contact volumes over time. Interestingly, the correcting factor showed some temporal correlation across models, suggesting the presence of unmodeled factors — such as changes in testing, masking, or contact duration — that affect transmission potential independently of contact frequency. Crucially, our tests showed that altering the structure of the contact matrices had substantial effects on age-specific outcomes even when the correcting factor remained stable, underscoring that this factor alone cannot compensate for mis-specification of mixing patterns. While it provides flexibility for the fit, it does not mask structural limitations in the underlying contact data. These results therefore also underscore the central role of matrix structure in shaping model projections.

- One suggestion would be to incorporate specific correcting factors for age and/or contact-types, which could then be used to examine age/type-specific discrepancies between estimated matrix contacts versus effective contacts.

We thank the reviewer for this suggestion. While we recognize the potential value of exploring stratified correcting factors, we would like to note that this recommendation appears to contrast with an earlier concern raised by the reviewer — namely, that the single correcting factor might provide *too much* flexibility in the current model. Introducing age- or contact-type-specific correcting factors would

significantly increase model flexibility and complexity, with a risk of overfitting, and thus exacerbate some of the issues previously identified.

In practice, there are three main challenges associated with incorporating multiple correcting factors: interpretability, identifiability, and computational feasibility. While a single correcting factor acts as a global rescaling of contact intensity — without fundamentally altering the structure of the contact matrix — introducing age-specific correcting factors would distort the matrix's internal structure. This would make it difficult to draw meaningful conclusions about which data sources better represent disease-relevant mixing patterns.

From a computational perspective, fitting multiple correcting factors would substantially increase the dimensionality of the parameter space and the risk of overfitting, especially given the short time windows used in the fitting procedure. To empirically test this idea, we attempted to fit four age-specific correcting factors during the exit phase from the first lockdown, using the age-stratified likelihood described earlier. We performed a grid search over values ranging from 1.16 to 1.36 (step size 0.02), and analyzed the 50 best-performing combinations. As shown in the figure in the Rebuttal Appendix (**Fig. R1**, left to right: children, adolescents, adults, seniors), we found that only the correcting factor for adults was well identified. This likely reflects the dominant role of adults in driving transmission dynamics in this pandemic phase, due to their larger population size, susceptibility and higher number of contacts. These results suggest that introducing age-specific α values does not meaningfully improve model inference and raises serious issues of identifiability and interpretability.

For similar reasons, we believe that adding correcting factors stratified by contact type (e.g., by location) would also be of limited value. Not only would this increase model complexity and reduce interpretability, but we also lack sufficient observational data to reliably parameterize such refinements.

In summary, while the idea of stratified correcting factors is conceptually appealing, it is not feasible or informative in the context of our current framework, and would undermine the interpretability and robustness of the results we aim to obtain.

+ Comparing contact matrices for aim (i):

- The level of detail in the contact matrices is quite limited for a paper focused on comparing contact matrices (e.g. only 4 age groups). Perhaps this is a limitation of the available data, but it doesn't appear so (for either SocialCov or COMES-F). Also, it's well known that epidemic dynamics are fundamentally different with vs without risk heterogeneity (e.g. increasing R_0 and differences in herd effects). I would have hoped to see more age groups and/or further stratification of age groups into high/low risk groups. Adding even two risk groups may help fit the data better, and thereby reduce the range of alpha in aim (ii).

In our study, we chose to use four broad age groups — [0–10], [11–18], [19–65], and 65+ — primarily to balance model complexity with computational feasibility and data availability for parameterization. First, the resolution of behavioral data required to parameterize the synthetic contact matrices imposed practical limits. For example, workplace mobility data — one of the key inputs used to model contact variation over time — was not disaggregated by age, preventing us from capturing differences in work-related mobility within the adult population. Using more age groups would have forced us to introduce additional assumptions to extrapolate or impute this missing behavioral information, thereby reducing the empirical grounding of the matrices.

Second, from an operational modeling perspective, working with a limited — but still epidemiologically and policy-wise meaningful — number of age groups strikes a balance between interpretability and real-time feasibility. While many outbreak response models used finer age stratification, they typically

assumed pre-pandemic static mixing patterns over time and simply recalibrated the largest eigenvalue of the matrix over time. Our framework, in contrast, explicitly integrates time-varying behavior, and keeping the number of age groups manageable enables us to capture dynamic changes with fewer assumptions.

As the reviewer also acknowledged the overall “impressive” complexity and level of detail achieved in our model, we believe that introducing additional layers of age or risk stratification — in the absence of sufficient real-time data — would compromise interpretability and increase the risk of overfitting.

This choice is now better motivated in the Methods section of the revised paper, and discussed among the limitations of the study. We report here the changes made to the manuscript.

Methods (lines 400-404)

We considered four age groups: children [0-10], adolescents [11-18], adults [19-64] and seniors (65+ years old). We chose a limited number of groups to parametrize the matrices using the available behavioral data (missing finer age-stratification) with minimal assumptions, while capturing epidemiologically relevant groups and ensuring interpretability and real-time feasibility of the framework.

Discussion (lines 340-345)

First, the mobility and behavioral data used to construct synthetic matrices (e.g., workplace attendance, preventive behavior) were not age-stratified, but some of these proxies indirectly introduced age specificity. However, no finer age breakdown was available. This limited us to four broad age classes, offering a practical trade-off between detail and real-time feasibility. In contrast, empirical matrices provide finer age resolution, which synthetic matrices could match if age-stratified mobility data were available [56].

- The comparison of empirical and synthetic contact matrices focuses almost entirely on total numbers of contacts per age group (rather than who contacts whom). This aspect is well done (e.g. time trends in discrepancies, and discrepancies by age group). However, differences in mixing *patterns* by both age and contact type should be explored to address this aim fully.

While **Fig. 3** is focused on total contacts per group, we addressed differences in mixing patterns by age in **Fig. 4** using four complementary metrics: scatter plot with element-by-element comparisons, cosine similarity, the assortativity index, and the proportion of contacts generated by young individuals.

To further strengthen this analysis, we have now expanded the investigation by including:

(i) illustrations of the mixing matrices for both the synthetic and SocialCov data (added as a newly

Fig. 2 in the main text)

(ii) illustrations of mixing matrices showing the element-by-element absolute differences and element-by-element ratios (**Fig. S10** and **S11** in the supplementary information);

(iii) a comparison of the fraction of contacts occurring within the same age group (i.e., on the diagonal of the matrix) to better assess age-assortative mixing (included as additional panels in the main text, **Fig. 4c,d**).

For the newly introduced metric (iii), we found that the proportion of contacts shared within the same group in the empirical matrices is lower compared to synthetic matrices, for all age groups except children. These new figures are also included here, in the next page of this rebuttal. **Given the more straightforward interpretation of the proportion of within-group contacts, compared to other metrics, we decided to replace the plots in the main text as Fig. 4c,d. We moved the panels**

showing the cosine similarity, the assortativity index, and the proportion of contacts generated by young individuals to the Supplementary Information (Fig. S12).

We have extended the corresponding section of the Results to provide more detailed insights into the observed differences in mixing patterns across the matrices, mentioning the newly added figures. We report here the changes made to the manuscript.

Results (lines 115-116, lines 136-154)

We first compared the two sources of matrices in terms of average number of contacts (overall and by age group). [...]

We then compared contact matrices in terms of mixing patterns using a set of metrics accounting for matrix elements, structure, and contact distribution. Element-wise comparisons revealed that, when restricting to mixing among adults and seniors, the number of contacts engaged by adults and seniors (both within and across these two age groups) was similar in the two matrix types (Fig. 4a). In contrast, substantial discrepancies were observed for contacts involving younger individuals — especially in within-group contacts during school-open periods (Fig. 4b, Figs. S10-S11). Analysis of age assortativity showed that empirical matrices exhibited weaker within-group mixing compared to synthetic matrices, particularly for adolescents, adults, and seniors. In contrast, children had more assortative contacts in the empirical data (Fig. 4c,d). In the empirical matrices, young individuals contributed to 50% of total connectivity, twice as in the synthetic contact matrices (Fig. S12b). This suggests that empirical matrices featured more intergenerational mixing and a stronger influence of younger age groups on overall contact patterns, compared to synthetic matrices. Cosine similarity — a metric defined between 0 and 1 assessing the similarity in mixing structure, independent of overall contact volume — was higher during periods of school closure, indicating a larger agreement in age-specific mixing patterns between the two matrix sources (Fig. S12a). When schools were open, particularly in March 2021, similarity dropped, reflecting greater deviations in mixing among young age groups. Notably, cosine similarity between empirical and synthetic matrices was consistently higher than that between synthetic matrices and random mixing (Fig. S12a), reinforcing that synthetic matrices retain meaningful age structure beyond proportional mixing.

Figure 2. Illustrations of contact matrices.

Figure S10. Element-by-element absolute difference between empirical and synthetic matrices.

Figure S11. Element-by-element contact ratio between empirical and synthetic matrices.

Figure 4c-d. Fraction of within-group contacts. These panels have been added to Fig. 4 in the revised manuscript.

It seems that contacts can be stratified by type in both data sources. For example, ages [0-10] and [11-18] had the largest discrepancies between empirical and synthetic matrices. While this was acknowledged, were differences driven by within or between age-group contacts, and for which types/locations of contacts?

For what concerns within vs between age-group contacts, element-by-element comparison was shown in (Fig. 4a,b). We have now highlighted in the scatter plot which matrix elements represent within-group contacts and which elements correspond to between-group contacts. Results support that differences between empirical and synthetic were mostly driven by discrepancies in within-group contacts, as highlighted in the comment above. **We have updated the Fig. 4a,b in the revised manuscript**, using void and filled symbols to distinguish between the within- and between-group contacts, and different symbols to distinguish the age group establishing the contacts.

Figure 4a-b. Element-by-element comparison in the empirical and synthetic matrices.

Additionally, as stated above, we also provided illustrations of a difference matrix, showing the absolute difference and ratio in each element of the contact matrices, as suggested by the reviewer in one of the minor points below. We can see that the largest absolute differences are observed in children-children contacts and adolescents-adolescents contacts when schools are open. **These figures have been added in the Supplementary Material (Fig. S10, S11).**

Regarding contact types (physical vs. non-physical) and contact locations, we did not compare these aspects across the synthetic and SocialCov matrices because our transmission model does not differentiate transmissibility by contact type or setting. In other words, the model assumes that all reported contacts contribute equally to transmission, regardless of their nature or context. As a result, only the total number of contacts between age groups was required to parameterize the force of infection, and contact disaggregation by type or location was not needed for inference or model fitting. **We now mention this modeling assumption more explicitly in the revised Methods and Discussion sections.** Incorporating heterogeneity by contact type or location would require both changes to the model structure and additional assumptions about differential transmission risks — an interesting direction for future work where relevant data are available. We report here the changes made to the manuscript.

Methods (lines 486-487)

We did not differentiate transmissibility by contact type or setting, assuming all contacts contribute equally to transmission.

Discussion (lines 350-352)

Moreover, lacking data to support differential transmission risks, our model assumes that all contacts contribute equally to transmission, regardless of their nature (physical or non-physical) or setting.

The four comparison metrics (Suppl 5) are useful, but these are discussed too briefly in the results and they remain rather abstract (e.g. what is a "low" measure of cosine similarity? Maybe a random mixing matrix can serve as a reference.)

As reported above, we have extended the Results section where we discuss the four metrics used to compare the contact matrices, together with the additional metric presented above, i.e. fraction of contacts established within the same age group.

In addition, in the revised version of our manuscript we computed the cosine similarity between the synthetic matrix and TEST2 (random mixing contact matrix). Results are illustrated below (and they have been in the Supplementary Information as Fig. S12a). We found that synthetic matrices are

more similar to empirical matrices than random mixing matrices (lines 152-154 in the revised manuscript). As already stated in the original manuscript, the synthetic and SocialCov matrices were more similar during periods of school closure rather than periods where schools were open (lines 148-152).

Figure 12a. Cosine similarity of the SocialCov matrices or random mixing matrices compared to synthetic matrices. The comparison with random mixing has been added in the revised manuscript.

- Relatedly, only Figure 1a (in main text or supplement) depicts a complete mixing matrix (all 4 x 4 age group combinations), but this reflects the pre-pandemic matrix. Give the aims of comparing contact matrices, I was expecting to see more illustrations of complete mixing matrices, so that changes to mixing *patterns* can be examined (not just total contacts as in Figure 2e-f).

As reported in the above reply, **we now provide illustrations of the complete mixing matrices in the revised version of the manuscript. The figure has been included in the Main Text as Fig. 2.**

Medium issues

- Overall, I found the methods-last structure of the paper (imposed by the journal I think) inefficient and confusing. The results section provides only a trickle of methods information, which distracted me from the results themselves with questions I had about the methods. However, reading the methods before the results section, ideas like the correcting factor were assumed to be already introduced. Still more methods are relegated to the supplement, so I am never quite sure where to look for a particular methods detail. An outline of the experiments is needed.

The structure of the paper with methods presented at the end is indeed imposed by the journal. For such a structure, it is common practice to include in the result section a summary information about the methods, to introduce the terms used and to allow the reader to delve into the results without reading the whole methods section beforehand. **We have now revised the Methods section to improve clarity.** Moreover, in our original submission, we had made explicit reference to Tables or specific sections of the Supplementary Material in relevant parts of the methods, to guide the reader to where to find specific methodological details. We have now included additional references to better guide the reader. We also expanded the summary of the methods at the start of each Results section.

- A related issue is that the research questions / aims are not stated explicitly anywhere. Also, the last paragraph of the introduction includes: "Although mobility-based synthetic contact matrices ... their accuracy relative to empirical contact matrices remains unexplored" which suggests that the empirical

matrices will be taken as a "gold standard" in comparisons, but this is not really how the results are framed. Similarly: "evaluating the effectiveness of these two approaches in modeling the pandemic dynamics" should define "effectiveness" and "pandemic dynamics" more concretely.

Empirical matrices are widely considered the conventional approach for estimating contact patterns in infectious disease modeling. However, in our study, they are not treated as a "gold standard," as they come with notable limitations — most importantly, their low temporal resolution — which hinder their suitability as a definitive benchmark for evaluating time-varying contact structures.

This is now made clear in the revised manuscript, together with the following two research questions, explicitly stated in the revised Introduction:

- 1) Compare the time-varying contact matrices generated using mobility-informed synthetic methods with empirical matrices derived from repeated contact surveys, focusing on differences in contact volumes and mixing patterns across age groups.
- 2) Assess the capacity of each matrix to support epidemic modeling of COVID-19 and to reproduce observed hospitalizations, infections, and age-specific trends. To this end, each matrix is embedded in an age-structured transmission model calibrated on hospitalization data and informed by real-world information on viral variants, vaccination, and immunity.

Our aims are now more explicitly presented in the abstract as well.

We report here the changes made to the Introduction (lines 78-90)

Although mobility-based synthetic contact matrices offer a promising alternative for real-time modeling, their comparative performance relative to empirical contact matrices remains unexplored. Meanwhile, traditional empirical contact matrices - though widely used - are limited by infrequent updates, which constrains their applicability for dynamic modeling. This study addresses two main objectives. First, we compared contact patterns derived from mobility-informed synthetic matrices [37] and from empirical matrices computed from social contact surveys [29,30], focusing on age-specific contact volumes and mixing patterns over time. Second, we assessed the capacity of each matrix type to support epidemic modeling of COVID-19 in France by embedding them in an age-stratified transmission model calibrated on hospitalization data and informed by real-world information on viral variants, vaccination, and immunity. This framework enables us to evaluate how well each matrix captures disease-relevant contacts over time to reproduce observed indicators, including hospitalizations and age-specific infections. By using the COVID-19 pandemic in France as a case study, we provide insight into how different matrix sources can inform real-time transmission modeling for future outbreak response.

- It should be clarified in the methods how contacts were reported by individuals < 18 in waves 2-7 of SocialCov. In Suppl 4.3 [310], this is indirectly suggested, but only the discussion acknowledges that parents might have reported for children. This is notable since ages [0-10] had the largest discrepancies between synthetic and empiric matrices.

We thank the reviewer for the observation, which was also raised by reviewer #2. For individuals under the age of 18, contacts were reported by their parents. This information was indeed mentioned in the Discussion but not mentioned in the Methods section. **We have now added the following sentence to the Methods** (lines 440-442):

If participants declared living with minors (i.e. individuals under the age of 18), they were invited to complete a second specific questionnaire for one of them, in his/her presence.

This element was already included in the Discussion in our original submission, when discussing the discrepancies in epidemic outputs for children (lines 317-321).

Second, SocialCov surveys relied on aggregated contact reporting by age group rather than individual contact listings, which may have facilitated reporting – especially for parents reporting for their children – at the cost of accuracy.

- Although acknowledged in the limitations, the *removal* of physical contacts (rather than replacing them with non-physical contacts) does not seem plausible. Since most covid-19 models make no distinction between these types, I think it is better to either ignore the level of avoidance (q at [273] in the Supplement) and treat all contacts equally, or use distinct rates of transmission (β) for physical vs non-physical contacts in order to use q .

We thank the reviewer for this comment. Since the survey data on physical contact avoidance did not specify whether avoided physical contacts were replaced by non-physical ones, the model required a simplifying assumption. We chose to remove the avoided physical contacts, acknowledging this as a limiting — but clearly defined — lower-bound scenario. We did not assume replacement with non-physical contacts because we lack data to support such a transition, and because no evidence exists to parameterize different transmission rates for physical vs non-physical contacts. For this reason, and in line with most COVID-19 models, we treated all contacts as equally transmissible.

To address the reviewer's suggestion, we conducted a sensitivity analysis (**TEST4**) in which we ignored avoidance and retained all physical contacts (i.e., assuming either no avoidance or full replacement with equally transmissible non-physical contacts). As expected, this increased the average number of contacts across age groups (**Fig. S28**). Results from TEST4 were then compared with those from our main analysis. We found that the model parameterized with the TEST4 matrix required a lower correcting factor, closer to that estimated for the SocialCov matrices (**Fig. S29**). However, while it improved the fit of adolescent infections relative to serological data, it worsened the fit for hospitalizations in seniors and infections in children (**Fig. S29-S30**). Both the MAE and AIC indicated that TEST4 model performed better than the empirical (SocialCov) model, but still worse than the original synthetic model using contact avoidance (**Fig. S29, Table S16**). Results did not change when fitting age-stratified hospitalizations (**Fig. R5, Rebuttal Appendix**).

Overall, this analysis suggests that our original assumption provides a conservative yet effective approach, and confirms the robustness of our conclusions to different treatments of contact avoidance. **These results have been added to the Supplementary Information (Section 9.6) and discussed in the main text.** We report here the changes made to the manuscript.

Methods (lines 421-423, lines 562-563)

We assumed that physical contacts were removed rather than replaced with non-physical contacts, thus representing a lower-bound scenario; this assumption was tested in a sensitivity analysis (see section Sensitivity Analyses below).

[...]

Finally, we tested an alternative parameterization of the synthetic matrices without assuming contact removal due to avoidance of physical contacts.

Results (lines 228-239)

*We first examined the role of physical contact avoidance in shaping the synthetic matrices. In the main analysis, we reduced total contacts based on self-reported avoidance of physical contact, as a proxy for reduced transmission risk (**Section 4.2.4** in the Supplementary Information). However, this assumes that avoided physical contacts were entirely suppressed, whereas some may have continued in a non-physical form (e.g., masked or distanced), likely leading to an overestimation of the contact reduction. To test the impact of this conservative assumption, we generated a set of weekly matrices without applying this reduction. As expected, the average number of contacts across age groups increased (**Fig. S28**). Both the AIC and the mean absolute error indicated that this model performed better than the empirical matrix model, but still worse than the original synthetic matrix*

model incorporating contact avoidance (Table S16, Fig. S29). While this change improved the agreement with serological data for infections in adolescents, it simultaneously worsened predictions of senior hospitalizations and children infections, underscoring age-specific trade-offs (Fig. S30).

Discussion (lines 312-314)

However, increasing child contact numbers in the synthetic matrices—by assuming that avoided physical contacts were replaced with non-physical ones rather than removed—did not improve model fit or infections in children.

Figure S28. Comparison of synthetic matrices and matrices in TEST4.

Table S16. Goodness of fit of the model, using the synthetic, empirical, pre-pandemic matrices, or matrices in TEST4.

Model	AIC
Synthetic	27624
TEST4	29329
Empirical	34356
Pre-pandemic	34690

Figure S29. Sensitivity of model fit when using the synthetic/empirical/pre-pandemic matrices, or matrices in TEST4.

Figure S30. Sensitivity of model validation when using the synthetic/empirical/pre-pandemic matrices, or matrices in TEST4.

- Figure S1 is very helpful, but analogous diagrams for the other strata (especially vaccination, prior infection, waning, and VoC) are needed too. These can be 1-2 separate subfigures (not drawing all compartments / joint strata in one diagram).

We thank the reviewer for mentioning this. In the revised manuscript, we have now included additional diagrams illustrating the strata for VoCs, prior infections, vaccination and waning (**Fig. S2**).

Minor Issues

Main

- [41] Somewhere in this paragraph, a more precise definition of "contact matrices" should be given, e.g. "in which matrix elements reflect the average number of contacts per day formed by individuals in one group with another."

Done. We included the following sentence (lines 43-45):

This aspect can be incorporated into transmission models through a contact matrix, where each element represents the average number of daily contacts that individuals in one age group have with individuals in another age group.

- [58] I think "convenient" should be "convenience" here?

Corrected. We thank the reviewer for spotting the typo.

- [72] The term "contact layer" here and throughout might be confusing. Why not simply "contact types"?

We originally used the term "contact types" to refer to physical/non-physical contacts, not to contacts by location. To avoid confusion, we have now removed the term "contact layer" in the revised manuscript. We refer to contacts by location using the term "contact setting".

- [73] "These data streams enabled to produce weekly ..." is not grammatically correct - add "enabled *us* to"?

Done.

- [101] "to reproduce age and gender distribution in France" same - add "reproduce *the* age"?

Done.

- [102] Consider adding "*average* daily contacts" for precision.

Done.

- [117] The meaning of "normalized contact matrices, which disregard contact intensity" is not clear - is this normalization redundant with the definition of cosine similarity?

We apologize for the confusion. The cosine similarity is a measure with the property of being invariant by a global factor. Therefore, we did not apply an additional normalization to the contact matrices, as it

would be redundant with the definition of cosine similarity and would provide the same result. The term “normalized” was used solely to help the reader interpret the cosine similarity measure, but it created confusion instead.

We have now removed the term “normalized” and **have revised the manuscript as follows** (lines 148-154):

Cosine similarity — a metric defined between 0 and 1 assessing the similarity in mixing structure, independent of overall contact volume — was higher during periods of school closure, indicating a larger agreement in age-specific mixing patterns between the two matrix sources (Fig. 12a). When schools were open, particularly in March 2021, similarity dropped, reflecting greater deviations in mixing among young age groups. Notably, cosine similarity between empirical and synthetic matrices was consistently higher than that between synthetic matrices and random mixing (Fig. S12a), reinforcing that synthetic matrices retain meaningful age structure beyond proportional mixing.

- [138] As suggested above, another comparator could be pure random (so-called proportionate) mixing matrix -- which could be used in both aims (i) and (ii).

As noted above, **we carried out a sensitivity analysis using random mixing (TEST2)**. Results have been reported in the Supplementary Information (**Section 9.5**). The value of the cosine similarity between synthetic and random mixing has been added to **Fig. 12a** of the revised manuscript.

- [145] Why is the IQR reported, rather than 95% range? I would think the latter is more useful, but maybe both could be given like: median (Q2.5 [Q25, Q75] Q97.5)

In the revised version of the manuscript, we now report both the IQR and the 95% range.

- [147] "suggesting that small adjustments were needed" as noted above, I do not agree these are small, especially those alpha outside IQR ...

We have revised the manuscript by writing the following (Results, lines 185-196):

The median correcting factor over time was estimated at 1.16 (0.6 [0.87, 1.33] 1.81, indicating Q2.5 [Q25, Q75] Q97.5 quantiles) for the model using synthetic matrices and 0.79 (0.46 [0.64, 1.08] 1.36) for the one using empirical matrices (Fig. 6a). In total, 64% and 52% of the correcting factors in the synthetic and empirical matrix models, respectively, fell within the range 0.75–1.25, indicating that modest adjustments were sufficient in most weeks across both cases. [...] By contrast, the model using a static pre-pandemic contact matrix required consistently larger adjustments (median 0.51, quantiles 0.26 [0.42, 0.68] 0.9), reflecting the need to substantially scale down contact volume to match observed dynamics during the pandemic (Fig. 6a).

- [163] "confirmed overall these findings" grammar - maybe **"generally"** confirmed these findings"

Corrected.

- [173] "strongly valuable" grammar - maybe **"highly"** valuable"

Corrected.

- [197] "survey frequency of surveys" typo?

Corrected.

- [233] I would not fully agree that "However, these limitations critically depend on the mobility metrics chosen" since masking can erode the correlation with effectively no impact on mobility metrics.

We agree that masking can erode the correlation between mobility and transmission. This aspect was explicitly acknowledged in the previous sentence. However, our statement refers specifically to the limitations of the simplifying assumptions or choices considered in the cited studies. The choice of mobility metric — for instance, tracking flows from location A to B that may have little bearing on actual social mixing in A — and assumptions such as linearity in the relationship with transmission critically affect the utility of mobility data. Our findings suggest that, even in the presence of preventive measures like masking, a *non-linear* relationship between workplace presence and transmission can still offer valuable insights for epidemic modeling and response, when appropriately integrated into a structured modeling framework.

We revised the text in the Discussion as follows, to improve clarity (lines 332-339)

These findings support the use of mobility data as a valuable proxy for time-varying contacts in workplace settings during a pandemic. While earlier studies showed strong correlation between mobility and COVID-19 spread in the early pandemic phase[49–51], this relationship weakened over time as behavioral adaptations and masking became more prominent[52–55], underscoring the limitations of inappropriate proxies or relying on simplistic linear associations between mobility and transmission[53]. Still, our findings demonstrate that workplace attendance data, when carefully selected as a setting-specific proxy and integrated non-linearly into a synthetic framework, remain a meaningful and robust input for capturing transmission dynamics.

- [245] It's not clear what is meant by "optimal methods for extending empirical contact matrices".

We thank the reviewer for raising this point. As detailed in the Methods: "The contact matrices available from 7 distinct surveys were extended beyond the survey period in order to cover the time span of the model, as pictured in **Fig. 5b**." In the Discussion, we mention that other possible extensions could be studied.

To avoid confusion, we removed the word "optimal" from the Discussion section, and we used the following (lines 281-282):

Alternative methods for temporally extending empirical matrices may hold promise, but they are often context-specific and remain an open area for future research.

- [262] The subject of "underestimate transmission risks" is not clear here: isn't transmission risk overestimated by ignoring repeat contacts?

We apologize for the lack of clarity. As explained in Ref. #53 (quoting literally) "accounting for the frequency of contacts impacts the number of new, distinct, contacts, revealing a lower total count than a naive approach, where contact repetition is neglected. As a consequence, failing to account for the repetition of contacts can result in an underestimation of the transmission probability given a contact". Therefore, transmission risk is underestimated when ignoring repeated contacts, and this is indeed what we meant in line 262 in the Discussion.

To be more clear, we have revised the wording as follows (lines 346-347):

Our model assumed distinct daily contacts, thus ignoring the repetition of contacts and potentially underestimating transmission risks [57].

- [274] The word "robust" here is too strong.

We replaced the word "robust" with "reliable".

- [333] The overall value of these correlations with Normalcy Index and Stringency Index with respect to the aims are not clear to me.

The correlations with the Normalcy and Stringency Indices were included to provide contextual validation of the temporal trends in our synthetic contact matrices. These indices summarize broad societal and policy shifts during the pandemic, and their alignment with the trends in contact patterns supports the plausibility of the behavioral signals integrated into our matrices. **We have now clarified this point in the Results section** (lines 120-126) by adding the following:

To contextualize these dynamics, we examined correlations with the Normalcy Index, which captures behavioral adaptation, and with the Stringency Index, which quantifies the intensity of social distancing measures. The synthetic contacts time series was highly correlated with both indices (Normalcy Index: Pearson's coefficient = 0.84, p-value < 10⁻¹⁹⁹, Fig. 3c, S13b; Stringency Index: Pearson's coefficient = 0.82, p-value < 10⁻¹⁹⁶, Fig. S13b), supporting the plausibility of the behavioral signals integrated into the synthetic matrices. This correlation reflects concurrent behavioral shifts rather than overlap in data inputs.

- [340] As far as I can tell, the only citation for the 7 SocialCov surveys is #25 in Eurosurveillance, which only comments on the first survey. This is notable since later surveys included all ages, while the first was restricted to 18+. I noticed when looking for information about how responses for young children/babies were obtained.

We thank the reviewer for noticing this. We did not include an additional citation to the SocialCov surveys because, at the time of submission of this manuscript, the paper detailing the SocialCov study was not published yet. The paper has now been published (Soussand et al., BMC Infectious Diseases 2025, <https://doi.org/10.1186/s12879-025-10611-4>). **We have now included this paper in the list of references in the revised version of our manuscript.**

- [365-372] The implications of assortativity for reproduction number & herd effects might be stated briefly somewhere here.

We thank the reviewer for the suggestion. We added the following sentence in the Methods (lines 470-471):

More assortative contacts have implications on disease spread, increasing the epidemic risk and leading to lower herd immunity thresholds (Kiss et al. 2008, Elshaba et al. 2021).

- [375-408] I just want to reiterate that the level of model complexity is impressive.

We appreciate the reviewer's comment. We refer to this comment in a previous reply, when discussing the trade-off between model complexity and associated data needs, on one hand, and the use of a larger number of age groups, on the other.

- [414] Perhaps Table S7 could be explicitly mentioned here regarding time windows.

Done.

- [436] It would help to explain the rationale / strengths for each performance metric. For example, AIC would "punish" 20% error in predicting 50 or 500 hospitalizations more equally, whereas absolute error would "punish" 20% error on 500 hospitalizations ~10 times more than on 50.

This was already done in the Supplementary Information, and it is now briefly recalled here.

AIC and MAE quantify different aspects of model accuracy. AIC is computed from the likelihood, which reflects how well the model explains the overall data distribution. A model with the better AIC might be giving a better fit to the data's distribution even if the absolute deviations are higher, while a model with lower MAE might be doing well in minimizing large deviations, but it may not reflect the distribution of the data as accurately.

- [437] Relatedly, is there any benefit to using AIC over simple (log-)likelihood? I think k is the same across approaches, but this is never stated explicitly.

The parameter k (i.e., the number of time-discrete correcting factors) is the same across approaches, except for the model informed with empirical matrices, for which we estimated an additional correcting factor by splitting the time-window $w_{52, 2020} - w_{2, 2021}$ into two windows. This detail was already included in the original submission (see **Table S8**). For this reason, we used AIC to compare the models rather than the simple likelihood.

- [462] I commend the authors on making their code available via GitHub.

We thank the reviewer for their appreciation regarding the availability of our code.

- [Figure 1] Ideally, the matrix figure (a) could use a perceptually uniform colormap, and the proportion figure (b) use a different, qualitative map. The hashing for non/physical is very nice. Perhaps (d) could extend the range from 30-100 to 0-100% to minimize misinterpretation.

We thank the reviewer for the suggestions. In panel (a), we decided to keep a non-uniform color map to highlight differences also between cells with very low number of contacts. In panel (b), we replaced the original palette with a different, qualitative color map (see below). We also incorporated the suggestion for panel (d) where we extended the y-axis to 0-100%.

Figure 1a,b. Age-stratified contact matrix, and age-specific proportion of contacts by setting and type.

- [Figure 2] Overall these subfigures are excellent, and nicely communicate a lot of information.

Thank you — we appreciate the positive feedback on this figure (now numbered as **Fig. 3** in the revised manuscript).

- [Figure 3] (a-b) Instead of these subfigures, which omit between-group mixing and have indistinguishable data points, perhaps an image like 1a can show the difference matrix ($M1_{ij} - M2_{ij}$) so readers can visually see where differences are largest; I think that is the goal here?

We thank the reviewer for the comment about this figure (now numbered as **Fig. 4** in the revised manuscript). In panels (a-b), we were not omitting between group mixing, as we were plotting all matrix elements (both diagonal and off-diagonal elements). However, data points were not distinguishable. We have now provided a version of the figure where diagonal and off-diagonal elements are distinguished using filled and void symbols (as mentioned in a reply above, **Fig. 4a,b**). Moreover, we included illustrations of matrices showing the element-by-element absolute difference ($M_{ij}(\text{empirical}) - M_{ij}(\text{synthetic})$) and contact ratio ($M_{ij}(\text{empirical}) / M_{ij}(\text{synthetic})$) in the Supplementary Information (**Fig. S10, S11**).

-[Figure 3] Also, perhaps (d) can use the same x-axis as (c), as I'm not sure what the current axis (% contacts < 19) is communicating?

Panel (d) of Fig. 3 (now moved to the Supplementary Information as **Fig. S12b**) was designed in order to compare mixing patterns with two different metrics: assortativity vs proportion of contacts produced by young individuals. **We have now expanded the paragraph of the Results section which describes these findings.** We added the following sentences.

Supplementary Information (section 8.1)

*In **Fig. S12**, we compare the matrices using different metrics [...]. Compared to the pre-pandemic contact matrix, synthetic matrices were slightly less assortative, while maintaining a similar proportion of contacts established by young individuals (around 25%, **Fig. S12b**). In contrast, empirical matrices showed larger differences with the pre-pandemic and synthetic contact matrices, with lower overall assortativity and a higher share of contacts generated by individuals under 19 years of age (about 50%, **Fig. S12b**).*

Results (lines 144-146)

*Moreover, in the empirical matrices, young individuals (i.e. children and adolescents) contributed to 50% of total connectivity, twice as in the synthetic contact matrices (**Fig. S12b**).*

- [Figure 4] Is excellent.

We greatly appreciate the positive feedback of the reviewer about this figure (now numbered as **Fig. 5** in the revised manuscript).

- [Figure 5] (and S4, S9): Perhaps alpha should be plotted in log2 y-scale, since it is a multiplicative factor? Also, is it possible to scale densities so their maximum width is consistent, to improve clarity (especially Figure S4)?

We thank the reviewer for the suggestions about Fig. 5 (now numbered as **Fig. 6** in the revised manuscript). In **Fig. 6a**, we now plot the correcting factor in log2 y-scale. We also applied this change to all plots displaying the correcting factor, including the ones involving the TESTS. Moreover, in **Fig. S5**, we scaled the densities so that their maximum width is consistent, to improve clarity.

Figure 6a. Distribution of the correcting factor in log2 scale.

Supplement

- [65] Perhaps "dimensions" is a better term than "layers" to describe independent population stratifications?

Corrected. We now use the term "strata".

- [76] Table S1 should clarify that the relative infectiousness with mild and severe symptoms (I_{ms} , I_{ss}) are the same.

Done.

- [78-105] To improve ease of use, this section (Suppl 2) could have the information added to Table S1 or summarized in a new Table S2?

Done. We summarized all variant-specific parameters of Section 2 in Table S2.

- [86] I cannot find the base probability of transmission (β_{pre-LD}) anywhere. I understand this is fitted, but can the posterior mean (95%) be reported?

We apologize for the lack of information. We have now provided the estimate of the baseline transmission rate per contact together with the 95% confidence interval in the Supplementary Information (Section 6).

- [178] The elements in matrix C_{ij} are described as "per-capita rates of contact between age-group i and j ". This key definition is not very precise, especially to distinguish C_{ij} from M_{ij} . My understanding is that C_{ij} is equivalent to the "intrinsic connectivity matrix" in Arregui et al (2018) - perhaps this terminology can be used or added in parentheses for readers already familiar? Though, it does not add much precision. Another definition of C_{ij} is the expected number of contacts formed by a group- i individual with group- j individuals, if 100% of the population was in group- j , or equivalently, C_{ij} / N is the expected number of contacts formed by a group- i individual with group- j individuals per individual in group- j .

We thank the author for the question. We confirm that the definition of the matrix C_{ij} is indeed equivalent to the intrinsic connectivity matrix described by Arregui et al. (2018). We included the following in the revised text (section 4.2 of the Supplementary Information):

The definition of the matrix C_{ij} is equivalent to the “intrinsic connectivity matrix” introduced in Arregui et al. (2018). In other words, C_{ij} is the expected number of contacts formed by a group- i individual with group- j individuals, if 100% of the population was in group- j , or equivalently, C_{ij} / N is the expected number of contacts formed by a group- i individual with group- j individuals per individual in group- j .

- [204-221] While I appreciate the authors' consideration that not all declared work contacts will be symmetrically reported (e.g. a bus driver and passenger may report one another as "work" and "travel" contacts, respectively), I would think that most public-facing occupations would be infeasible for remote-work, and that occupations where remote-work is feasible would include mainly "symmetrically reported" (work-work) contacts. So, I don't agree that the two situations described here are equally likely, hence the average: $(1-a) + (1-a)^2 / 2$; and I would suggest using the $(1-a)^2$ approach throughout, unless the authors can provide some examples of public-facing occupations where remote-work is feasible?

We thank the reviewer for raising this point and agree that the two scenarios are unlikely to occur with equal probability. However, in the absence of empirical data quantifying the frequency or nature of asymmetrically versus symmetrically reported work contacts — particularly for public-facing occupations — we opted for a simple average between the two. This choice reflects a pragmatic compromise and is consistent with the approach we adopted for real-time modeling during the pandemic, which we deliberately maintained in this retrospective assessment to preserve methodological comparability.

We also note that understanding the extent to which setting-specific contacts scale with individual presence — and how this varies by occupation type or contact symmetry — is itself a complex modeling and data challenge. This question is the focus of ongoing work that goes beyond the scope of the present study, which aims to assess the performance of mobility-informed synthetic matrices against empirical alternatives rather than optimize their construction.

- [322-340] Suppl 4.4: The direct removal of contacts among those isolating essentially implies a density-dependent effect, since the isolating contacts are not "replaced" among everyone else. A frequency-dependent approach would add a lot of complexity to calculations with probably negligible difference in transmission because infection prevalence is never very high, so the current approach is fine, but it may be worth spelling out the implied assumption.

We thank the reviewer for pointing this out. **We have revised the text by explicitly stating the implied assumption and its relation with the density-dependent effect** (section 4.4 of the Supplementary Information).

We note that the choice adopted to model the effect of isolation assumes a density-dependent effect, as the contacts removed due to isolation are not replaced with other contacts among non-isolated individuals. An alternative frequency-dependent would require additional complexity, but it is however expected to have a negligible effect as prevalence in the I_{ss} compartment is limited in size.

- [365-394] Suppl 5.4: It may be more clear to simply omit the continuous domain functions and integrals, and focus on the discrete implementation via sums.

We thank the reviewer for the suggestion. The definition of assortativity is based on Farrington et al. (2009), which provides only the continuous formulation. Building on their definition, we derived the discrete version of assortativity. For completeness, we believe it is helpful to include both the continuous and discrete formulations, allowing the reader to understand the derivation more easily.

- [398] The force of infection equation shown here is extremely simplified compared to the modifiers described (and the GitHub code), such as infectiousness, susceptibility, etc. The true equation spelling out the various factors involved would be much clearer. Also, since the denominator "N" differs from the usual " N_{ij} ", it may be worth reminding readers how this derives from the definition of " C_{ij} ".

We thank the reviewer for this comment. **We have now reported a full form of the force of infection**, which accounts for infectiousness, susceptibility, disease stage, variants and vaccination status. Furthermore, we have made explicit that, given our definition of C_{ij} , the expression of the force of infection is equivalent to the usual form with the denominator N_{ij} . These revisions have been included in section 6 of the Supplementary Information.

- [492] Figure S5d should likely log-transform the y-axis to improve the visualization of differences, and because I think 50% relative difference is "equally bad" as 200% ?

We thank the reviewer for the suggestion. We kept the linear scale on the y-axis, as a log-transform would not allow us to display negative values. However, we acknowledge that the relative variation hides the fact that the sizes of total hospitalizations in the different age groups are very different, e.g. hospitalizations in seniors and adults are one order of magnitude larger than hospitalizations in children and adolescents. Therefore, a 10% relative variation in adults or seniors corresponds to a 100% relative difference in children. **To highlight this, we have now included an additional plot displaying the absolute difference between the hospitalizations predicted by the model and the observed data, by age group. This plot has been added in the Supplementary Information as Fig. S8b, and also reported in the main text as Fig. 5d.** When reporting results of the sensitivity analyses (tests with variations in the synthetic matrices, and age-stratified likelihood), we now show the plot with the absolute difference rather than the relative variation.

Fig. S8. Comparison of model predictions with hospital admission data by age class. Fig. S8b is also reported in the main text as Fig. 5d.

Reviewer #1 (Remarks on code availability):

I have only briefly examined the code on GitHub to verify questions I had. It seems reproducible but would need to be fully tested.

We thank the reviewer for checking the code to verify their questions.

REVIEWER #2

Overall, this is an interesting and well-written manuscript based on a thorough analysis, comparing the use of weekly synthetic transmission matrices, building on an elaborate method for exploring mobility data with matrices from online social survey waves to model the evolution of the pandemic in France. The study adds novelty and deserves publication. However, some clarification on the methods is warranted. As pointed out by the authors, during a crisis where situational awareness are needed for policy recommendations, their method is easily automatized and can readily be used, compared to social survey studies that require data washing and more elaborate measurement.

We thank the reviewer for the positive assessment, and for their suggestions that allowed us to improve the clarity of our methodology. We provide our answers here below.

Main comments

The “empirical matrices” are based on data from the SocialCov survey, with a reference to Bosetti et al. Eurosurveillance 26 (2021). However, this reference only addresses the primary wave exclusively conducted among adults 18+ years. Hence, it is unclear how data was collected among children and adolescents in the later waves. Do they involve parents reporting on behalf of their children? Are adolescents included in the survey? This is important information and should be properly documented. A sentence in the discussion hints that parents were answering on behalf of their children.

We thank the reviewer for this comment, which was also raised by reviewer #1. We did not include an additional citation to the SocialCov surveys in the original version of the manuscript because, at the time of submission, the paper detailing the SocialCov study with multiple waves was not published yet. The paper has now been published (Soussand et al., BMC Infectious Diseases 2025, <https://doi.org/10.1186/s12879-025-10611-4>). **It has now been included in the revised version of our manuscript.**

For individuals under the age of 18, contacts were reported by their parents. This information was indeed mentioned in the Discussion but not mentioned in the Methods section. We have now added the following sentence to the Methods (lines 440-442):

If participants declared living with minors (i.e. individuals under the age of 18), they were invited to complete a second specific questionnaire for one of them, in his/her presence.

The methods for calculation of a typical weekday is not specified. Presumably, the SocialCov survey includes information collected on different weekdays. How is a “typical” weekday calculated? There are likely differences between contacts on Mondays vs. mid-week.

Contacts from the pre-pandemic survey and for SocialCov were filtered on regular weekdays, defined as Monday-Friday excluding holidays. This is now added to the Methods (lines 391-393, lines 451-453).

[Synthetic] We considered the contact matrix estimated for a regular weekday, defined as Monday-Friday excluding holidays.

[SocialCov] We used contacts reported on Monday-Friday excluding holidays, to allow comparison with the synthetic contact matrices that were built based on a regular weekday pre-pandemic matrix.

Although we acknowledge that contacts may vary from Monday to Friday, averaging contacts across weekdays is common practice in studies analyzing social contact data, as the most relevant differences are observed between weekdays and weekends rather than within weekdays.

It is not clear to this reader what specific contacts were used to construct the “empirical matrices”. It is stated that both physical and non-physical contacts were collected –if the total number of contacts were used, it should be stated.

We thank the reviewer for the opportunity to clarify. We indeed used the total number of contacts in the SocialCov matrices, including both physical and non-physical contacts. **We have now included this detail in the Methods** (lines 450-451).

Total contacts (physical and non-physical) for each individual were truncated at 50 to reduce the impact of outliers.

The imputation process (for the first wave of data collection) is not specified, and there are no specifics on the sampling procedure when reweighing the data. Were the matrices bootstrapped to account for sampling variance?

Yes, the matrices were bootstrapped to account for sampling variance and to create a population representative of France’s age and gender distribution. Sampling with replacement was performed from the pool of participants. For each campaign, we used data from the Institut national de la statistique et des études économiques (INSEE – National Institute for Statistics and Economic Studies) to estimate the appropriate weights for each gender and age group. **This is now added in the revised Methods section, together with a reference to the study fully presenting the computation of the SocialCov matrices** (lines 512-516).

Since the survey sample was not representative of the French population, synthetic populations were generated for each campaign to better reflect the age and gender distribution in France, using sampling with replacement from the SocialCov participant pool; the matrices were bootstrapped to account for sampling variance [30].

The paper does not consider the French CoMix data because the survey was conducted over only 5 months (late 2020 to spring 2021). However, the CoMix survey design allows for a more detailed description of contacts in children. I believe it would be informative and of general interest to include this data and compare average connectivity, cosine similarity and the proportion of young connectivity, even if the time period is restricted.

We thank the reviewer for the suggestion. As explained in the manuscript, we did not include this source of empirical matrices into our transmission model due to the limited temporal resolution of the CoMix surveys carried out in France (only spanning from late 2020 to April 2021). However, following the reviewer’s suggestion, we carried out the comparison in terms of contact patterns.

The results of the comparison have been included in the revised manuscript. We report here changes made to the manuscript.

Methods (lines 472-476)

We further compared the synthetic and SocialCov matrices with two empirical matrices from the CoMix study (Section 8.3), which was conducted in France between December 2020 and April 2021. CoMix provided complementary age-stratified data, but its limited temporal coverage and inclusion of minors in only two of seven waves restricted this comparison to two periods.

Results (lines 155-159)

We also compared synthetic and SocialCov matrices with empirical contact data from the CoMix study [42], conducted in France between December 2020 and April 2021. Due to its limited temporal coverage and inclusion of minors in only two waves, CoMix was not used in the transmission model; however, its matrices showed lower contact rates than SocialCov and age assortativity similar to synthetic matrices (Fig. S15-S16).

Discussion (lines 322-325)

Previous work has shown that aggregated formats tend to report higher contact numbers compared to detailed individual listings [27]. This is further supported by our comparison with CoMix matrices, which reported substantially fewer contacts for children compared to SocialCov, while their assortativity was more similar to that of the synthetic matrices.

We report the detailed results of the comparison below. **They have also been included in the revised Supplementary Information (Section 8.3).**

The CoMix study in France carried out 7 survey waves for adults (from A1 to A7), and 2 survey waves for children, C1 and C2 (below 18 years of age). The calendar of the survey waves is illustrated in Fig. S14 and Table S11.

Figure S14. Periods of data collection for SociaCov and CoMix surveys.

Table S10. Periods of data collection and survey participants in the CoMix study in France.

Survey wave	Survey participants	Survey period
A1	Adults	December 21-29, 2020
A2	Adults	January 7-12, 2021
A3	Adults	January 20-22, 2021
C1	Children (< 18 y.o)	February 3-9, 2021
A4	Adults	February 17-20, 2021
A5	Adults	March 3-8, 2021
A6	Adults	March 17-20, 2021
C2	Children (< 18 y.o)	April 2-8, 2021
A7	Adults	April 14-15, 2021

We estimated 2 CoMix contact matrices, by pulling together the results of the closest adult-child survey waves (therefore, we merged A4-C1 and A7-C2 to estimate a full contact matrix). We considered only contacts on regular weekdays (Monday-Friday excluding holidays), to ensure comparability with our main analysis. We compared the resulting CoMix matrices with the weekly synthetic matrices for the week corresponding to the CoMix children wave (i.e., February 2021 (w5) and April 2021 (w13)) and with the closest SocialCov matrices (i.e., January 2021 and March 2021). The matrices are illustrated in **Fig. S15**.

Figure S15. Illustrations of contact matrices.

We computed the average number of contacts, overall and by age group. We found that participants in the CoMix survey reported substantially fewer contacts compared to those in the SocialCov survey (**Fig. S16a**). The average number of contacts was around $\frac{1}{3}$ in the CoMix matrices compared to SocialCov matrices (**Fig. S16c**). Fewer contacts were reported in all age groups (**Fig. S16d-g**), and the strongest difference was measured in children (around $\frac{1}{4}$, **Fig. S16c**). Compared to synthetic matrices, the number of contacts in CoMix was lower in adults and seniors, and comparable in children and adolescents (**Fig. S16b**).

We also computed the four metrics of comparison (cosine similarity, assortativity, young connectivity and age-specific fraction of assortative contacts). The cosine similarity between the CoMix matrices and the two other sources (either synthetic or SocialCov matrices) was comparable (**Fig. S16l**). Contacts in the CoMix matrices were substantially more assortative than SocialCov matrices, and they were more similar to synthetic matrices in terms of overall assortativity and age-specific proportion of within-group contacts (**Fig. S16h-k**, **Fig. S16m**). However, CoMix and SocialCov matrices were similar in terms of young connectivity (around 50%) in contrast with synthetic contact matrices (around 25%, **Fig. S16m**).

In conclusion, our analysis showed that CoMix matrices reported substantially fewer contacts than SocialCov, particularly in children, while their structure was more similar to synthetic matrices in terms of assortativity and age-specific within-group mixing. Still, young individuals contributed to overall connectivity in CoMix matrices at levels comparable to SocialCov and higher than in synthetic matrices.

Figure S16. Comparison of contact volumes and matrix structure of two CoMix matrices with the two closest synthetic and SocialCov matrices.

How was the serological status calculated for vaccinated individuals?

In our model, the proportion of AB+ (Fig. 6) reflects infection-derived immunity only. We did not compute the serological status for vaccinated individuals as we limited our validation to serological data to the period before vaccination started. This is because serologically tested populations were not asked whether they were vaccinated or not, and the test did not distinguish between antibodies acquired from vaccination or infection. We considered data points from March 2020 to June 2021 for children and adolescents, and data points from March 2020 to February 2021 for adults and seniors. **This is now explained in the Methods to improve clarity (lines 510-514):**

Since the IgG test does not distinguish between antibodies from infection and vaccination, and no information on vaccination status was collected from serologically tested individuals, we restricted our analysis to seroprevalence data collected before the start of the vaccination campaign. This ensures comparability with model estimates of antibody-positive individuals, which reflect infection-derived immunity only.

Minor comments

Line 55: It is stated that (in the CoMix survey) only UK, Belgium and the Netherlands covered the first pandemic wave (also expressed in the reference that it cited). However, Norway was part of the CoMix survey and conducted six waves between April and September 2020, <https://bmcpublichealth.biomedcentral.com/articles/10.1186/s12889-024-18853-8>

We thank the reviewer for spotting this. We have now included Norway in the sentence, and we have provided the corresponding reference.

Line 76 “accuracy” relative to empirical contact matrices. I would rephrase as both the synthetic and empirical matrices are estimates.

We thank the reviewer for the opportunity to rephrase. This issue was also raised by reviewer #1 (medium issue). We report here the same reply as provided to reviewer #1.

Empirical matrices are widely considered the conventional approach for estimating contact patterns in infectious disease modeling. However, in our study, they are not treated as a “gold standard,” as they come with notable limitations — most importantly, their low temporal resolution — which hinder their suitability as a definitive benchmark for evaluating time-varying contact structures.

This is now made clear in the revised manuscript, together with the following two research questions, explicitly stated in the revised Introduction:

- 1) Compare the time-varying contact matrices generated using mobility-informed synthetic methods with empirical matrices derived from repeated contact surveys, focusing on differences in contact volumes and mixing patterns across age groups.
- 2) Assess the capacity of each matrix to support epidemic modeling of COVID-19 and to reproduce observed hospitalizations, infections, and age-specific trends. To this end, each matrix is embedded in an age-structured transmission model calibrated on hospitalization data and informed by real-world information on viral variants, vaccination, and immunity.

Our aims are now more explicitly presented in the abstract as well.

We report here the changes made to the Introduction (lines 78-90)

Although mobility-based synthetic contact matrices offer a promising alternative for real-time modeling, their comparative performance relative to empirical contact matrices remains unexplored. Meanwhile, traditional empirical contact matrices - though widely used - are limited by infrequent updates, which constrains their applicability for dynamic modeling. This study addresses two main objectives. First, we compared contact patterns derived from mobility-informed synthetic matrices [37] and from empirical matrices computed from social contact surveys [29,30], focusing on age-specific contact volumes and mixing patterns over time. Second, we assessed the capacity of each matrix type to support epidemic modeling of COVID-19 in France by embedding them in an age-stratified transmission model calibrated on hospitalization data and informed by real-world information on viral variants, vaccination, and immunity. This framework enables us to evaluate how well each matrix captures disease-relevant contacts over time to reproduce observed indicators, including hospitalizations and age-specific infections. By using the COVID-19 pandemic in France as a case study, we provide insight into how different matrix sources can inform real-time transmission modeling for future outbreak response.

Suppl. L 316. The matrix obtained from the first wave of data collection is named LD, while the national lockdowns are named LD1-LD3. Does this imply that this matrix was also used for later national lockdowns?

We thank the reviewer for spotting this. No, in the model informed by survey-based matrices, the empirical matrix called “LD” was only used for the first lockdown. For the second and third lockdowns, we used extended empirical contact matrices as detailed in Fig. 6b. Schools were kept open and social distancing restrictions were less stringent compared to the first lockdown, therefore the empirical matrix estimated during the first lockdown would not be representative for these periods. To avoid confusion, we have renamed the empirical matrix from the first wave as “LD1”.

REVIEWER #3

This manuscript addresses an important question about whether it is sufficient to use ‘synthetic’ contact matrices, constructed using mobility and other publicly available data, or if it is essential to undertake resource-intensive efforts to collect empirical contact data to capture changing behaviors. The authors address this question in the context of the COVID-19 pandemic, though their findings are likely generalizable to other circumstances in which behavior may be changing substantially over the course of an epidemic. The authors undertake a detailed assessment of this question, by descriptively comparing contact matrices generated by each approach, evaluating the extent to which different contact assumptions align with hospitalization data and seroprevalence estimates from 2020-22. This is a thorough, clear, and important paper. I have a few minor comments for the authors to consider.

We greatly appreciate the reviewer's positive assessment of our manuscript. We provide here below our answers to the minor points.

Minor points

1. Some discussion of the range and distribution of contacts reported, and if / how this was taken into account in the modeling, should be included. The matrices are nicely shown in Fig 1, but all that is shown is the averages (and averages by age group). Were the full distributions of contact accounted for in the modeling, or were only the average rates by age used? I think the latter is defensible, but for an airborne infection such as SARS-CoV-2 where contact heterogeneity (‘superspreading’) may play an important role in transmission dynamics, the distributional assumptions should be noted.

We thank the reviewer for the comment. We only considered average rates, as commonly done in age-stratified compartmental models which do not account for individual-based superspreading events. **We have included this as a limitation of the study in the Discussion section** (lines 349-350):

Relatedly, we used average contact rates and did not account for individual heterogeneity in contacts or superspreading events.

2. Were there any important differences (contact definition, mode of survey, survey structure, etc.) in the data collection for the empirical contact studies during the pandemic and the study from which the pre-pandemic baseline was drawn for the synthetic matrices (2012 survey)? In the author's opinion, would these design differences contribute to the differences in fit under the synthetic vs. empirical data?

The definition of “contact” was the same in the two surveys. There were differences in the sampling method and representativeness of the surveyed sample, and in the survey structure. The original survey used a representative sample of the population (recruited from the market research company IPSOS), and asked participants to list their contacts individually. The SocialCov survey used online convenience sampling, and corrected for representativeness; contacts were collected in an aggregated form.

The survey mode used for reporting contacts may impact the estimated contact matrices. As mentioned in the Discussion (lines 317-321):

Second, SocialCov surveys relied on aggregated contact reporting by age group rather than individual contact listings, which may have facilitated reporting – especially for parents reporting for their

children – at the cost of accuracy. Previous work has shown that aggregated formats tend to report higher contact numbers compared to detailed individual listings [27].

An overestimation of contacts in children in the SocialCov may have contributed to the difference in the model performance. However, our analysis suggests that not only the contact structure, but also the frequency of updates in the matrices is key, as weekly synthetic contact matrices allowed to better capture the epidemic trajectory, compared to sparse empirical matrices derived from infrequent contact surveys.

3. Please provide an explanation of ‘cosine similarity’. I am unfamiliar with this method and a brief explanation would help a reader to understand how to best interpret this analysis.

We thank the reviewer for this comment, which was also mentioned by reviewer #1. The formal definition of the cosine similarity is reported in the Supplementary Information (**Section 5.2**). **We have now provided a clearer explanation of the cosine similarity and the interpretation of the analysis in the Results section** (lines 148-154):

Cosine similarity — a metric defined between 0 and 1 assessing the similarity in mixing structure, independent of overall contact volume — was higher during periods of school closure, indicating a larger agreement in age-specific mixing patterns between the two matrix sources. When schools were open, particularly in March 2021, similarity dropped, reflecting greater deviations in mixing among young age groups. Notably, cosine similarity between empirical and synthetic matrices was consistently higher than that between synthetic matrices and random mixing, reinforcing that synthetic matrices retain meaningful age structure beyond proportional mixing.

In addition, prompted by reviewer #1, in the revised version of our manuscript we computed the cosine similarity between the synthetic matrix and TEST2 (random mixing contact matrix). Results are illustrated in **Fig. S12a**.

4. I have not heard of the Normalcy Index (only the Stringency Index), but it is the only one shown in Fig 2c as a comparison to the synthetic contact rates. Can the Stringency index be shown as well? I understand this to be the more commonly used measure. If the authors would like to show the Normalcy Index, some additional explanation of how the indicator is constructed would be useful. Are the ‘indicators’ being used to construct the Normalcy Index similar to those being used to generate the synthetic contact patterns? If so, we would already expect them to be correlated.

As described in the Methods, the Normalcy Index is a measure of the impact of the pandemic on human behavior, integrating multiple daily indicators of human activities in a score from 0 to 100, with 100 representing the pre-pandemic level. More in detail, the Normalcy Index tracks eight variables (sports attendance, time at home, traffic congestion, retail footfall, office occupancy, flights, film box office and public transport) (Ref. <https://www.economistgroup.com/press-centre/the-economist/the-economist-launches-normalcy-index-to-quantify-the-return-of-pre-pandemic>). Some of these indicators could be related with the indicator of Google mobility related to workplaces used for the synthetic matrices, although the definitions do not completely overlap; besides, the Normalcy Index does not take into account changes in avoidance of physical contacts, which is another main indicator used in the construction of the synthetic contact matrices.

Unlike the Stringency Index, which reflects the strictness of implemented policies, the Normalcy Index captures adaptive behavioral changes and population response. As a result, two periods with identical Stringency Index values may exhibit different Normalcy Index levels, depending on how the population actually adjusted its behavior. This was observed for example during the second lockdown in France (Di Domenico et al., Comms Med 2021). For this reason, we compared the average number

of contacts to both the Normalcy and the Stringency Index, as they represent two different quantities. **This is now better explained in the manuscript, Results section** (lines 120-126):

To contextualize these dynamics, we examined correlations with the Normalcy Index, which captures behavioral adaptation, and with the Stringency Index, which quantifies the intensity of social distancing measures. The synthetic contacts time series was highly correlated with both indices (Normalcy Index: Pearson's coefficient = 0.84, p-value < 10⁻¹⁹⁹, Fig. 3c, S12b; Stringency Index: Pearson's coefficient = 0.82, p-value < 10⁻¹⁹⁶, Fig. S12b), supporting the plausibility of the behavioral signals integrated into the synthetic matrices. This correlation reflects concurrent behavioral shifts rather than overlap in data inputs.

and Methods section (lines 428-432):

It is important to note that the Normalcy Index and the indicators used for constructing synthetic matrices are not the same: the Normalcy Index aggregates eight broad measures of societal activity (e.g. retail, transport, leisure, office occupancy), while our synthetic matrices mainly relied on workplace mobility and physical contact avoidance. The latter indicator has no equivalent in the Normalcy Index.

5. Lovely figures, but I think Figure 3 is largely redundant with Figure 2. Would recommend placing Figure 3 in the Supplement.

We thank the reviewer for the suggestion. **However, we note that there is no redundancy.** As explained to reviewer #1, Fig2 displays comparison in terms of total number of contacts per age group, while Fig3 compares contact mixing patterns accounting for matrix elements and structure. For this reason, and given the comments of the other reviewers, we decided to keep this figure in the main text. We included a sentence in the Results section to highlight the two types of comparison (lines 115-116 and 136-137):

We first compared the two sources of matrices in terms of average number of contacts (overall and by age group).

[...]

We then compared contact matrices in terms of mixing patterns using a set of metrics accounting for matrix elements, structure, and contact distribution.

6. I do not understand Figure 3D; more explanation may be required.

We apologize for the lack of clarity. As stated in reply to a previous reviewer, **we have now included additional explanation in the Results section. We also moved this panel to the Supplementary Information (Fig S12b), and replaced the figure in the main text with illustrations of the proportion of within-group contacts (a more straightforward measure of assortativity).**

Supplementary Information (section 8.1)

In Fig. S12, we compare the matrices using different metrics [...]. Compared to the pre-pandemic contact matrix, synthetic matrices were slightly less assortative, while maintaining a similar proportion of contacts established by young individuals (around 25%, Fig. 12b). In contrast, empirical matrices showed larger differences with the pre-pandemic and synthetic contact matrices, with lower overall assortativity and a higher share of contacts generated by individuals under 19 years of age (about 50%, Fig. 12b).

Results (lines 144-146)

Moreover, in the empirical matrices, young individuals (i.e. children and adolescents) contributed to 50% of total connectivity, twice as in the synthetic contact matrices (Fig. S12b).

7. A comment - the authors note this in their Discussion, but I feel that setting this up as a comparison of the empirical and synthetic contact data is somewhat unfair, since the empirical studies were conducted so sparsely across the pandemic. In fact, the empirical studies seem to do surprisingly well given the low frequency sampling.

We thank the reviewer for the comment. This issue was also raised by reviewer #1 (medium issue) and reviewer #2 (minor comment).

As explained in reply to reviewer #1 and #2, empirical matrices are widely considered the conventional approach for estimating contact patterns in infectious disease modeling. However, in our study, they are not treated as a “gold standard,” as they come with notable limitations — most importantly, their low temporal resolution — which hinder their suitability as a definitive benchmark for evaluating time-varying contact structures.

This is now made clear in the revised manuscript, together with the following two research questions, explicitly stated in the revised Introduction:

- 1) Compare the time-varying contact matrices generated using mobility-informed synthetic methods with empirical matrices derived from repeated contact surveys, focusing on differences in contact volumes and mixing patterns across age groups.
- 2) Assess the capacity of each matrix to support epidemic modeling of COVID-19 and to reproduce observed hospitalizations, infections, and age-specific trends. To this end, each matrix is embedded in an age-structured transmission model calibrated on hospitalization data and informed by real-world information on viral variants, vaccination, and immunity.

Our aims are now more explicitly presented in the abstract as well.

We report here the changes made to the Introduction (lines 78-90)

Although mobility-based synthetic contact matrices offer a promising alternative for real-time modeling, their comparative performance relative to empirical contact matrices remains unexplored. Meanwhile, traditional empirical contact matrices - though widely used - are limited by infrequent updates, which constrains their applicability for dynamic modeling. This study addresses two main objectives. First, we compared contact patterns derived from mobility-informed synthetic matrices [37] and from empirical matrices computed from social contact surveys [29,30], focusing on age-specific contact volumes and mixing patterns over time. Second, we assessed the capacity of each matrix type to support epidemic modeling of COVID-19 in France by embedding them in an age-stratified transmission model calibrated on hospitalization data and informed by real-world information on viral variants, vaccination, and immunity. This framework enables us to evaluate how well each matrix captures disease-relevant contacts over time to reproduce observed indicators, including hospitalizations and age-specific infections. By using the COVID-19 pandemic in France as a case study, we provide insight into how different matrix sources can inform real-time transmission modeling for future outbreak response.

The Discussion has also been revised (lines 263-282):

All models well captured the overall trajectory of total hospitalizations, and results remained robust when using age-stratified hospitalizations as calibration targets. [...] the model informed by weekly synthetic contact matrices outperformed those using empirical or pre-pandemic matrices [...] The higher performance of the synthetic matrices in these contexts originates from their key advantage, i.e. the ability to be updated weekly. This update is essential for real-time modeling applications, supporting improved situational awareness, scenario planning, and timely policy response. In contrast, empirical matrices generally have limited temporal resolution, therefore requiring strong

assumptions to fill gaps between infrequent survey waves. [...] While more frequent surveys could improve the empirical matrix model, they come with substantial resource demands and require automated tools and streamlined methods to transform survey data into actionable insights efficiently[23,48]. Alternative methods for temporally extending empirical matrices may hold promise, but they are often context-specific and remain an open area for future research.

8. From Figure 4a, it appears that all of the contact assumptions allow for a reasonably good fit to hospitalization data. How do the authors reconcile this with the quite large discrepancies between the synthetic and empirical contact rates? Can this be attributed to the low rate of hospitalization among kids, among whom the discrepancy between synthetic and empirical rates was highest?

We thank the reviewer for this comment. A similar comment was mentioned by reviewer #1 (major issue). We want to clarify that the ability of all models to reconstruct total hospitalizations reasonably well is due to the modeling framework itself. In particular, the correcting factor was fitted over short, discrete time windows (averaging 3 weeks), which naturally limits the accumulation of large fitting errors in total hospitalizations. In this sense, the structure of the fitting approach plays a role in enabling good agreement with aggregate outcomes. **We have added this element to the Discussion.** However, we argue that, despite the same time-discretization applied across models, the mean absolute error and AIC of the model using empirical matrices remained significantly worse than those of the synthetic model. Moreover, we also showed that the differences in matrix structure between the two approaches (synthetic and empirical) led to substantial discrepancies in the age-stratified model outcomes, namely hospitalizations and infections by age group (Fig. 5d, Fig. 6c-f).

We report here the changes made to the manuscript.

Discussion (lines 263-269, lines 302-308)

All models well captured the overall trajectory of total hospitalizations, and results remained robust when using age-stratified hospitalizations as calibration targets. This consistency comes from the inference framework, which relies on short, discrete time windows for estimating the correcting factor, limiting the accumulation of fitting errors and allowing sufficient flexibility to align model outputs with observed data. Despite this flexibility, the model informed by weekly synthetic contact matrices outperformed those using empirical or pre-pandemic matrices, achieving better fit metrics (AIC and mean absolute error).

[...]

Differences between synthetic and empirical contact matrices were most pronounced in children and adolescents. Empirical matrices consistently reported higher contact rates for these age groups, particularly during school-open periods and in within-group contacts, contributing to an overestimation of seroprevalence in both children and adolescents. In contrast, the synthetic matrices underestimated infection levels in children but provided a closer fit to adolescent serological data during certain periods. These discrepancies reflect the challenge of accurately capturing school-related and intergenerational mixing dynamics. They could be explained by a set of factors [...]

Overall, an important contribution to the literature on this topic.

We thank the reviewer for their positive assessment of our manuscript.

REBUTTAL APPENDIX

Figure R1. Age-stratified correcting factor. Histogram of the values occurring in the top 50 best combinations in terms of likelihood.

Figure R2. Comparison of TEST1 model results when using total or age-stratified calibration targets.

Figure R3. Comparison of TEST2 model results when using total or age-stratified calibration targets.

Figure R4. Comparison of TEST3 model results when using total or age-stratified calibration targets.

Figure R5. Comparison of TEST4 model results when using total or age-stratified calibration targets.

POINT-BY-POINT REPLY TO REVIEWERS' COMMENTS

Mobility-driven synthetic contact matrices: a scalable solution for real-time pandemic response modeling

Laura Di Domenico, Paolo Bosetti, Chiara E. Sabbatini, Lulla Opatowski, Vittoria Colizza

Reviewer #1

Summary

The authors have done extensive edits and additional simulations, which have fully addressed the medium/minor issues, and mainly addressed the major issues. Some issues remain.

We thank the reviewer for acknowledging the value of our extensive revisions. We are glad that these revisions successfully addressed the majority of the issues. We provide below specific replies to the remaining issues.

Major Issues

+ Inference framework for aim (ii):

- Interpretation of correcting factor (α): Following new sensitivity analyses of several alternate synthetic contact matrices, where α distributions remained similar to those in main analyses, the authors conclude: "the correcting factor is not a reliable indicator of contact matrix performance." I disagree -- this was not my concern with α : values near 1 would indeed indicate contact matrices are "correct" given the model & calibration targets. Rather, I still feel the issue here is that the calibration targets provide little information / error signal regarding mixing *patterns*, especially in the main paper, without age-stratification.

We apologize, this was a mistake in reporting this piece of text in the previous rebuttal. We do agree with the reviewer that the correcting factor (and its distance from 1) is an indicator of the global adjustment needed for the contact matrices. This interpretation was indeed included in the revised manuscript as follows:

Methods: lines 541-547

This correcting factor can be thus interpreted as a global correction of the contact matrices used in the model. The product of the fitted α_{phase} at time t with the corresponding contact matrix represents the effective contacts needed to reproduce the observed epidemic dynamic. The closer the correcting factor to 1, the better the alignment between the time-varying contact matrices and the effective contact rates needed to reproduce the total epidemic dynamic. However, acting as a global rescaling of the contact matrix, the correcting factor cannot alter the underlying matrix structure and mixing, responsible for the age-stratified model outcomes.

In addition, the conclusion of our sensitivity analyses was that the correcting factor is not a standalone reliable indicator of fit quality, in the sense that two models may have an associated α distribution with a similar distance from 1, while having a different performance, measured by the AIC and the validation of age-stratified model outcomes. This was explained in the revised manuscript as follows:

Results: lines 222-225

Across all tests, the correcting factor remained comparable to that of the model parameterized with the original synthetic contact matrices. However, substantial differences

emerged in age-stratified outcomes – both for hospitalizations and seroprevalence – highlighting that the correcting factor alone cannot compensate for structural changes in contact patterns.

Discussion: lines 293-297

Crucially, our tests showed that altering the structure of the contact matrices had substantial effects on age-specific outcomes even when the correcting factor remained stable, underscoring that this factor alone cannot compensate for mis-specification of mixing patterns. While it provides flexibility for the fit, it does not mask structural limitations in the underlying contact data. These results therefore also underscore the central role of matrix structure in shaping model projections.

- Calibration targets: I still feel that all available age-stratified data (hospitalizations & seroprevalence) should be used as calibration targets, and in the main analyses (not just supplement). The justification about mimicking real-time modelling does not make sense to me, because it discards information we do actually have now, that could better answer the overarching question before the next epidemic (whether new contact surveys are needed during the epidemic). This real-time framing is also not really internally consistent, given the need to fit alpha using hospitalization data from multiple weeks into the future.

Done. We have updated our main analysis using all available age-stratified data (both hospitalizations and seroprevalence) as calibration targets and included the results in the main text. We moved the previous results, based solely on hospitalization data (either age-stratified or not), to the Supplementary Information. We also updated all the sensitivity analyses (Section 9 of the Supplementary Information) using age-stratified hospitalizations and seroprevalence data as calibration targets. The results remain consistent with our original study.

Importantly, the results show that serological data are not critical for the real-time use of synthetic matrices — a reassuring finding given that such data were not available in France during the pandemic and are not currently planned for future crises.

This is now mentioned in the Results (lines 210-216):

We tested alternative calibration strategies, fitting the model using (i) age-stratified hospitalizations only or (ii) total hospitalizations, while reserving age-stratified serological and hospitalization data for validation (Figs. S17-S21). In both cases, model performance, the distribution of the correcting factor and the outcomes of model comparisons remained robust to the choice of calibration targets. The model informed by the synthetic contact matrices reproduced the main age-stratified hospitalization and serological patterns—though systematic deviations persisted for children—even when age-stratified hospitalization and serological data were not used for fitting.

And in the Discussion (lines 297-303):

Importantly, model comparisons and correcting-factor estimates remained consistent when fitting to different calibration targets—including total hospitalizations, age-stratified hospitalizations, or both age-stratified hospitalization and serological data—indicating that our conclusions do not depend on the specific choice of calibration inputs. Notably, incorporating serological data did not materially alter the performance of the synthetic matrices, suggesting that these matrices remain informative for real-time use even in settings where serological surveillance is absent.

Regarding the real-time framework, our modeling approach remains internally consistent for synthetic matrices: each time window is fitted sequentially, without assuming knowledge of future hospitalization data. This feature is essential for frameworks intended for response in future emergencies, and it is now more clearly explained in the Methods. In contrast, when using empirical matrices we break this real-time framework, since we use later survey waves to fill earlier gaps (e.g., during school holidays), highlighting an inherent limitation of survey-based approaches for real-time response. This limitation was already discussed in the Discussion section (lines 271-273).

+ Sensitivity analyses contact matrices in aim (ii):

- More intuitive names for cases would be very helpful, e.g. TEST1: "adult-,elderly+", TEST2: "random mixing", TEST3: "less within-group", TEST4: "keep physical" etc.

Done. We changed the notation in figures and text.

- Random mixing: For TEST2, "random mixing" should not modify the total numbers of contacts per group. If $X_i = N_i * C_i$ is the total numbers of contacts "offered" by group i , then $X_{ij} = \text{outer}(X_i, X_i) / \text{sum}(X_i)$ gives the random mixing matrix at the population-level, and $M_{ij} = X_{ij} / N_i$ gives the per-person matrix. This is notable because TEST2 performed much worse than the main synthetic or TEST3 matrices, likely due to changing the numbers of contacts per group, but a comparison with random mixing that maintains group-specific contacts would be very useful.

Done. Our original TEST2 considered a scenario where a contact between any two individuals occurs randomly with equal probability (often called "homogeneous mixing" in the literature). We have now carried out an additional sensitivity analysis with a new sequence of random mixing matrices, as defined by the reviewer, where the total number of age-specific contacts are preserved, but redistributed proportionally to the contacts offered by each group. The results of this new test (TEST2: random mixing) have been included in **Fig. S25-27**. Similarly to TEST3 (less assortative), also TEST2 (random mixing) produced higher hospitalizations in seniors and lower hospitalizations in adults compared to observations (**Fig. S27a**), and also led to lower predicted infections in adults and in adolescents relative to serological data (**Fig. S27c,d**). In terms of AIC, TEST2 (random mixing) performed worse than the original synthetic model (**Table S15**). Thus, the conclusions of this new sensitivity analysis are robust with our previous findings. The detailed presentation of the test results is provided in the Supplementary Information, and mentioned in the main text (lines 219-222).

- Reciprocity: Throughout the additional contact matrices explored (e.g. TEST 1-3, Supp 9.4), it's not clear if resulting contact matrices are re-balanced for **reciprocity** after modifications. Contact matrices violating reciprocity will effectively cause modelled infections to be transmitted "into the void" or acquired "from the void" for various groups, depending on the direction of imbalance -- which evidently undermines the objective of assessing contact matrix performance.

* I suspect matrices have not been re-balanced because, in TEST3, it is not possible to reduce within-group contacts by 25% for every group simultaneously, while maintaining reciprocity. For example, in the 2x2 case: suppose $N_i = [80, 20]$, $C_i = [1, 6]$. Under random mixing (without loss of generality) we have $X_{ij} = [[32, 48], [48, 72]]$ at the population-level. If we reduce the diagonal 32 and 72 each by 25% (yielding 24 and 54), the missing contacts to add back to the off-diagonals are 8 and 18, which would violate reciprocity since X_{ij} must remain symmetric.

In all tests, where needed, we corrected for reciprocity. For example, for TEST3, we first reduced within-group contacts by 25%, then we added the missing contacts outside of the diagonal, and finally we corrected for reciprocity, using the following adjustment $M_{ij}(\text{rec}) = (M_{ij} * N_i + M_{ji} * N_j) / (2 * N_i)$. We acknowledge that the reciprocity correction will slightly alter the age-specific total number of contacts, because it operates on off-diagonal elements referring to different rows. However, the variations remain small (around 10%). This is now mentioned in the text and shown in **Fig. S22a,b** for TEST1 (adults+, seniors-) and **Fig. S25a-d** for TEST3 (less assortative).

+ The added results on element-wise comparison of contact synthetic vs empirical matrices for aim (i) is great.

Thank you.

Medium Issues

- Figure 2: This is a great addition. However, I think appending the absolute & relative differences as shown in Figures S10 & S11 to Figure 2 would significantly improve the ease of appreciating differences among contacts within the main text. Relatedly, the colour scales should be symmetric (so +1 vs -1 absolute differences are the same intensity, and likewise for 2 vs 1/2 relative differences).

Done. We added the absolute and relative differences figures to Figure 2 of the main text (panels b and c), as suggested by the reviewer. We also adjusted the colour scales to ensure symmetry.

- [182] Regarding: "empirical and pre-pandemic matrix models showed abrupt and unrealistic shifts in epidemic trends" - is this not just the result of modelling of alpha as piecewise constant, rather than deficiencies in the empiric data themselves? It could likely be solved using a spline for alpha instead, with knots at the current intervals. I would rephrase this.

We note that the synthetic model, which successfully captured smooth transitions, also relied on a piecewise definition of the correcting factor. Therefore, the abrupt shifts observed in the empirical and pre-pandemic matrix models cannot be attributed solely to the piecewise formulation of this factor. Rather, they reflect limitations of the empirical and pre-pandemic matrix in capturing weekly updates of behavioral adaptations. Nonetheless, we have included a sentence in the discussion (limitations) to acknowledge that this issue could, in principle, be mitigated by adopting a continuous functional form for the correcting factor, such as a spline.

Discussion, lines 368-374:

Finally, we used a piecewise constant function for the correcting factor. The abrupt changes in epidemic trends produced by the empirical and pre-pandemic matrix models could, in principle, be mitigated by adopting a continuous functional form for the correcting factor, such as a spline. Nevertheless, our findings show that the synthetic model, despite relying on a piecewise definition of the correcting factor, successfully captured smooth transitions, indicating that the observed discontinuities arise from limitations in the empirical and pre-pandemic matrices themselves.

- The manuscript makes several references to "infections" as a modelled outcome to compare with data, but the available data (and actual modelled outcome) reflect seroprevalence, which can wane over time due to seroreversion, so is not synonymous with cumulative infections. The more precise term "seroprevalence" should be used throughout.

Done. We changed the notation in the manuscript, using "seroprevalence" when referring to the corresponding model outcome.

- GitHub: I cannot find the code which actually runs the model fitting. Is alpha for each phase calibrated separately, or jointly?

The correcting factor alpha is calibrated sequentially in each phase, building on the results of the previous phases (**Table S8**). We added a script with the model fitting in the GitHub repository.

Minor Issues

- I appreciate the addition of explicit aims at the end of the introduction. However, I feel the language used lacks precision: what are "contact patterns" and "contact volumes" - why not use language defined above: "contact matrices"? e.g. "First, we compared *synthetic* contact matrices derived from mobility data versus *empirical* matrices derived from social contact surveys..." and to sign-post the key words "synthetic" and "empirical" used throughout. Similarly, what is "capacity of each matrix type to support epidemic modelling of covid-19 ..." - why not "reproduce (age-specific) covid-19 hospitalization (and seroprevalence) data within a transmission modelling framework"?

Done. We revised the text as follows (lines 81-89):

First, we compared synthetic contact matrices informed from mobility and behavioral data and empirical matrices derived from social contact surveys, focusing on age-specific contact numbers and matrix structure over time. Second, we assessed the capacity of each matrix type to reproduce age-specific hospitalization and seroprevalence data within a COVID-19 transmission model accounting for viral variants, vaccination, and immunity. This framework enables us to evaluate how well each matrix captures disease-relevant contacts over time. By using the COVID-19 pandemic in France as a case study, we provide insight into how different matrix sources can inform real-time transmission modeling for future outbreak response.

- Figure 2: are these data before or after correcting for reciprocity?

The matrices in Fig. 2 are corrected for reciprocity. This is now specified in the caption.

- [resp.p15] "We also argue that the ability of all models to reconstruct total hospitalizations is not primarily due to the flexibility of alpha, but rather to the modeling framework itself. In particular, alpha was fitted over short, discrete time windows (averaging 3 weeks), which naturally limits the accumulation of large fitting errors in total hospitalizations." Since alpha is essentially the only fitted parameter throughout the epidemic, I do not understand the point being made here.

This point is explained clearly in the discussion (lines 174-177):

We evaluated each matrix type based on model fit quality. All models captured the observed trajectory of daily hospital admissions well (Fig. 5a, Fig. S3). This consistency reflects the use of short fitting windows (averaging three weeks), which allow flexible alignment with the hospitalization curve. However, the quality of fit varied across models [...].

- [resp.p18] Regarding age-specific correcting factors: To reduce overfitting, multiple group-specific calibration targets can be used together (e.g. hospitalizations, seroprevalence, etc.), alongside informative priors for fitted parameters (e.g. log-normal with mean 1 for alphas), with iterative sampling methods to improve inference efficiency (e.g. doi.org/10.1111/j.1541-0420.2010.01399.x). Relatedly, such factors do not need to distort mixing, since mixing preferences can be maintained even with changing group-specific marginal contact rates (e.g. doi.org/10.1016/0025-5564(91)90014-a). So, age-specific alphas are indeed feasible and would be useful.

Thank you for the suggestions. We now mention this point in the discussion, in the paragraph concerning the correcting factor (lines 303-306):

Future research could investigate the feasibility of fitting age-specific correcting factors that change age-specific total contacts while maintaining mixing preferences; although feasible in principle, achieving parameter identifiability would likely require advanced iterative sampling methods [Irons & Raftery, 2021].

- [resp.p18] Relatedly: "Using more age groups would have forced us to introduce additional assumptions to extrapolate or impute this missing behavioral information, thereby reducing the empirical grounding of the matrices." Sorry, this is not true: you have already made the same number of assumptions, they are just grouped together and less explicit (e.g. all individuals aged 19-65 are identical). In other words, when counting unique assumptions, some are "bigger" than others.

It is true, we agree with the reviewer. In the manuscript, we discuss this aspect as a limitation, and modified the text as follows (lines 347-352):

First, the mobility and behavioral data used to construct synthetic matrices (e.g., workplace attendance, preventive behavior) were not age-stratified, but some of these proxies indirectly introduced age specificity. However, no finer age breakdown was available. Relatedly, while additional age classes with the same assumptions were possible, we limited our synthetic matrices to four broad age classes; future work could assess a better trade-off between detail and real-time feasibility. Generating age-detailed synthetic matrices in real time remains constrained in the absence of age-stratified mobility and behavioral data, which would be essential to refine assumptions and capture more accurate age-specific changes in contact patterns over time.

- The supplement is missing page numbers.

Done.

Remarks on code availability

I could not find the code which actually runs calibration / model fitting on github - e.g. the "likelihood" function is defined but not called, and data is loaded from a file calibration.csv. Some parameters seem to have different names from the paper, which is confusing, but not terrible. I did not try running the code myself.

We included a script that runs the calibration on the GitHub repository.

Reviewer #2

The authors have conducted a thorough revision of the manuscript, improving its clarity, exposition, and methodological descriptions, and adding analyses that further support their conclusions. I find the article substantive and offering novel insights into the complex challenge of representing contacts in infectious-disease models, with clear relevance for real-time crisis response.

All of my prior comments have been addressed appropriately, and I have no further comments.

We thank the reviewer for their positive assessment of our revised manuscript.

Reviewer #3

This is a thorough response and the authors have adequately addressed my previous concerns.

Remarks on code availability: I reviewed the code previously.

We thank the reviewer for their positive assessment of our revised manuscript.